

**The 1538 eruption at Campi Flegrei resurgent caldera: implications for future unrest and**
**eruptive scenarios**
**Giuseppe Rolandi[1], Claudia Troise[2], Marco Sacchi[3], Massimo di Lascio[4], Giuseppe De Natale[2]**
[1] Università di Napoli Federico II, Dept. Earth Sciences, Naples (I)
[2] Istituto Nazionale di Geofisica e Vulcanologia, Osservatorio Vesuviano, Naples (I)
[3] ISMAR-CNR, Naples (I)
[4] Free Lance Geologist, Naples (I)
Corresponding author: Giuseppe De Natale, giuseppe.denatale@ingv.it
Abstract
The recent unrest in the Campi Flegrei caldera which began several decades ago, poses a high risk to
a densely populated area, due to significant uplift, very shallow earthquakes of intermediate
magnitude and the potential for an eruption. Given the high population density, it is crucial, especially
for civil defense purposes, to consider realistic scenarios for the evolution of these phenomena,
particularly seismicity and potential eruptions. The eruption of 1538, the only historical eruption in
the area, provides a valuable basis for understanding how unrest episodes in this caldera may evolve
toward an eruption. In this paper, we provide a new historical reconstruction of the precursory
phenomena of the 1538 eruption, analyzed considering recent volcanological observations and results
obtained in the last few decades. This allows us to build a coherent picture of the mechanism and
possible evolution of the present unrest, including expected seismicity, ground uplift and eruptions.
Our work identifies two main alternative scenarios, providing a robust guideline for civil protection
measures, and facilitating the development of effective emergency plans in this highly risky area.
**1. Introduction**
The Campi Flegrei area has been a benchmark of modern geology and volcanology since the middle
XVIII century, due to the clear evidence of significant ground movements, associated with both uplift
and subsidence, imprinted on the columns of the ancient Roman Market (Macellum) in the town of
Pozzuoli. These movements were famously depicted on the cover of Charles Lyell's seminal book,
'Principles of Geology'.. By the XIX century, it became evident that the impressive relative





movements between sea level and ground were due to ground uplift and subsidence. Consequently,
numerous efforts have been made to reconstruct the timeline of these movements, during the
centuries,. One of the most convincing reconstructions was proposed by Parascandola (1947), later
modified by Dvorak and Mastrolorenzo (1991), Morhange et al. (2006), Bellucci et al. (2006) and,
more recently, Di Vito et al. (2016). However, all these reconstructions exhibit evident discrepancies,
and do not rely on the full body of historical evidence, as we will demonstrate. These significant
ground movements have predominantly involved a long-term trend of subsidence, punctuated by
occasional episodes of rapid ground uplift, culminating in the only eruption occurred in historical
times, in 1538 (Di Vito et al., 2016). After the 1538 eruption, a new period of subsidence began,
which was interrupted in 1950, when a new series of uplift episodes commenced (Del Gaudio et al.,
2010). Two major uplift episodes occurred between 1969-1972 and 1982-1984, characterized by
significant and rapid uplift (with a cumulative uplift of about 3.5 m) accompanied by intense
seismicity. These events led to the evacuation of 3000 residents from the oldest part of Pozzuoli town
(Rione Terra), in 1970, and the entire town of Pozzuoli comprising 40.000 people, in 1984 (Barberi
et al., 1984). After approximately 20 years of subsidence, a new uplift phase began in 2005-2006,
with a much lower uplift rate (0.01 meters per month on average, compared to about 0.06 meters per
month in the 1970s and 1980s), but long-lasting and still ongoing. This new unrest has been
accompanied by progressively increasing seismicity, which has substantially intensified, both in
frequency and maximum magnitude.  The maximum magnitude reached M=4.4 on May 20, 2024,
once the maximum ground level attained at the end of 1984 was reached (in July 2022) and surpassed.
The progressively increasing seismicity confirms the predictions of Kilburn et al. (2017) and Troise
et al. (2019), who based their forecast on the correspondence of the ground level with stress levels at
depth. This seismic activity represents a significant and continuous hazard for the edifices in such a
densely populated area, given the very shallow depth of the earthquakes (about 2-3 km). Furthermore,
the current crisis poses an even higher threat as it could potentially be a precursor to a future eruption
in the area.
The present study is aimed to reconstruct and interpret the events before and after the 1538 eruption.
This analysis follows three main paths: i) the accurate reconstruction, of the ground movements in
this area since early historical times, using historical testimonies and documentation; ii) the accurate
reconstruction of the uplift movements that evolved from 1430 to 1538, accompanied and followed
by significant seismic events; iii) the analysis of stratigraphic and geophysical parameters, which,
although collected in the recent era, provide important elements for the reconstruction and
interpretation of the unrest related to the 1538 eruption.



Finally, the interpretation of the events preceding, accompanying and following the 1538 eruption is
used to provide insight into possible evolution scenarios for the present unrest, which started in 1950
and is still in progress (Troise et al., 2019; Scarpa et al., 2022)

**2. Caldera formation and post-caldera volcanic activity 14 ka - 3.7 ka**
Campi Flegrei caldera has been generated by a huge eruptive event, the 15 ka Neapolitan
Yellow Tuff (NYT), as demonstrated by recent research based on drilling results (Rolandi
et al., 2020a; 2020b). The caldera collapse resulted in many new fractures, which gradually
became eruptive vents. Through these vents, the eruptions continued, exhibiting the
characteristics of a volcanic field (Druitt and Sparks, 1984), resulting in the so-called post-
caldera activity. Dome-shaped uplift of NYT occurred after the caldera formation in the
central zone of Campi Flegrei, with uplift up to hundreds of meters on the caldera floor (Rolandi
et al., 2020b). The significant uplift involved a large intra-calderic NYT block, making Campi Flegrei
a typical example of resurgent caldera (Rolandi et al., 2020b). The post-caldera activity gave rise
to numerous craters, predominantly tuff cones and tuff rings (Fig. 1a,b), displaying the
typical characters of monogenic volcanoes (Marti et al., 2016). Within Campi Flegrei, 35
small eruptive centers have been identified, since the NYT eruption (Di Vito et al., 1999;
Smith et al., 2012), producing more than 60 eruptions. The magmas associated with these
eruptions are typically trachytes and alkali trachytes, with smaller amounts of latite and
phonolite (Di Girolamo et al., 1984; Rosi and Sbrana, 1987; D'Antonio et al., 1999). The
post-caldera eruptions can be then classified in two periods, occurring between 14 ka
and 8.2 ka BP and 5.8 and 3.7 ka BP., respectively, with an interval of significant
subsidence without eruptions from 8.2 to 5.8 ka BP (Rolandi et al., 2020b).





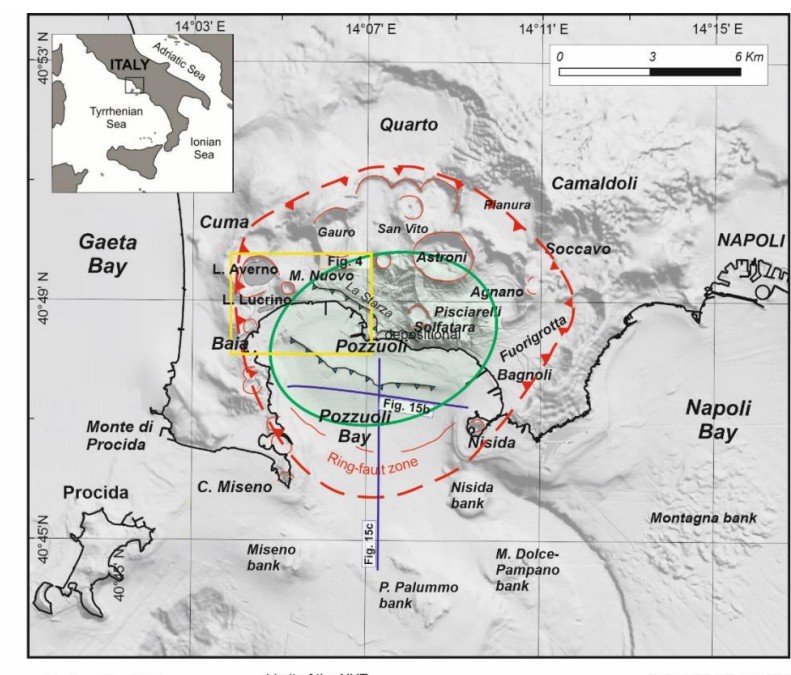


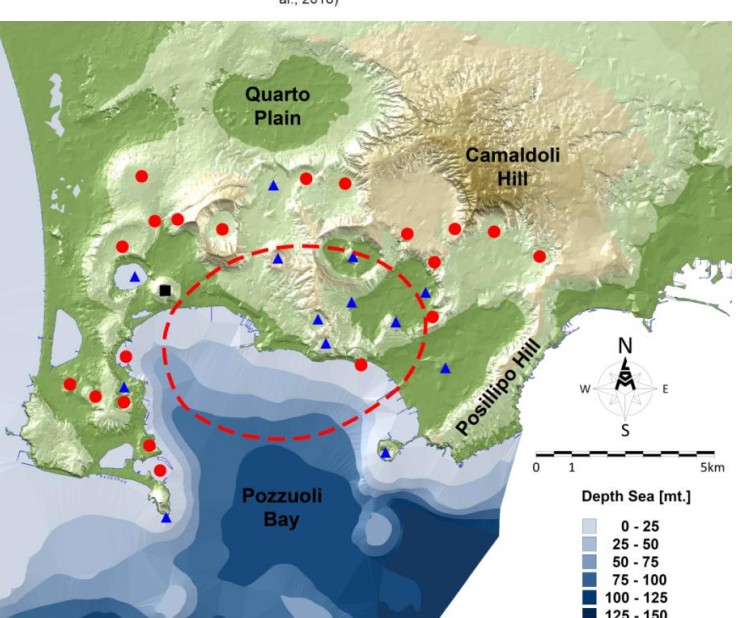


**Fig. 1 – *Top*: Location map of the study area with indication of relevant toponyms and major volcano-tectonic and morpho-structural lineaments associated with the Campi Flegrei caldera. *Bottom*: Map of Campi Flegrei caldera. Red circles indicate the craters of the first post-caldera volcanic phase, blue triangles indicate the craters of the second phase. The red hatched area represents the resurgent block of NYT extended in the Pozzuoli Bay.**




The second post-caldera eruptive phase was preceded by the uplift of 30m, above sea
level, of La Starza marine terrace (Cinque et al., 1983; Rolandi et al., 2020b). The
distribution of eruptive centers reveals that, during the first post-caldera phase, they were
distributed around the resurgent block. In the second phase, among thirteen volcanic edifices,
seven occurred within the resurgent area (Fig. 1).
It seems likely that the second post-caldera phase (5.8 - 3.7 ka) can be considered the primary
reference for defining possible future eruptive scenarios, following the eruption of 1538 AD.

**3. Subsidence and uplift evolution before the 1538 eruption**
As inferred from historical chronicles, as well as from studies on the incrustations and traces of
bioerosion on the Pozzuoli Serapeum marble columns (Parascandola 1947; Bellucci et al. 2006), after
the two post-caldera phases previously defined, large ground uplift and subsidence in the order of
tens of meters, occurred. Historical documents allowed us to precisely reconstruct such ground
movements in Pozzuoli area (central part of the caldera) and in the Averno area (3 km west of
Pozzuoli, close to the area where the 1538 eruption occurred. The reconstruction reported here, based
on all reliable historical documents, is the most complete and rigorous, correcting several
misinterpretations and/or erroneous reconstructions that appeared in previous literature.
The first evidence of subsidence in the Campi Flegrei area dates back Greek times, as reported by
Diodoro Siculo (VIII century BC) and is related to the area in front of the Averno Lake, and the 1538
eruption which generated the Monte Nuovo cone. We will start to describe the historical documents
to shed light on the ground movements in this area, then we will reconstruct ground movements in
the most deformed, central Pozzuoli area.
A fundamental historical marker for inferring the ground movements west of Pozzuoli, is the Via
Herculea, which has been used since the Greek times (beginning in the 8th century BC) and continued
to be very important during the Roman times. Via Herculea, whose detailed history is shown in the
supplementary material for a reconstruction of its movements as reported by several sources during
the past centuries, was the name given to a road running on a thin land strip, likely formed by
aggradation in coastal shallow water settings of volcaniclastic sandy deposits (Parascandola, 1943),
mostly erupted from the 5ka and 3.7 ka eruptions of the Averno and Capo Miseno volcanoes (Sacchi
et al., 2014; Di Vito et al. 2011; Di Girolamo et al., 1984), giving rise to a Lake (Fig. 2a). Since the
elevation of this land, used as a road running along the coast from Pozzuoli to Baia, was only few
meters above the sea level, ground subsidence strongly perturbed its use as a road, and such troubles



were often reported in historical documents. For this reason, it provides compelling evidence for the
evolution of ground subsidence in this area during the centuries.
The Greeks coming from Euboea in the 8th century BC, firstly settled on the island of Ischia
(Pithecusa), then founded the polis of Cuma, which represents the first Greek colony of Magna
Graecia and of the entire western Mediterranean. Thus, since the 8th century BC the thin land stipe
assumed the function of a road taking the name of Via Herculea, to reach the cultivated countryside
around Pozzuoli (Fig. 2b). Diodoro Siculo (see Appendix 1) reported that, already at their times,
continuous subsidence affected this area, thus generating problems to the practicability of Via
Herculea.
In Roman times, since the beginning of the 1st century BC, the body of water enclosed by the Via
Herculea, purchased by Sergio Orata, played an important role in fish-farming since 90 BC, taking
the name of Lucrino, much larger than the present-day Lake Lucrino. After his death, due to
continuous subsidence which menaced both the practicability of the Via Herculea and the fish farming
activities, the new owners around 60 BC, turned to the Roman Senate calling for appropriate
interventions. For this purpose, in 59 BC Julius Caesar was commissioned, which built a barrier (*Opus*
*Pilarum*) and special shutters to protect the road and the Lucrino Lake from sea ingression (see
Appendix 1). Towards the end of the same century, for military purposes, in 37 BC Agrippa cut both
the Via Herculea and the barrier with the crater of Avernus. Having understood, unlike Julius Caesar,
the continuous subsidence of the Via Herculea, which at the end of the century was only few meters
above sea level (Fig. 2c), also ***increased its height*** (Strabo, 1st century BC). About four centuries later
Theodoric (King of the Ostrogoths), upon request for the protection of fish farming, restored the dam
by increasing again the height of via Herculea with respect to the sea level (Parascandola, 1943).
Due to continuous subsidence, the Via Herculea finally sank below the sea level between 6th - 7th
century A.D, when the sea penetrated the crater of Averno, the Lake Lucrino having disappeared (Fig.
2d). Proof of the disappearance of the Via Herculea and of the Lucrino Lake was also testified by
Boccaccio, who lived in the Naples area from 1327 to 1341 AD and described the Averno area in its
geographical book 'De montibus' (…*to Avernus, connected in ancient times with the nearby lake*
*Lucrino where it recalls the waters of portus Iulius*).



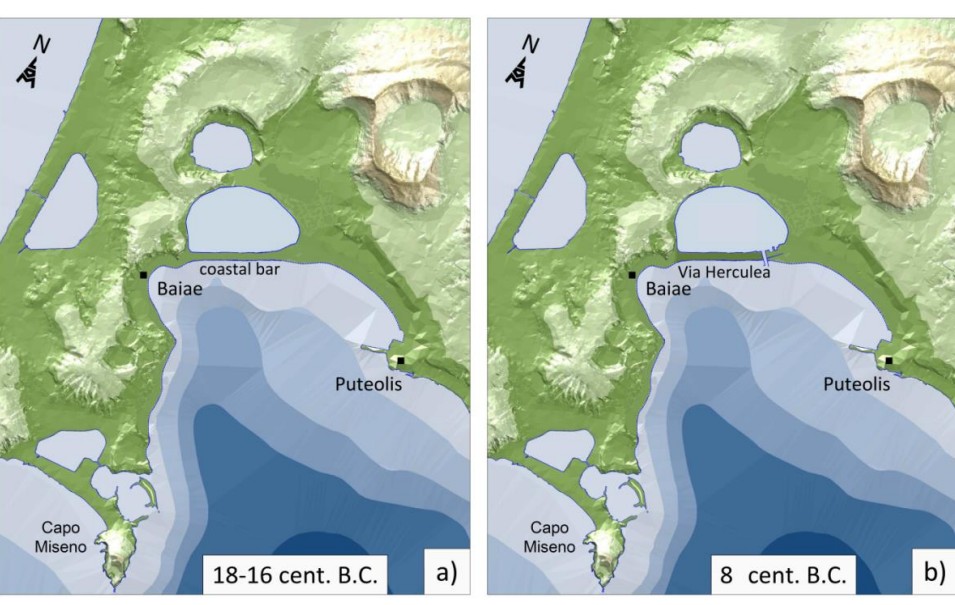

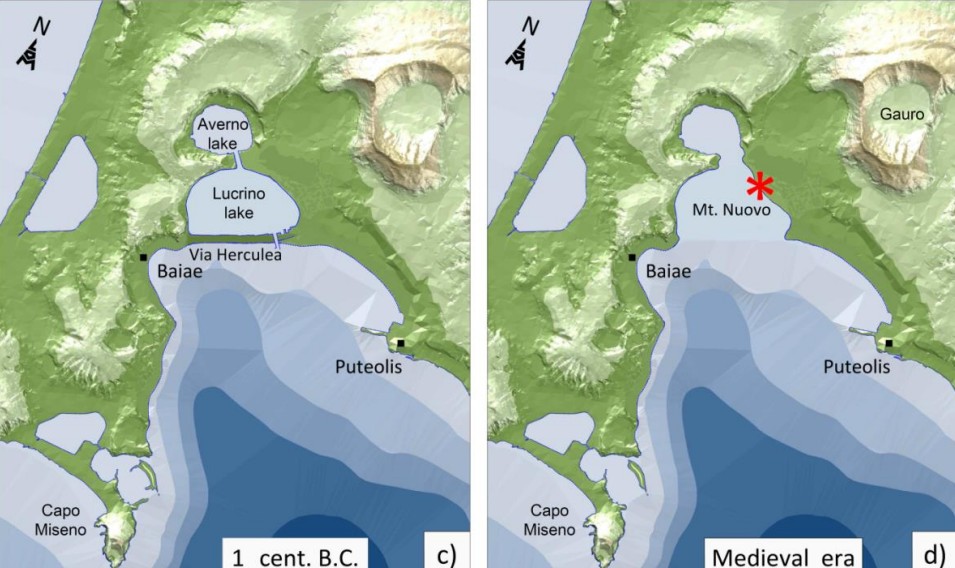

**Fig. 2 - a,b,c,d) position of the via Herculea in relation to the bradiseismic phases along 33 centuries. The red dot indicates the central point around which the volcanic edifice of 1538 was formed.**

Via Herculea never rose above the sea level again, despite the large uplift phase, occurred before and during the 1538 eruption (see Fig. 2d).





The tentative reconstruction of the level of Via Herculea, approximately shown in Fig. 2 as briefly
described above, is shown in detail in Fig. 3, where each point of the curve refers to a specific
documented historical period, starting from the Greek age (8th century BC), through the Roman era
and the late Middle Ages, until the eruptive event of 1538 (see Appendix 1). Note that on the Via
Herculea, at the end of the 1st century BC and at the end of the 4th century AD, works were carried
out to increase its height above sea level due to the incipient submersion. Due to these works, the
submersion of the structure was delayed from ca. the 3rd, 4th century BC, up to the 7th century AD
(Fig. 3). The date of submersion around 6-7th century is also consistent with the observations reported
by Parascandola (1943), indicating that the land strip of Via Herculea still emerged above sea level
for much of the 6th century.
It is fundamental to note is that Via Herculea never reemerged again, not even immediately before
and during the eruptive phase of 1538 (Parascandola, 1943).
The submerged relics of the Via Herculea are still visible  today located at about 4.5 meters bsl, as
shown in the high-resolution bathymetry (Fig.4) recently obtained by Somma et al. (2016).

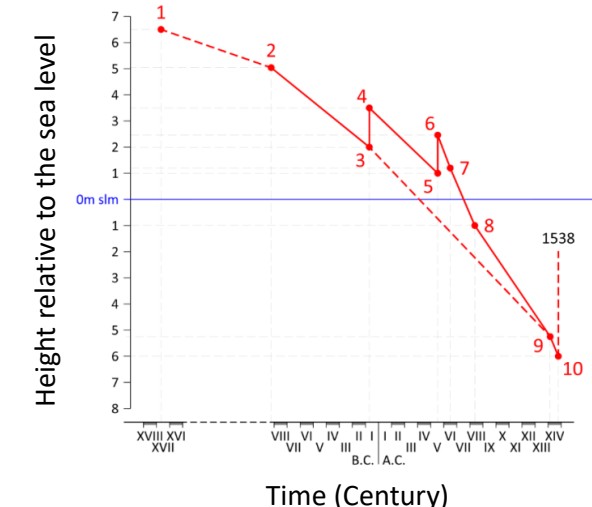

**Fig. 3 – Diagram  showing the trend of  ground movements at the Via Herculea, as referred to sea level, along 33 centuries.**



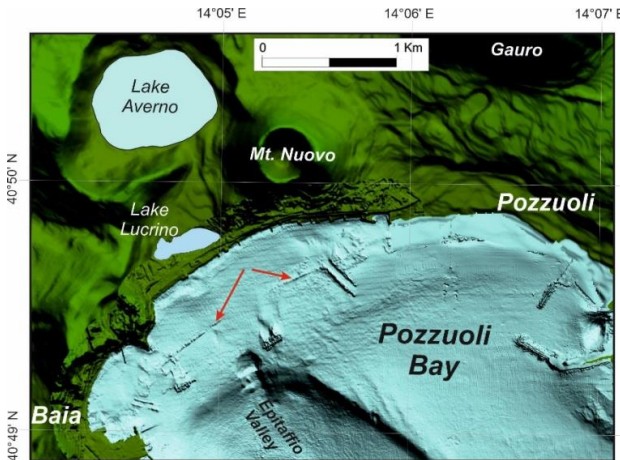


**Fig. 4 – Shaded reflief map of the coastal area of the Pozzuoli Bay based on high resolution**

**multibeam bathymetry (Somma et al., 2016). Arrows indicate the submerged remains of the**

**breakwater pilae of the via Herculea.**


Meanwhile Via Herculea records the most ancient subsidence in the whole area, the best evidence for
subsidence in the Pozzuoli area, where maximum ground movements are recorded, comes from the
historical-archaeological elements linked to the Serapis Temple (Serapeum), although subsidence in
the Pozzuoli area is also testified since Greek times (Gauthier, 1912).
Recently, Amato and Gialanella (2013) discovered, by drilling into Serapeum area, four successively
superimposed floors, ranging from the Augustan age (31 BC-14 AD) to that of the Severi (193-235
AD), thus indicating the progressive subsidence of the manufact (Fig. 5). The most elevated 4th floor,
was built in the Severi Age, indicating at that time the previously built three floors where all below
the sea level, and from this epoch we will follow the historical traces of further subsidence and
subsequent uplift. The resulting time evolution of the approximate level of the 4th floor of the
Serapeum is reported in Fig. 6. Also in this figure, as for the Fig.4, each number refers to a given
historical document supporting that level (see supplementary material, Appendix 2). From historical
information we know that the 4th floor subsided below the sea level in the 5th century, i.e., about 200
years after its construction during the Severi Age. When the 4th floor reached a level of 3.6 m bsl,
around the 7th century AD, the columns were wrapped by layers of sedimentary materials, which
formed the so-called "fill" (Parascandola, 1947). Then, due to the impact of the relative sea-level
change on the coastal area colonies of lithodomes attached the part of column at the mean sea level,
between 3.6 and 6.30 water depth (see the two red arrows in Fig. 7c) and creating a pitted band above
the sedimentary materials, for a thickness of 2.70m. This process occurred until the 9th century AD,
when the fourth floor located to a depth 6.3 m below sea. Such a depth was considered by some



authors (Parascandola 1947, Amato and Gialanella, 2013) to be the maximum submersion reached in
the 9-10th century. In the same period, however, the ground subsidence caused the flooding by
thermal and rain waters, of the Agnano plain, an area located to east of Pozzuoli, and resulted in the
formation of a lake (Annecchino, 1931). This event indicated a general persistence of subsidence in
the Pozzuoli area, which was in fact confirmed very clearly even in the following centuries, as
highlighted by numerous historical documents, resumed here (Fig. 7a) and reported in detail in
Appendix 2.

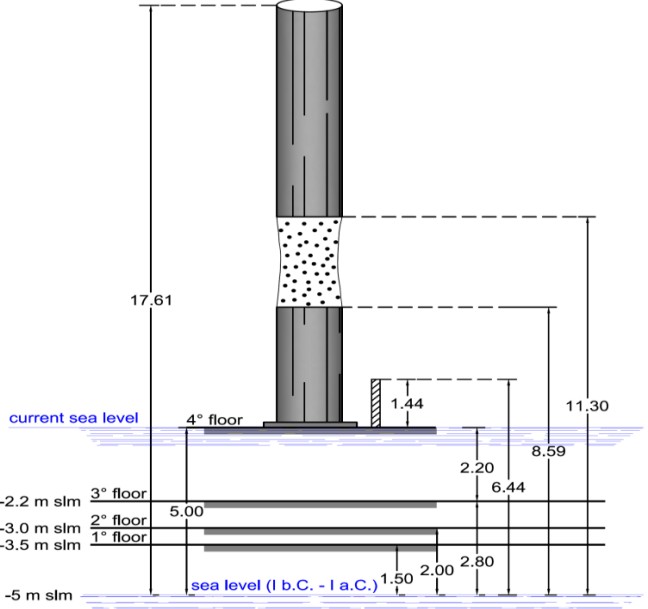


**Fig. 5 – Floors underlying columns of Serapeo (redrawn from Amato and Gialanella, 2013).**
**The dotted part of the column indicates the boring due to colonies of *Lithodomus Litophagus*.**

In the 11th century the Arab geographer Idrisi and other historians of 12[th] century (Benjamin ben
Yonah de Tudela) and 13th century (Nicolò Jamsilla), clearly highlighted the morphology of Rione
Terra as a medieval castle surrounded by the sea on three sides, due to the continuation of the
subsidence, which was still underway at that time (Costa et al., 2022) (see points 6 and 7 in Appendix
2). Moreover, in 15th century there is the account of Boccaccio (1348), as reported by Parascandola
(1943), who wrote that the fisherman's wharf in the Bay of Pozzuoli became completely submerged
(point 8 in Appendix 2).



We can prove again the subsidence continued further in the following century, since it is possible to
get a more precise estimate of the depth below sea level reached by the 4th floor of the Serapeum, by
observing the painting "Bagno del Cantariello" (Fig. 7a), part of the famous Balneis Puteolanis of the
Edinburgh Codex of 1430 AD (Di Bonito & Giamminelli, 1992). The painting depicts the Rione
Terra encircled by vertical yellow tuff walls, from which the beach of Marina Della Postierla extends
(towards the observer) to the base of the S. Francesco hill, the source of the thermal spring Cantariello
(foreground) near the coast northeast of the submerged Serapeum. Behind the visitors of the thermal
spring, the painting clearly shows the upper part of the three marble columns of Serapeum emerging
from the sea. Also depicted are people fishing directly from the shore (Fig. 7b). From this painting
we can make a roughly estimate of the portion of columns below the sea level at that time, taking in
account that significant part of the columns is submerged. Historical records from the 1750
excavations, (see further) indicate that the buried part of the columns amounted to about 10 m (see
Parascandola, 1947); the shallowest 2 meters of the excavations were formed by pyroclastic flow
deposits of the 1538 eruption 8 (see further paragraphs).

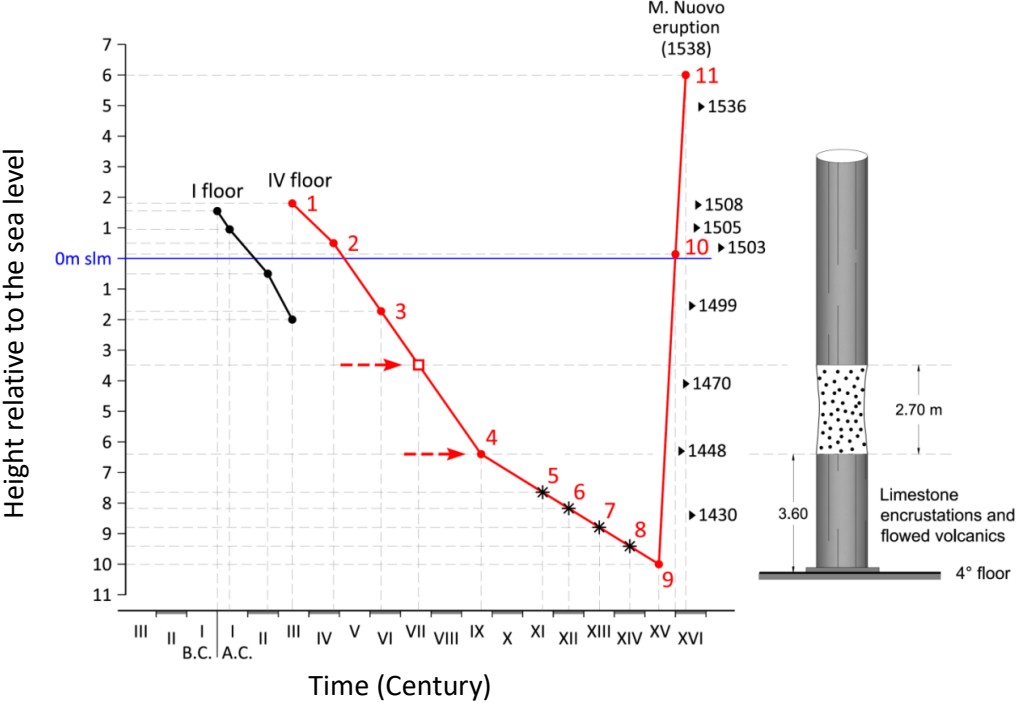


**Fig. 6 – Diagram of ground deformations with reference to the fourth floor of the Serapeo**
**(points 1-4). Points 5-7 indicate the submersion of the Pozzuoli area through the topographic-**
**morphological variations acquired by the Rione Terra due to submersion (see supplementary**
**historical material). Finally, points 8-9 indicate the extent of the submersion referring to the**



**Caligolian pier and to the 4ᵗʰ floor of Serapeum, the latter lasted until 1430. The rapid ascension phase is also shown, associated with earthquakes of greater energy that accompanied the emergency of the 4th floor from the sea in the early 1500s, until the eruption of 1538.**

This observation constitutes an indication that during the time of the painting (1430), in the absence of 1538 products, the buried part of the columns should then have been approximately 8 meters. Moreover, the presence of trawling fishermen in the scene (Fig. 7b) suggests that sea depth there did not exceed 2 m (the maximum water depth for this type of fishing not far from the beach). Given that the total height of the columns is 12.7 m, we estimate that the emerged part of the column in 1430 was around 2.0-3.0 m (Fig. 7a,c).

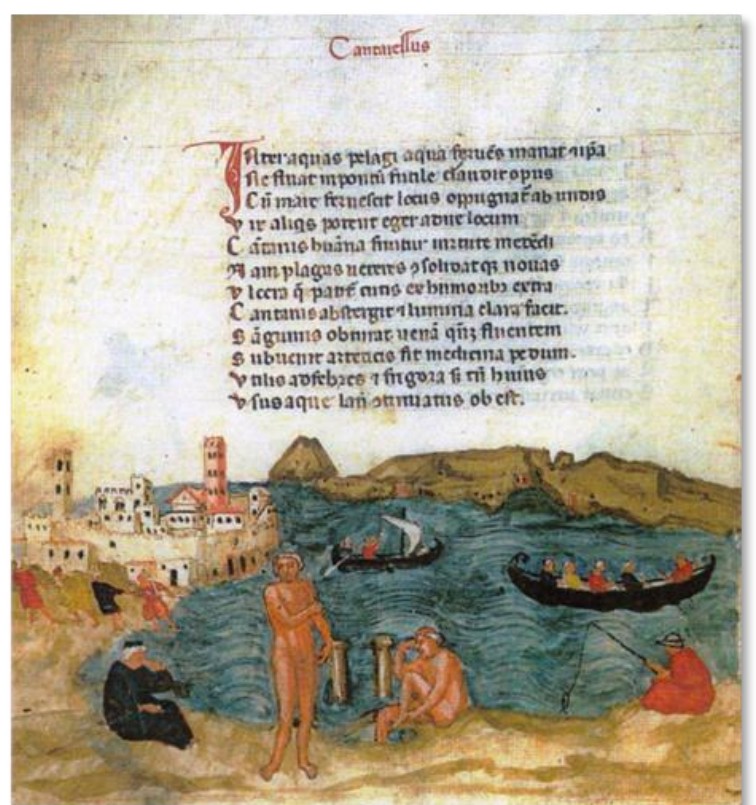

a)



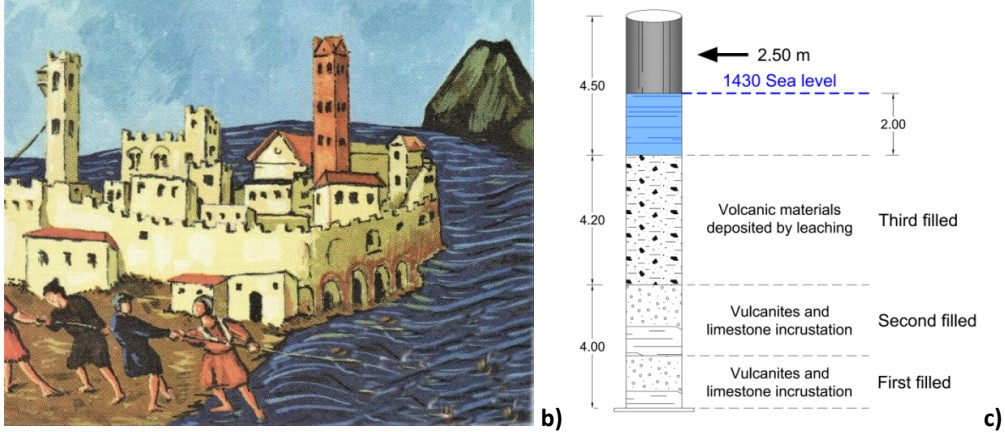



**Fig. 7 – Gouache of de' Balneis Puteolanum from 1430: a) Stumps of the Serapeum columns**
**that protrude from the sea to a height of 2-3m, b) Fishing from the shore, highlighted in the**
**box, indicates a draft depth of approximately 2m of sea, c) Since the columns are 12.70m high,**
**it can be deduced that the remaining part of the columns wrapped in the underlying sediments**
**is approximately 8m. From the figure it can therefore be deduced that the 4th floor of the**
**Serapeum in 1430 was 10m below sea level**

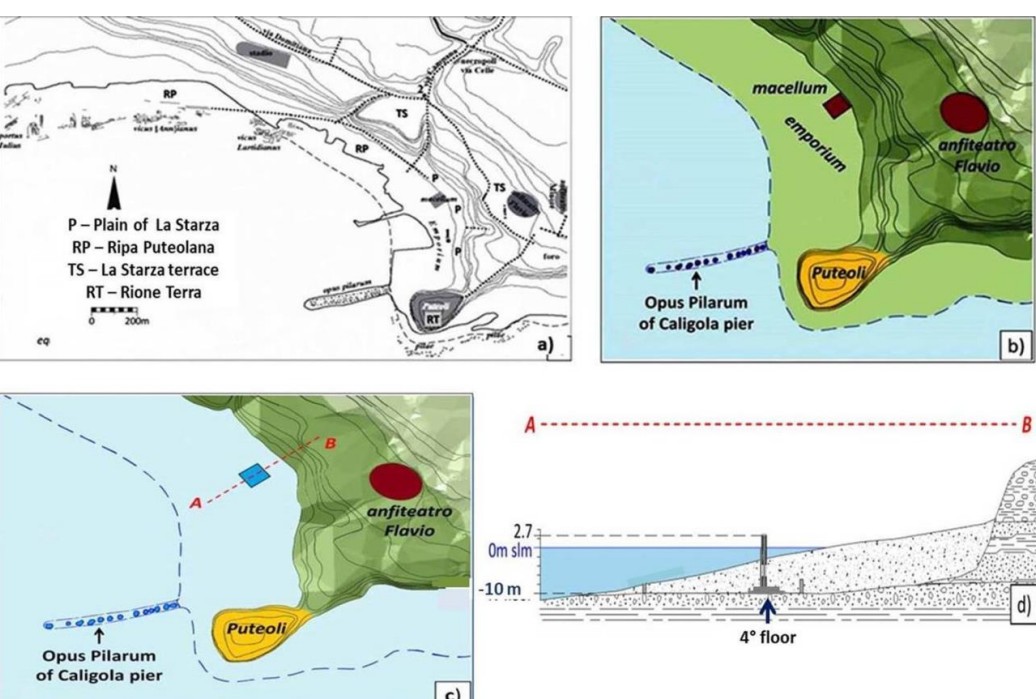






Consequently, we infer that in 1430 AD the floor was about 10 m (+/-1 m) below sea level (Fig. 6).
Such deduction derived from the context represented in Fig.7a, can be explained in even greater detail
with the help of the topographic map of the Pozzuoli area in Roman times (Soricelli 2007)
(Fig. 8a).

**Fig. 8 – a) Map from the Roman era (Soricelli 2007), with our own reworking, based on the**
**indications of Aucelli et al. (2020) and Taravera (2021). The map shows the lower part of the**
**emporium which extends along the Puteolana bank (RP), until reaching the base of the hill, the**
**so-called Starza plain (P) and the upper part of the Rione Terra cliff (RT) which, in turn,**
**connects with the upper hilly part of the Starza terraced area (TS), b) Part of the previous map**
**limited to the Emporium Area, c) the area b subject to the subsidence phase which ended in**
**1430, during which the hill areas (TS, RT) were surrounded at the base by the sea, according**
**to a description of the lower area of Pozzuoli from 1441 "***the sea covered the littoral plain, today***
***called Starza"*** **(De Jorio, 1820; Dvorak and Mastrolorenzo, 1991), d) note that in the profile A-**
**B the sea extended behind the Serapeum on the plain of La Starza hill, intersecting the columns**
**at a height of 10m (also shown).**

The map (contour lines of 5m), shows that in the period of greatest development, the city included
the Greek Acropolis (the ancient Dicearchia nowadays called Rione Terra), with a maximum height
of 40 m asl, the lower part of the city, i.e. the western area overlooking the ancient emporium and the
Serapeum (Roman macellum) placed near the bay area and the upper city, on the Starza terrace, with
elevation between 30-50 m asl. The latter was the site of the ancient monumental edifices
(amphitheatre, stadium, forum, necropolis, etc.). From this map, considering only the area of the
Emporium (lower part) and amphitheater (upper part), a sketch of topographical relief above the sea
level (in Roman times, Fig. 8b) and underlying sea level (in 1430 AD, Fig. 8c) has been obtained  and
described as follows:
- from profile A-B of Fig. 8c, as reported in Fig. 8d, it can be seen that the 4th floor of the Serapeunm
is located at a depth of 10m, packed in the sediments that form the Ripa Puteolana (RP), with the
columns protruding from the same sediments for 4.5m, of which approximately 2m are sea water. It
is indicated, ultimately, that the sea level intersects the columns of the Serapeum at a height of
approximately 10 m, connecting with the contour line of 10 m, on the La Starza Plain (P) (Fig. 8c,d).
- Fig. 8c also allows us to highlight the morphological conditions of the Rione Terra, which, as we
have already observed, has been described by the chroniclers who visited this place from the 11th to





the 13th century as "*an unapproachable mountain completely surrounded by the sea*" (see Jamsilla
and Fuiano, 1951 and Varriale, 2004, in supplementary historical material).
The historical data presented here are not in agreement with some results that appeared in a recent
work (Di Vito et al 2016), based on the following considerations:
1) the subsidence in the area started in 35 BC;
2) the local uplift in the area of the 1538 vent, from 1536 to 1538, amounted to about 19 m.;
3) the maximum subsidence was reached in 1251.
The first claim is in contrast with at least two strong evidences, coming from historical documents:
the first one, that already at the times of Greek colonization (end of 8th century BC) the Via Herculea
used by Greeks, showed signs of subsidence (see Diodoro Siculo in Appendix 1) (Fig. 2). Limiting
ourselves to the 1st century BC, it is sufficient to observe that since 60 BC, due to the subsidence of
this dam, Giulio Cesare himself was sent by the Roman Senate in 58 BC, to fix the problem, which
was resolved more constructively by Agrippa in 37 BC, raising the surface of the Via Herculea with
respect to the sea level (see again detailed explanation in Appendix 1).
Claim 2) can be easily demonstrated to be not realistic, because in case of uplift in the Monte Nuovo
area higher than few meters, the Via Herculea would have risen back above the sea level (Fig.3d).
Claim 3), finally, is not confirmed by the testimonies collected until 1430, which instead indicate the
continuation of this phenomenon (Di Bonito and Giamminelli, 1992; Bellucci et al., 2006).

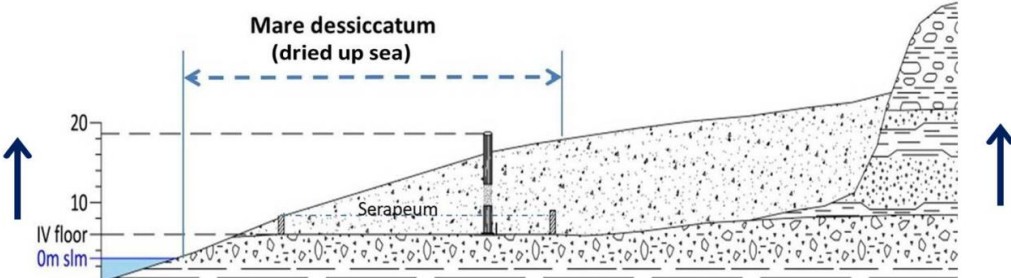


**Fig. 9 – The uprise of the land (marked by the two arrows on the sides) was observed and**
**described by Loffredo Ferrante in 1530: *"the sea was very close to the plain which was at the foot***
***of the Starza hill"*. In this context, the 4th floor of the Serapeum had reached a height of**
**approximately 4 m above sea level.**

From our reconstruction, based on reliable historical documentation, we demonstrate that the
hypothesis that maximum submergence depth of the 4th floor of the Serapeum was reached in the 9-
10th century, proposed by Parascandola (1947) and Amato and Gialanella (2013), is not realistic. Nor



it is the hypothesis by Di Vito et al. (2016), who place the date of the transition between subsidence
and uplift in the 13th century and precisely in 1251.
Let us remember that, as observed in recent unrests, uplift at Campi Flegrei area, which will be
described well later, is accompanied by seismicity (Dvorak and Gasparini, 1991; Kilburn et al., 2017;
Troise et al., 2019). For many centuries, after the 9th century, and for two centuries, after the 13th
one, there is absence of historical evidence for significant seismicity. In the period since 1430 to 1580,
on the contrary, there is abundance of chronicles describing significant seismicity, how will be
detailed later in this work (see Fig. 19a). Our findings dating the starting phase of uplift around 1430
is also supported by the documented occurrence of a powerful earthquake in 1448 (Colletta, 1988:
see also next paragraph), which induced King Ferdinand I of Aragon to suspend the so-called
"fuocatico" (a mediaeval tax collected for each fire lit by a family unit). It is also well known that,
between 1503 and 1511, the municipality of Pozzuoli granted the lands that emerged, as a result of
the increasingly "drying up sea" (Fig. 9), expanding the available land, to citizens requesting them
(Parascandola, 1947). The next important question is then: was the 4th floor of the Serapeum above
sea level as early as at the beginning of 16th century? Parascandola (1947) answered this question
through a sentence found in an account by Loffredo Ferrante from 1580*: In 1530 the sea was very
close to the plain which was at the foot of the Starza hill* (Fig. 8). So, it can be deduced that the floor
of the Serapeum in the 1503 was just above sea level, that is, it had risen about 10m in about 73 years,
with a minimum rate of 160 mm/y. There is clear evidence that the uplift phase continued until 1538,
when the eruption occurred, whereas seismicity continued for the next 40 years, until 1580 (we
postpone the discussion of this topic to the next section). The maximum uplift occurred in the
Pozzuoli area, close to the Rione Terra cliff, that up to the 1538 eruption reached an elevation in the
order of 5-6 m asl (Fig. 6).
In the nearby area facing Averno to the west, the uplift, as already said, was unable to cause emersion
of the Via Herculea, and only a small area including the vent was affected by an uplift of about 7m,
i.e. slightly higher than the uplift at Pozzuoli. In the eastern sector of the caldera, at Nisida island, the
pier did not emerge above sea level (Parascandola 1947). It is then very likely that the uplift phase
had a bell-shaped trend, very similar to what we see in the recent unrests, with the sole anomaly of
the sharp pre-eruptive uplift of Monte Nuovo, likely due to the upward migration of the dyke feeding
the eruption.

4. **Ground movements after the 1538 eruption**
The period between the end of the 16th century and the beginning of the 17th century lacks any written
historical document testifying the ground movements at Pozzuoli. It is likely that after the 1538



eruption a subsidence phase started. We can anyway learn something from some paintings, the oldest
one by Cartaro, dated 1584 (Fig. 10a), which highlights the Rione Terra in the foreground, with the
Neronian pier which emerges almost completely above sea level, which means for about 5-6 m.

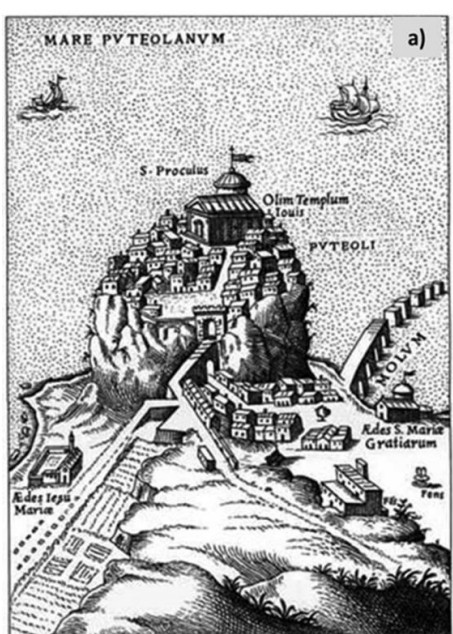
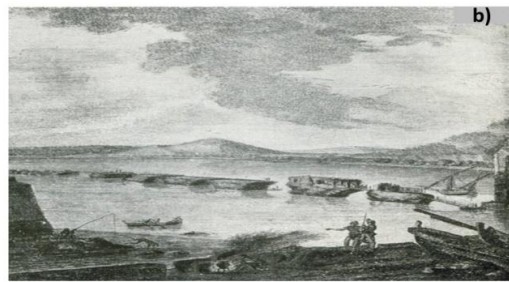
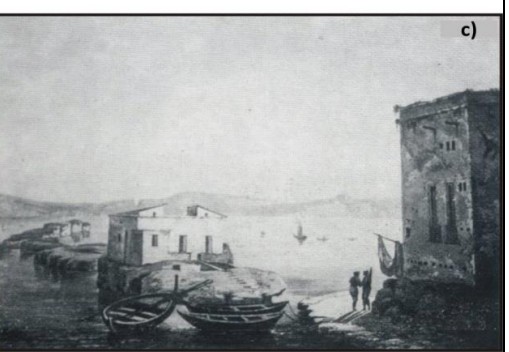


**Fig. 10 – a) Engraving by Cartaro (1584) showing the Neronian pier at the base of the Rione**
**Terra, emerging from the sea for 5-6m, showing 10 of the 15 piles of which it was made up in**
**roman epoch, b) The remains of the pier piles, without the upper arches, highlighted in an**
**engraving from the mid-18th century, c) Detail of the same piles highlighted in another**
**engraving from the same period, where the height of the 1-2m piles is observed in more detail,**
**subject to marked erosion**

It also appears still partially complete, with about half pylons still connected with arches (*Opus*
*Pilarum*). In comparison, paintings from the middle XVIII century (Fig. 10b,c) report the pier
completely destroyed, and clearly almost completely submerged; the painting of Fig. 10c represents
the pylons in more detail, allowing to estimate the height of the emerging part asl around 1-2 m. Fig.
11 shows another famous painting of 1776, by Hamilton, which shows the ruins of the Neronian pier
almost the same way than in Fig. 10b,c and, in addition, shows the columns of Serapis Temple, with
its floor almost at the same level than the Neronian pier.

 

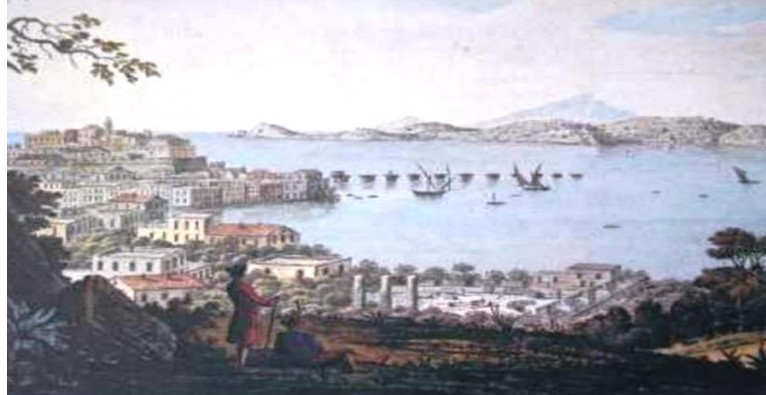


**Fig. 11 – a) View of the Gulf of Pozzuoli and the Cape Miseno peninsula (Hamilton 1776).**

**Both the remains of the Neronian pier and the newly excavated Serapeo are also visible**


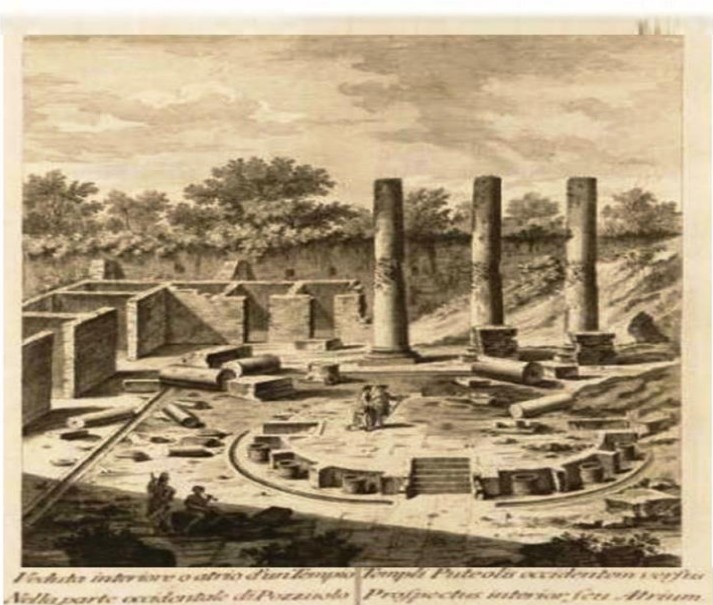


**Fig. 12 – Serapeo excavated in the three-year period 1750-1753. It can be noted that the height**
**of the lighter parts of the columns, including the pitted band of the lithodomes, is preserved by**
**oxidation, because packed by the newly removed sediments. The darker upper part, oxidized**
**since staying outside the cover, has a height of approximately 2.50m, estimated on the same**
**figure. This leads us to consider that the band of sediments removed had a thickness of**
**approximately 10m, that is, the height of the hill where the *vineyard of the three columns* was**
**located before the excavation (Niccolini, 1842).**




From the comparison between Fig. 10a and 10b, c it can be deduced **t**hat the Roman opus pilarum
underwent a subsidence of about 4-5 m. from 1580 to 1750.
Since the floor of the Serapis Temple appears to be at the same level than the pier, its level in 1538
can be estimated as 5 – 6 m. above sea level (Fig. 6), while in 1750 it should be at about 1m above sea
level, with an estimated subsidence 1580-1750 of about 4-5 m. This approximate estimation is however
confirmed by Parascandola (1947), who reports some measurements by Niccolini (1846), who found
the 4th floor of Serapeo to have a height above sea level varying in the range 0.9 - 0.6m throughout
the 18th century. It can then be deduced that during the three years of the excavations (Fig. 12) the
floor could have been approximately at 0.7 m above sea level.
Finally, we want to highlight, in agreement with Parascandola (1947), that the subsidence of 4 - 5 m,
started after 1538-1580, could have evolved at higher initial rate, in such a way that, around the middle
of the 17th century, it already had a value of 2 -3 m, and then slowed down towards the end of the
century, until the 1750.
We are hence able to describe in more detail the whole evolution of ground movements at the Pozzuoli
area since Roman times, including the period following the 1538 eruption and until today. Such a
reconstruction is shown in Fig. 13c. In particular, regarding the post-1538 subsidence phase, the data
shown, starting from the 17th century, have been combined with those obtained by the most
significant measurements carried out by numerous researchers who dealt with this phenomenon
during the 1800s, as reported by Parascandola (1947), who suggested the reconstruction shown in
Fig. 13a. High precision, frequent measurements started to be collected since 1905, initially based on
leveling survey carried out by the Military Geographic Institute (IGM). Data from the levelling
surveys were still provided also during the occurrence of the most recent unrest phases, i.e. in 1950 -
52, 1969 – 72, 1982 – 84 and until 2001. Since 2001, continuous measurements are provided by GPS
(RITE, see Fig. 13b,c) installed at Rione Terra (Del Gaudio et al 2010).



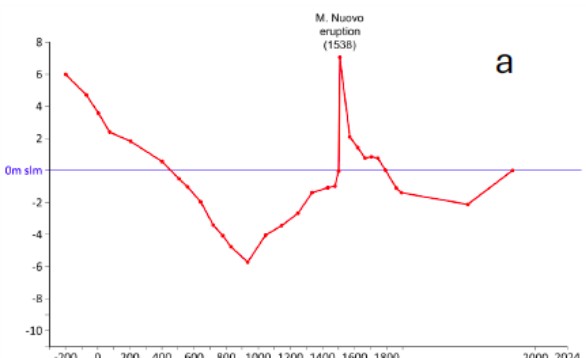

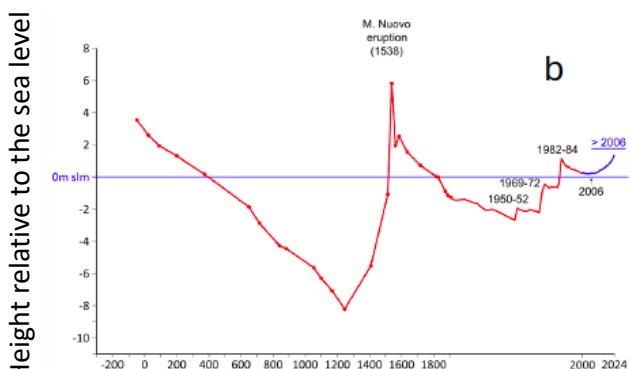

Height relative to the sea level

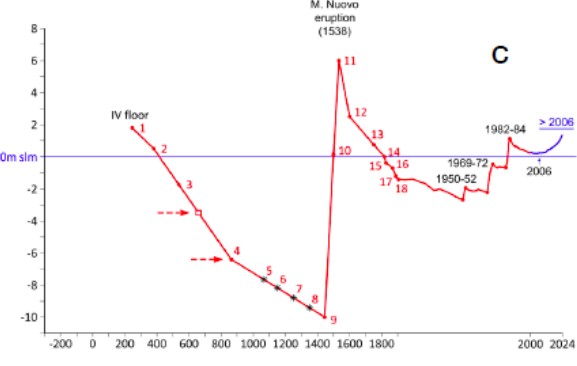

Time (Years AC)

**Fig. 13 a) Reconstruction of the ground level of the Serapeum floor, with respect to the mean sea level (blue line), as proposed by Parascandola (1947); b) The reconstruction of the Serapeum floor ground level, since the III century A.D. to present, recently proposed by Di Vito et al. (2016); since the III century A.C. to today; c) The reconstruction of the ground level of the Serapeum IV floor, since III century A.D. to present, inferred by this study. Each point in the**



**diagram corresponds to an appropriate historical indication reported in the text and/or in the appendix.**

## 5. Schematic model for the preparatory phases of the 1538 eruption

### 5.1 Dynamics of the resurgent block in response to temperature and pressure perturbations

The ground deformation at Campi Flegrei, during the phases preceding and following the 1538 eruption, has been likely very concentrated in a small area of few km of radius around Pozzuoli, just as during the recent unrests (De Natale et al., 2001; 2006; 2019). Such a concentration is in agreement with the presence of a resurgent block.

Evidence for the involvement in the Campi Flegrei unrest episodes of a resurgent block comes from the first observations and modeling by De Natale and Pingue (1993). These authors pointed out that the concentration of the uplift in a small area, the high uplift values, and the invariance of the uplift and subsidence shape, as well as of the maximum seismic area, indicated the up and down movement of a resurgent block, bordered by ring faults focusing the occurrence of earthquakes (see also De Natale et al., 1997; Beauducel et al., 2004; Troise et al., 2003; Folch and Gottsmann, 2006). In recent times, new evidence has been collected about the location and limits of the resurgent block (Rolandi et al. 2020b). Active high-resolution reflection seismic surveys have pointed out and imaged the presence, in the Gulf of Pozzuoli, of an inner resurgent antiformal structure or "block" bounded by a 1-2 km wide inward-dipping ring fault system associated with the caldera border, whose limits have been also documented by the survey (Sacchi et al., 2014 Steinmann et al, 2016; Sacchi et al., 2020a). Further constraints for the extent on-land of the resurgent block come from stratigraphic evidence. In particular, the old well CF-23, drilled in the Agnano area, presents about 900 m of NYT pyroclastic deposits, topped by only 100 m of more recent deposits (Rolandi et al. 2020b). The presence of uplifted, thick layers of NYT, characterizes the stratigraphy of all the wells contained in the resurgent block (Fig.14a,b,e), thus allowing to map its extent on-land, although only the CF-23, by far the deepest one, clarifies the whole thickness of the NYT deposits in the resurgent area (Fig. 14a,c,d). The extent of the resurgent block on-land appears also reasonably well defined by a clear relative gravimetric maximum (Capuano et al., 2013). It is crucial to emphasize that the differential movement of the resurgent block, mostly detached from the external caldera rocks, is responsible for the almost constant, highly concentrated shape of ground displacement, during both uplift and subsidence. The resurgent structure is also associated with distinct seismicity along the bordering ring fault zone (see





also Troise et al., 2003). Fig. 15a-c shows how the resurgent block is well evidenced by passive
seismic data (Fig. 15b, c) and by earthquake locations (Fig. 15a).
The presence of the central, resurgent block significantly influences the dynamical behaviour in
response to temperature and pressure perturbations. This is particularly evident in the central, most
uplifted and seismic area, where the shallow crust comprises approximately 1.5 km of tuff. This
contradicts substructure models proposed by various authors (Rosi and Sbrana, 1987; Vanorio et al.,
2002; Lima et al., 2021; Kilburn et al., 2023), which often assume a thick shallow layer of loose
pyroclastics from recent eruptions, typically represented by the stratigraphy of well SV1 (see Fig.
14e).
The physical state of the shallow structure within the resurgent block can be inferred by seismic
tomography analyses presented by several authors (e.g. Aster and Mayer, 1998; Vanorio et al., 2005;
Vinciguerra et al., 2006; Battaglia et al., 2008; Calò and Tramelli, 2018). These analyses consistently
indicate a high Vp/Vs ratio centered below Pozzuoli town down to 1-2 km, interpreted as highly water
saturated tuff.



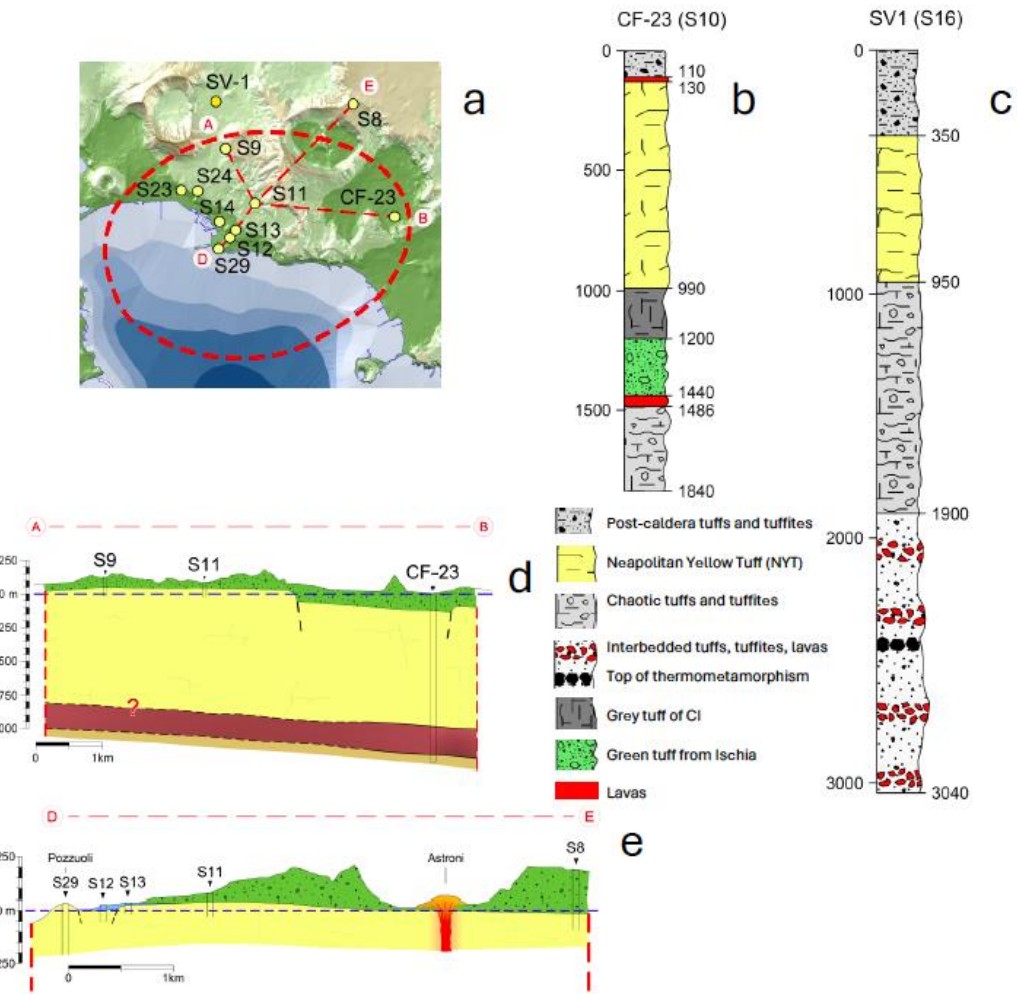

**Fig. 14 - a) Location of the wells explored within the resurgent tuff block, as reported in literature; b) Stratigraphy of the CF23 (S10) well, within the resurgent block; c) Stratigraphy of the SV-1 well, outside the resurgent block, which highlights a stratigraphy where the NYT tuff blocks are not present with significant thicknesses; d-e) Profiles in the resurgent block which highlight the shallow depth of NYT because of the resurgence.**

Of particular significance is the work by Vinciguerra et al. (2006) which compared the results of seismic tomography with laboratory tests. They demonstrated that the tuffs present in the central area of the Campi Flegrei caldera can be either water or gas saturated, and that inelastic pore collapse and cracking produced by mechanical and thermal stress can significantly alter the velocity properties of Campi Flegrei tuffs at depth. The effect on velocities becomes significant when the temperature rises





sufficiently to induce physical changes, such as volume change and the generation of free water
associated with the dehydration of zeolite phases. This can lead to thermal crack damage, further
influencing the dynamic behavior of the area. At higher depths, well CF-23 indicates the presence of
pyroclastic deposits from a depth of approximately 1.5 km to at least 1.8 km, where a temperature of
300°C was measured (Fig. 14b). Likely, at even greater depths of about 3km, marine silt and clay
layers induce silica mineralization and the formation of low-permeability horizons. Due to the high
temperatures, estimated to be at least 400°C, these layers undergo thermal alteration, forming a
thermo-metamorphosed layer (Fournier, 1999; Lima et al., 2021; Cannatelli et al., 2020).
Is important to note that Battaglia et al. (2008) interpreted a low Vp/Vs body, extending to about 3–
4 km of depth, as due to the presence of fractured overpressured gas-bearing formations, confirming
the data of Vanorio et al. (2005). This depth range of 3-4 km likely represents a primary accumulation
zone


**Fig. 15 – a) Campi Flegrei map showing the limits of the resurgent block, which concentrates ground deformation and seismicity. The thinner black line indicates the ring fault marking the limit of the resurgent block at sea, and the thicker one the ring fault associated to the offshore**



**caldera border. b) The N-S and c) W-E profiles of the high-resolution seismic survey, showing the offshore signature of the NYT ring fault system and resurgent structure (from Sacchi et al., 2014, 2020a, 2020b; Steinmann et al., 2016).**

for shallow intruded magma, which is unable to reach the surface and instead forms magma sills (Woo and Kilburn, 2010; Di Vito et al., 2016; Troise et al., 2019; Kilburn et al., 2023). The magma at this depth is likely to be a mush state, solidified, but still at temperature high enough to be remobilized by the inflow of new magma or hot magmatic fluids (De Natale et al., 2004).

At even greater depths, approximately between 7 - 8 km, the main magma chamber is located. This chamber contains both liquid magma and residual mush from past eruptions (Judenherc and Zollo, 2004).

### 5.2 The preparatory phases of the 1538 eruption

A tentative model can be now constructed for the preparatory phases of the 1538 eruption, which accounts for all available data. It is shown in Fig. 16, and can be summarized as follows:

the Pozzuoli area experienced a long period of subsidence, beginning at the end of the second phase of post-caldera volcanism (3.7 ka B.P.) and lasting until 1430 AD. This subsidence was likely triggered by the collapse of the upper and middle crustal blocks into the underlying magma chamber, situated deep within the limestone basement at depths of 7-8 km (Judenherc and Zollo, 2004). The viscoelastic behaviour of the shell encasing the magma chamber may have also contributed to the subsidence, along with the decrease in magma volume due to cooling and crystallization (Fig. 16a).

Since the end of the second phase of post-caldera volcanism, approximately 3.7 ky ago, the primary magma chamber, located at 7-8 km of depth, likely contains a mixture of liquid magma and mush. It's important to note that mush refers to a non-eruptible phase of trachytic magma, composed of 25%–55% volume by crystals (Marsh, 1996; Bachmann and Huber, 2016; Cashman et al., 2017; Edmonds et al., 2019). When heated by several tens of degrees, typically through the injection of hotter magma, mush can revert to a liquid state, thereby regaining the ability to trigger a volcanic eruption (e.g. De Natale et al., 2004; Caricchi et al., 2014). However, the way the mush is rejuvenated by intrusion plays a fundamental role in this mechanism (Parmigiani et al., 2014). One plausible scenario is that the new magma from the deeper crustal levels forms sills at the base of the mush, revitalizing it through the supply of heat, but not of magmatic mass, i.e. only exsolution occurs (Bachmann and Bergantz, 2006; Bergantz, 1989; Burgisser and Bergantz, 2011; Huber et al., 2011; Bachmann and Huber, 2016; Cashman et al., 2017; Carrara et al., 2020). To explain the rapid uplift observed in the interval between 1430 and 1538, the temperature contrast between the two layers could play a fundamental role: the



mafic melt positioned at the base, being hotter than the overlaying layer, undergoes cooling and
crystallization, leading to an increase in the volatile content (primarily H2O and CO2) of the residual
melt (Fig. 16b). Lower ductile rocks tend to deform gradually, allowing magmatic gases to permeate
into the brittle zone above, thereby inducing a thermo-metamorphic separation layer.
The presence of supercritical fluids, within this zone is indicated by a seismic anomaly displaying
low Vp/Vs at approximately 4 km depth (Battaglia et al., 2008). Above this depth the earthquakes are
concentrated, suggesting the occurrence of fractured formations rich in overpressured gas. This
condition likely results in triggering additional earthquakes (Fig. 16a). A similar condition has been
often hypothesized to occur in the Yellostone volcano (Shelly and Hurwitz, 2022). Intense degassing
from the main magma chamber would lead to increased pressure in the shallow aquifers. Moreover,
the rise in temperature would cause the water contained in the tuffs' zeolites to convert into steam,
generating additional overpressure. Such a situation is shown by the CF-23 well, where its
stratigraphy indicates the presence of a lava layer approximately 30 m thick beneath the overlying
tuff blocks, which are approximately 1.5 km thick (Fig. 14b).
It is noteworthy, when considering the correct stratigraphy of the resurgent block, as represented by
the CF-23 well, that some previous models suggesting the presence of two low-permeability layers
at depth (Vanorio and Kanitpanyacharoen, 2015; Kilburn et al., 2023), inferred from the SV1 well
(which is situated outside of the resurgent block) (Fig. 14a), appear to be incorrect. Therefore, above
the thermo-metamorphic zone, magmatic gases do not accumulate below the hypothetical second
low-permeability layer, as postulated by Kilburn et al. (2023), but rather between the summit lava
base and the underlying thermo-metamorphic horizon, corresponding to a depth of 2.5 km, which is
the fragile layer. Consequently, at the base of the lava body, conditions of high temperature and
pressure result in widespread brittle deformation of this layer due to uplift, rendering it highly
permeable by fracturing (Fig. 16b).
Finally, super-compressed magmatic gases were likely contained within approximately a 2.5 km thick
fragile zone, while a limited release of the increased pressure occurred directly through the fractures
connecting the intermediate depth area with the Solfatara and Pisciarelli areas, resulting in the escape
of CO2-rich vapor, as evidenced by the reported increase in fumarolic activity (Chiodini et al. 2021).
Following this hypothesis, it is noteworthy that, at a depth of 1.8 km, the CF23 drill-hole indicates a
very high temperature of 300°C, not far from the supercritical temperature. It is plausible that, if the
temperature significantly increases, due to the supply of deeper, hot magmatic fluids, the water
contained in the basal part of the tuff block could reach supercritical conditions, leading to thermal
fracturing within the tuff block (Vinciguerra et al., 2006), over a certain thickness (Fig. 16b).





As previously mentioned, the increase of pressure resulting from such intense heating caused by deeper
magmatic fluids should be attributed to both the overpressure of shallow aquifers and the vaporization,
of water contained in the zeolites, likely in the form of superheated steam.
In conclusion, the uplift in the Pozzuoli area cannot be attributed solely to magmatic fluids
pressurizing the layer above the thermo-metamorphic horizon (Nespoli et al., 2023; Kilburn et al.,
2023), nor solely to the increase of pore pressure in the shallow aquifers (Casertano et al., 1975; De
Natale et al., 1991; Scafetta and Mazzarella, 2021). Our hypothesis includes the presence of various
sources originating at different depths, collectively contributing to the uplift dynamics underlying the
area of maximum uplift during the period from 1430 to 1503 and coinciding with the phase we
designate as the 'long-term seismic precursors' (see next section). These sources include
approximately 2.5 km of brittle crust enriched in gases under supercritical conditions, along with
overheated conditions in the upper part. Together, these factors act as the driving force behind the
uplift observed in the region during this specific period.
The pressure increases in the main magma chamber, resulting from the input of new magma and/or
magmatic fluids, can trigger the formation of magma sills (Troise et al., 2019). The progressive
intrusion of several magma sills likely leads to the ascent of magma towards the surface. This process
may be further facilitated by phreatic explosions caused by the heating of shallow aquifers, resulting
in depressurization pulses. Intruding magma may encounter layers that are more resistant to
penetration at certain depths. In this case further magma intrusion may be inhibited and lateral
expansion, to form sills, may occur (Gretener, 1969). Previous studies of recent unrests have indicated
that depths between 2.5 and 4 km, close to the upper limit of the ductile zone, are locations where
magma intrusions can halt (Woo and Kilburn, 2010; Troise et al., 2019). Before the 1538 eruption, a
small plumbing system, in the form of flattened intrusions near the contact between a lower ductile
zone and an upper brittle zone in a high-pressure environment, was hypothesized (Fig. 16b) (Pasquarè
et al., 1988). From such a shallower magma chamber, magma can further progress upward towards
the surface. A dynamic in which early intrusions in the shallow crust create small plumbing systems
(i.e. stalled intrusions), from which a dyke later propagates, bringing a small quantity of magma to
the surface, is typical of monogenic volcanoes (Marti et al., 2016). The ability of intruded magma
sills to erupt at surface is also influenced by the relatively short timescale of sill solidification,
typically in the order of ten to twenty years (Troise et al., 2019).
Shallow solidified magma sills, in the form of mush, can be remobilized due to the arrival of new
magma and/or the introduction of hot deeper magma fluids. The significant uplift preceding the 1538
eruption, amounting to more than 16 meters in the initial phase involving the entire resurgent block,




could be interpreted solely in terms of magma intrusion suggesting a total intruded volume, in the

shallow plumbing system, on the order of a cubic kilometer of magma, at least (Bellucci et al., 2006).

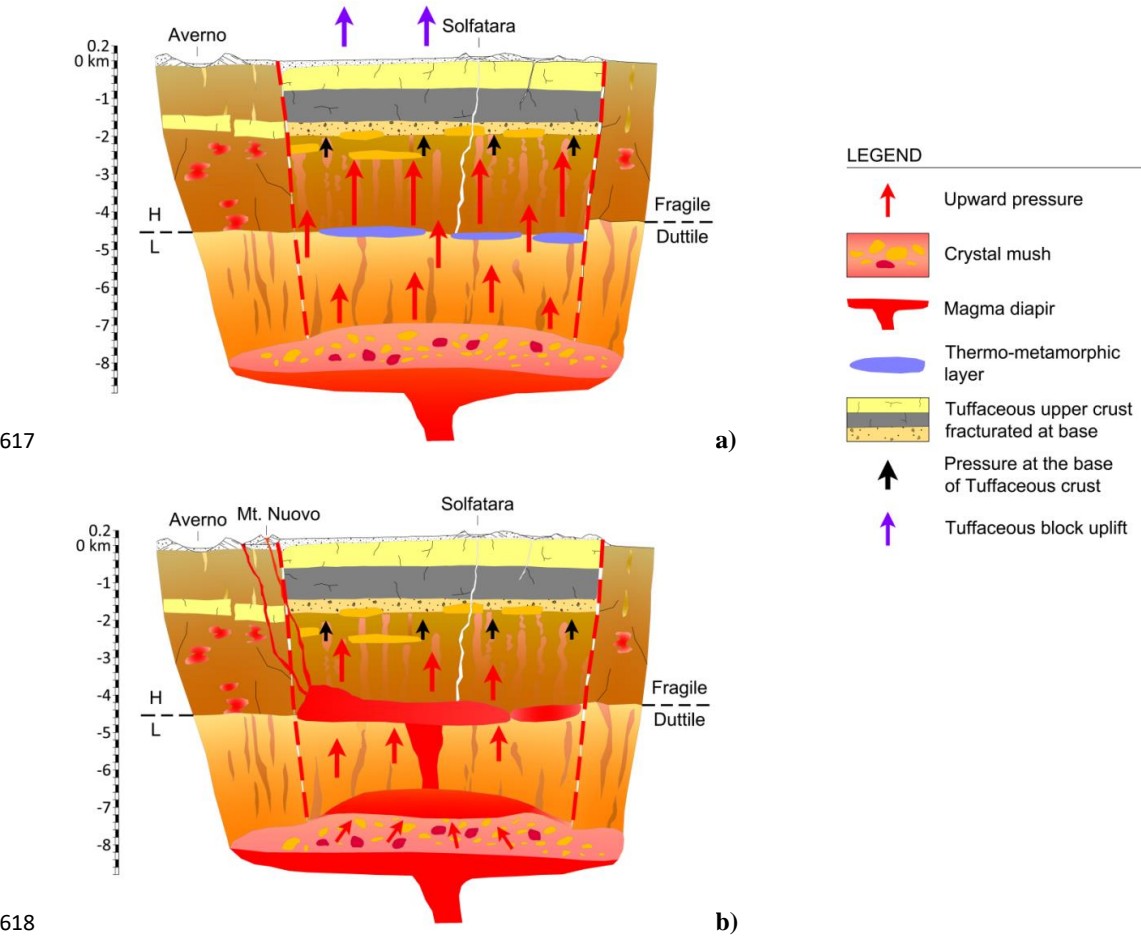

**Fig. 16 – Schematic cross sections of the hydrothermal and magmatic systems underlying the Campi Flegrei resurgent block in the 1538 AD, showing:**

**a) Process of gas sparging according to Bachmann and Bergantz (2006) model, related to the transfer of hot gas from a mafic intrusion underplating the trachytic mush and the hypothesized relation with earthquake swarms of the exsolved fluids, accumulated at lithostatic pressures in the ductile region and episodically injected into the brittle crust at very high strain rates. The sudden increase of fluid pressure, in the brittle region, triggers earthquake swarms in the 2-4 km depth range.**



**b) Remobilization of mush by mafic magmas then occurs, so that the magma remobilized from the mush accumulates at the top, fueling its rise upward to accumulate, in a sill-like shape, along the ductile-brittle transition surface. Eruption from the magma sill is then likely to occur at the faulted borders of the resurgent block.**

However, despite such a large volume of shallow intruded magma, the eruption of 1538 only produced about 0.03 km³ of pyroclastic deposits (see next section). This discrepancy likely suggests that multiple sill intrusions occurred over more than one century, with most of them solidifying without contributing to the eventual eruption. Only the most recent intrusion events, and/or some portion of magma mush from prior intrusions remobilized by subsequent heating, would have fed the eruption.

Another characteristic of eruptions from small monogenic volcanoes is their difficulty to forecast, as they occur at unexpected locations (Marti et al., 2016). Both distinctive traits were evident in the eruption of Monte Nuovo, which represents a prototype of a small monogenic volcano in the Campi Flegrei. Despite the relatively small volume of magma (0.03 km³), the eruption occurred at a considerable distance, approximately three km westward, from the area of maximum uplift. The position of the 1538 vent is approximately on the border of the resurgent block: such a border, marked by ring faults, clearly represents a weak zone, where magma can more easily intrude.

### 5.3 The eruption of 1538

The week preceding the eruption, was marked by a series seismic events (Guidoboni and Ciuccarelli, 2011). The shoreline gradually retreated 200 steps (ca. 370m) seaward, because of an occasional uplift occurred on the eastern shore of Lake Averno (see Fig. 2d) and during the 36 hours preceding the eruption, there were 7 meters of ground uplift (Parascandola, 1943; Costa et al., 2022). The local uplift rapidly attenuated as a function of distance, adding about 1-2m to the maximum uplift in Pozzuoli (Rolandi et al., 1985) (Fig. 6). The uplift, involving a local marine regression, was accompanied by strong rumbles on the night between 28 and 29 September, culminated in a further explosion, at 2 am on the following night, which marked the vent opening and the start of the eruption. The early eruptive column, initially white in colour, ejected muddy ashes and lithic and scoriaceous lapilli upwards. The presence of wet ash on the slopes of the gradually growing volcanic cone led Parascandola (1943) to hypothesize that it was a mud eruption. This description, present in the chronicles of the time (Parascandola 1943), indicates that the first eruptive phase was phreatomagmatic in character, although it evolved with a peculiar characteristic, because the volcanic cone was formed by massive pyroclastic units, made up of loose and wet deposits, ascribable to pyroclastic flows products with a prevalent sandy matrix, incorporating lithic and scoriaceous clasts. In Fig. 17a we recognize three main flow




units, each of them made up of sub-units. These sub-units are mostly evident in the finest basal part
(a), while in the intermediate part (b), showing abundance of scoriaceous clasts, an inverse gradation
is observed. Finally, the hydromagmatic activity, lasted about 12 hours, built a small tuff cone, formed
by successive waves of pyroclastic flow units, whose deposits reached a height of approximately 120
m. This particular type of hydromagmatic deposit implies an eruption in which the magma-water
interaction process is characterized by a low efficiency, considering the thermal energy of the magma
and the mechanical energy generating the eruption. In the classic Wohletz experimental diagram
(Wohletz et al., 2013), besides the fields 1 and 3 which include, respectively, eruptions with zero or

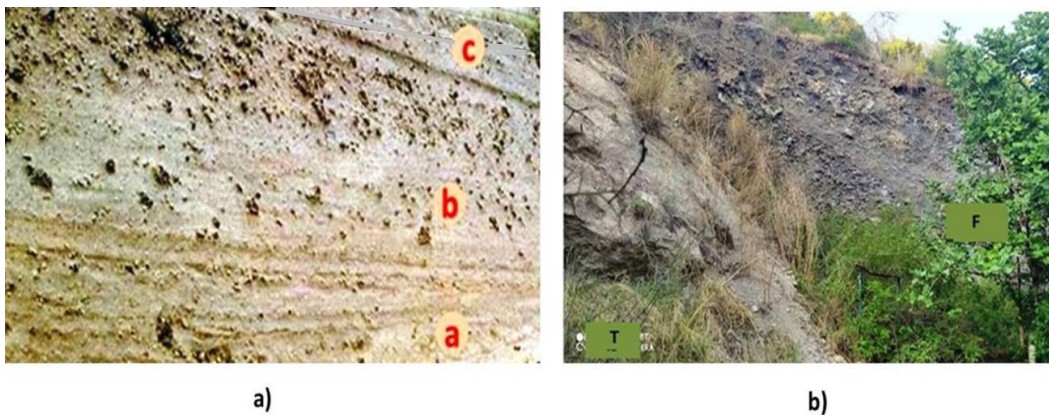

a)                                         b)

**Fig. 17 – a) Flow units in the phreatomagmatic Pyroclastic flows, b) Deposit of the final scoria flow (F) deposited in the western depression of the phreatomagmatic Tuff cone (T).**

low magma/water ratio (0 – 0.1) and those with extremely high ratios (100-1000), field 2 includes
hydro-magmatic explosive eruptions with an interaction ratio between 0.1 – 10, indicative of a greater
value of mechanical efficiency (Fig. 18). It is evident, however, that even in field 2 there is a
differentiation in efficiency, due to the condition characterizing the expansion of the water vapor that
develops during the magma-water interaction process, that is:

1) If the magma/water ratio is around the value of 0.3, the maximum efficiency is achieved. The
quantity of water is optimal and expands entirely as superheated steam, that is, the maximum volume
that can be generated is obtained without dispersing heat. Under this condition, the so-called Base
Surges are formed;

2) If the water content increases, the efficiency drops because not all water is vaporized, and, as a
resultsteam saturated with water is formed. Under this condition, Pyroclastic flows are formed.



This last type of flow is therefore associated with the collapsing eruptive columns that developed in
the night between 29 and 30 September, to be ascribed to a phreatomagmatic eruption with a high
magma-water ratio, which gave rise to the non-welded ignimbrites described in typology 2 and located
in the diagram of Fig.18a, at point a. This implied that in the initial phase of the eruption the magma
absorbed a considerable quantity of sea water present above the eruptive vent, so in these conditions,
the collapsing eruptive columns which gave rise to the pyroclastic flows on the night between the 29th
and 30th September, reached a maximum height of less than 3 km, (Parascandola, 1943), depositing
in a radius of approximately 3 km, as follows:
- with thickness of 5-10m, in sections obtained by cutting the slope in the area around the volcano (Fig.
17a);
- in a depression on the SE sector of the volcano. The materials of the Tuff Cone of Monte Nuovo (T)
are present, together with the products of the scoria flow (F) deposited in the SE depression (Fig. 17b).
It should be noted that, about 1km away towards the SE, in the direction of the Serapeum, the products
of the Tuff Cone display a thickness of about 5m and around the Serapeum itself

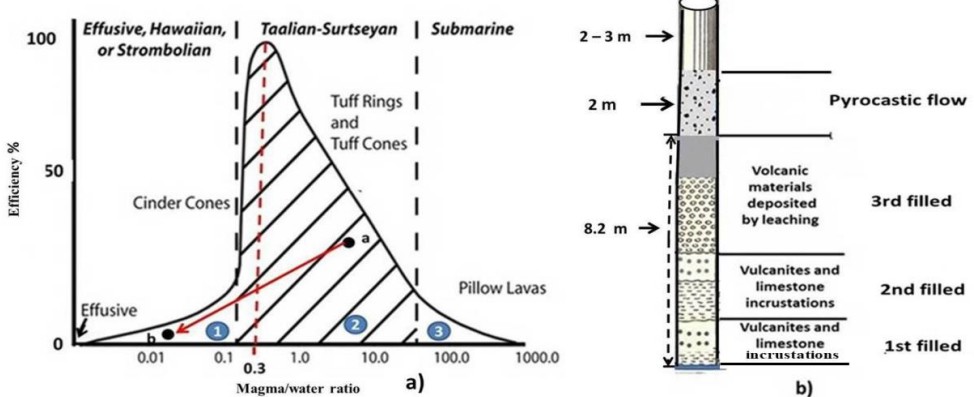


**Fig. 18 – a) Wohletz (1983) diagram for the evaluation of the mechanical efficiency of the**
**products emitted in the form of Pyroclastic flows and fall/flow from Strombolian eruption**
**column collapse, b) products emitted by the 1538 eruption in the first eruptive phase as wet**
**pyroclastic flow, which bury the upper part of the Serapeum columns (above 8.2 m of height).**

(about 3km away), the products show a thickness of about 2m (Fig. 18b). According to the chronicles,
on October 6th there was a new eruptive phase and 24 unwary visitors died, surprised by the
resumption of eruptive activity, which revealed itself with different characteristics, mainly magmatic,
that is, with a low water-magma interaction ratio (point b in Fig. 18a). In the hydromagmatic-magmatic
transition, the eruptive cloud took the characteristic 'cauliflor' shape of Strombolian eruptions, with a





height of about 4 km, which, driven by winds from the NW and then from the N, distributed the
scoriaceous products towards the SE in the direction of Nisida and the Neapolitan coast, then towards
the S, in the direction of Bacoli and Capo Miseno (Parascandola, 1943). The scoriaceous products of
the second Strombolian magmatic eruptive phase uniformly covered the basal units that formed the
volcanic edifice during the first phase, with an average thickness of about 0.5 m. The final phase of
the eruption occurred with the collapse of the Strombolian eruption column, which deposited a scoria
flow in a depression on the eastern side of the underlying cone of materials formed by phreatomagmatic
pyroclastic flow units (Fig.17b). Overall, the eruptive event of 1538, with the emission of 0.03 km$^3$ of
pyroclastic material, can be classified with a VEI = 2.

**6.    The seismicity before and after the 1538 eruption**

The main precursors of the eruption, as reported by chronicles, were the earthquakes. Earthquake
sequences preceded, accompanied and followed the 1538 event. In this context, seismic precursors
may depend on the occurrence of stress perturbation, determined by the arrival of magmatic gases, as
well as directly by magma intruded at shallow crustal levels (typically at depth of 3-4 km), originating
from the main reservoir located at about 7.5-8.0 km depth.
We analyze here the earthquake sequences that occurred before the eruption.

**6.1 Comparing past and recent earthquakes: from intensity to magnitude**
To better compare the past **earthquakes** with the recent and present-day seismicity recorded at Campi
Flegrei we must convert intensities in magnitude. In Fig. 19, we present a tentative correlation
between the epicentral intensity (Io) and the magnitude (ML).  Choosing the correct relation between
Io and ML is not straightforward, particularly in this case involving peculiar volcano-tectonic
earthquakes. Nonetheless, it is important to establish such a relation to compare the seismicity
observed during the 1430-1582 period, as inferred by Guidoboni and Cucciarelli (2011), with the
seismicity experienced during the recent unrests. To determine the Io-$M_L$ relation, we are confident
that, despite the availability of several formulas in the literature, the best approach is to consider a
precise geographical and seismotectonic context, especially in a volcanic setting. Different features
allow to discriminate between volcanic and tectonic earthquakes, which suggests caution in using
correlations derived from tectonic areas for volcanic earthquakes, and vice versa (Milana et al., 2010).
In order to build a realistic relation between seismic intensity and magnitude in this area, we utilized
the computed intensities of two earthquakes that occurred in the Campi Flegrei region in 1983
(Branno et al., 1984; Marturano et al., 1988; Milana et al., 2010; Charlton et al., 2020), during the



previous unrest of 1982-1984 (Troise et al., 2019). Additionally, we considered a M=5.0 earthquake
that occurred in the similar volcanic area of Colli Albani (Sabetta and Paciello, 1995). The M=4.0
earthquake occurred on October 4, 1983, at Campi Flegrei, was found to have a maximum intensity
Io=VII (Branno et al., 1984; Marturano et al., 1988). An earthquake of magnitude M=3.5, which
occurred in the same swarm on   October 4, 1983, was found to have a maximum intensity Io=V (Fig.
19: Marturano et al., 1988). Furthermore, Sabetta and Pugliese (1995) reported an earthquake of
M=5.0, with a maximum magnitude Io=VIII.
These correlations between intensity and magnitude were utilized to assign realistic magnitude values
to the macroseismic intensities deduced from the analysis of historical seismicity (Guidoboni and
Cucciarelli, 2011), as shown in Fig. 19.  They were also used to transform the magnitude of
earthquakes associated with recent unrest phases into macroseismic intensities, as we will discuss
later.

**6.2 The seismic phases that accompanied the ground uplift and the eruption**
We can classify the precursory earthquake sequences into three categories: long-term, medium-term
and short-term precursors.
- The phase of ***long-term seismic precursors***, preceded by historical reports of earthquakes of
doubtful occurrence, began to be well documented since 1468 - 1470, when a paroxysmal seismic
phase occurred (Io = VII) (Guidoboni and Ciuccarelli, 2011; Francisconi et al., 2019) (Fig. 19a –
interval A), resulting from a progressive increase in fracturing. This culminated into intense
fumarolic-hydrothermal activity recorded at the Solfatara volcano. The historical chronicles report
widespread damage to the vegetation, both spontaneous and cultivated, in all the areas surrounding
the volcano. This appears to be an important piece of information, indicating a broadening of the area
affected by intense degassing, (Francisconi et al., 2019).  In 1475, another seismic phase was reported
(Guidoboni, 2020), with maximum intensity Io = IV - V. Over the following twenty years, ground
uplift continued at an accelerated rate. This period culminated with a strong seismic phase occurring
in October 1498, reaching considerable maximum intensity (Io = VII). A low-intensity seismic phase
then followed during the period 1499 - 1503 (maximum intensity Io = V) (Fig. 19a – interval A).
Such a long-term precursory phase could likely be interpreted as mainly due to intense degassing,
coming from the deep magma chamber and progressively increasing pressure in the shallow layers
of the geothermal system, without significant contribution from direct magma intrusion at shallow
depth.
- After this first initial long-term precursory phase, a new phase of *medium-term precursors* followed.
This phase was characterized by stronger seismic events in 1505 and 1508, which were of higher





intensity with respect to the previous ones (maximum intensity Io = VIII) (Guidoboni and Ciuccarelli,
2011). Additionally, there was a faster ground uplift during this period, resulting in serious damage
to buildings and several casualties. This seismic phase could have been caused by either a higher
stress associated with increased uplift level, or magma intrusion, from the deep magma chamber into
shallower levels. This intrusion could have produced higher stress resulting in seismic activity of
greater intensity. Although it is obviously difficult to identify, from historic sources alone, the
respective roles of the deep degassing into the hydrothermal system versus shallow magma intrusion,
we believe that the reported evidence of vegetation damage and increased degassing in the first phase,
and the increase of earthquake intensity in the second phase, indicate respectively a main contribution
of degassing perturbing the hydrothermal system, in the first phase, and shallow magma intrusion in
the second phase. This phase concluded in 1520, with a medium intensity earthquake (Io = V-VI)
(Fig. 19a – interval B), likely again associated with perturbations in the hydrothermal system.


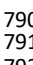

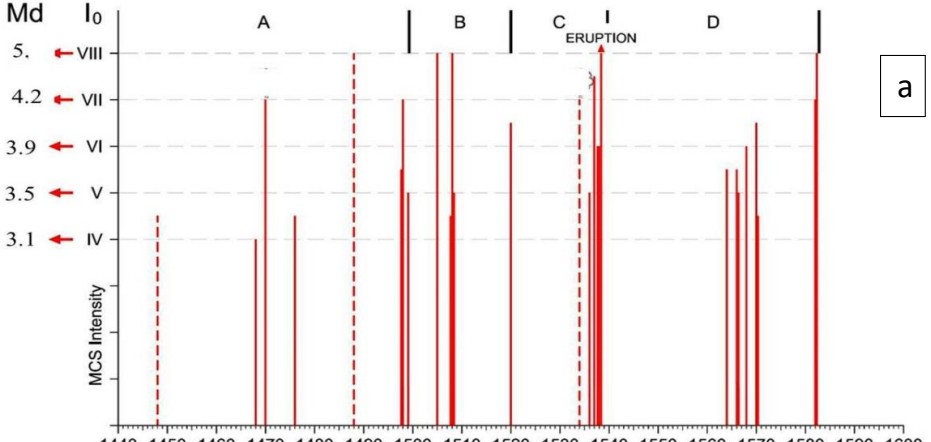






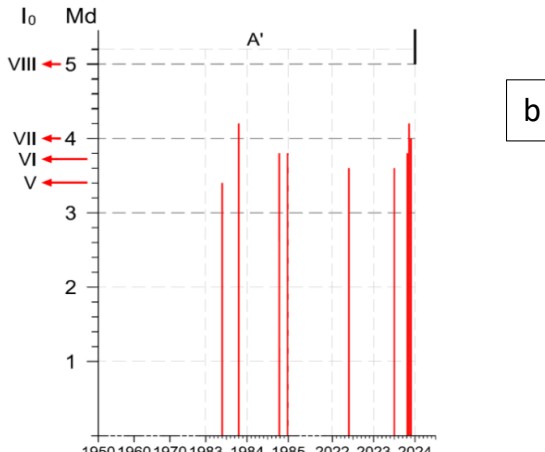

**Fig. 19 – a) Reported earthquakes occurred before and after the 1538 eruption (from Guidoboni and Ciuccarelli, 2011). The computed intensities of these earthquakes have been converted in magnitudes using the considerations made in the text. b) Highest magnitude earthquakes (M≥3.5) occurred since 1950 to present.**

- After 16 years of relative seismic quiescence, likely characterized by low-intensity earthquakes not reported in chronicles, a short-term precursory phase began in 1536. It commenced with continuous seismicity, without major damage (Io = III -IV), continuing with similar features until the early 1537. It is possible that this last seismic phase, characterized by relatively low magnitude, was caused by low-frequency seismicity, resulting from magma oscillations during the fractures opening (see Chouet, 1996). This seismicity became more frequent just before the eruption. In February of the same year, the seismic activity peaked with stronger events (Io = VI - VII), accompanied by an increase in the fumarolic activity at Solfatara. This provides clear evidence that this seismicity was again related to perturbations in the hydrothermal system. A final increase in seismic activity (Io = VIII), began in mid-June 1538, accompanied by a 7-meter ground uplift at the eruption site, located 3 km away from the center of previous maximum uplift. (Fig. 19a – interval C) (Parascandola, 1943, Rolandi et al., 1986; Guidoboni and Ciuccarelli, 2011; Guidoboni, 2020). The claim made by Di Vito et al. (2016) regarding very large local uplift at the eruption site, exceeding 18 m., appears to be inaccurate. Historical chronicles from the time indicate that the Roman road 'Via Herculea', which was submerged during the subsidence phase and was at about 7 m of depth in 1430, did not re-emerge during or after the 1538 eruption. Given the proximity of the Via Herculea to the 1538 vent, this suggests that local uplift there should not have exceeded  ca. 7 m.



- On 1538, approximately 0.03 km$^3$ of emitted products, through phreatomagmatic activity with low
mechanical efficiency (Rolandi et al., 2023). After six days the eruption resumed with Strombolian-
type magmatic activity, mantling the tuff cone with a 0.5 m thick blanket of dark trachytic scoria. The
final phase of activity ended with the collapse of the Strombolian eruptive column, resulting in the
deposition of a scoria flow in a depression on the south-east side. Monte Nuovo has been the second
smallest volcanic eruption (and volcanic edifice) of the post-caldera activity, with VEI = 2 (Rolandi
et al., 2023).

**6.3 The post-eruption seismicity**
We will now consider the seismic phase following the eruption just described which we will indicate
as the ***aftereffect of the 1538 eruption***. This phase was likely triggered by continuing degassing from
the deep magma chamber, and/or by new episodes of shallow magma intrusion not reaching the
surface to erupt. It began in 1564 with earthquakes of medium intensity (Io = V - VI), followed by a
phase of lower intensity 2 years later. In 1570 seismic intensity increased (Io = VI - VII), causing
damage to the buildings of the city of Pozzuoli. Between 1575 and 1580 a new phase of low seismic
intensity began, culminating, in 1582, with two earthquakes, respectively of intensity Io = VII – VIII.
These earthquakes caused partial collapses in several houses and serious damage to churches and
buildings, as well as numerous casualties (Parascandola, 1943; Guidoboni e Cucciarelli, 2010;
Guidoboni, 2020).

7. **Comparison of precursory phases of 1538 eruption with current unrest**
This study is mainly aimed at understanding how the evolution of the ground movement phases linked
to the 1538 eruption can help build realistic scenarios for the evolution of the same recent phases at
the Campi Flegrei caldera. Common features between the medieval and present-day ground movement
phases are described in the following:
The main similarity is that the seismicity, in the past and in the recent unrest, has been clearly correlated
both with the total uplift and the uplift rate; it is practically absent in periods of subsidence (Dvorak
and Gasparini, 1991; Kilburn et al., 2017; Troise et al., 2019).
We found, in particular, that seismicity of period 1950-2024 is on the same order than the period
1430-1503, whereas the latter, as we have previously observed, was the first phase of preparation of
the 1538 eruption. Although the total amount of uplift in the period 1430-1503, about 10 m, was more
than double than the total uplift recorded since 1950-2023, of about 4.1 m., the seismicity in the two
periods has been remarkably comparable. The maximum magnitude, M=4.2 recently occurred on



October 2$^{nd}$, 2023, is in fact very similar to the maximum magnitude reconstructed for the period
1430-1503 (Fig.19a interval A and Fig.19b interval A').
Another common feature is that both seismic phases can be mostly ascribed to the effect of
pressurized hydrothermal fluids. So, till now there is a close analogy between the 'long term
precursory phase' preceding the 1538 eruption and the recent unrest 1950-2023; the only clear
difference is, as we already noted, the much lower cumulative uplift of the recent unrest.
Such observations led us to consider two possible scenarios for the evolution of the present unrest.

**7.1 First scenario**
The first scenario would imply that the present unrest progresses towards a new eruption. Although
there is, presently, no evidence for shallow magma intrusions occurring during the present unrest
since 2006 (see Moretti et al., 2017, 2018; Troise et al., 2019), a new shallow magma intrusion, in
the near future, cannot be ruled out. Another possibility is that the mush, which should be present at
low depth, could be re-mobilised by hot fluids coming from the main magma chamber. Troise et al.
(2019), showed in fact evidence for a likely shallow magma intrusion occurred at about 3 km of depth,
during the 1982-1984 unrest, with a volume of about 0.03 km$^3$, i.e. the same order of magnitude of
the erupted volume in the 1538 event. The same authors calculated that such a sill intrusion should
have solidified, in form of mush, after about 20 years, i.e. around 2003. If the actual unrest will
progress towards an eruption, it is also very likely that seismicity will increase, in frequency and
magnitude, possibly reaching magnitudes around 5 or even higher. Earthquakes of magnitude 5, in
this area, would occur at very shallow depths (not higher than about 3 km), so producing high
intensities (higher than VIII MCS, see Fig. 19). Finally, from a civil protection perspective, we must
also take into account the possible onset of a post-eruptive seismic phase, which after the 1538
eruption lasted about 40 years. In conjunction with the prefigured scenario, the problem of forecasting
the position of a new eruptive vent is also extremely relevant because, in principle, it could be opening
in any sector of the caldera. Despite the indications contained in several probabilistic studies on the
subject (Alberico et al., 2002; Selva et al., 2011), we must consider they are biased by the assumption
of stationary conditions, which is implied in any probability computation based on the frequency of
past events. As the most evident example that such probabilistic determinations have a poor
reliability, it is enough to note that, on the basis of such calculations, the site of the 1538 Monte
Nuovo eruption would have never been predicted. The most reliable indication of the most likely
future vent could come from the most seismic areas, because they reflect the areas of maximum shear
stress. In this perspective, the Solfatara-Agnano area (see Fig. 15a), which is by far the most
seismically active, could be the most probable site for future vent opening. However, the most




effective way to address this problem would be the prompt determination of localized uplift in
addition to the usual bell-shaped one centered on Pozzuoli harbor. Although some recent eruptions
(e.g. at Hekla volcano: Wonderman, 2000) show that the rise of magma from several km to the surface
can be so fast to be practically useless for civil protection purposes, localized and considerable
ground uplift was actually observed well before (months or years) the 1538 eruption, making it likely
that this precursor will be observed before any future eruptions in the area.
We should however mention the possibility that, even without new shallow magma intrusions, and/or
in absence of mobilized mush eruption, the increase of pressure for aquifer heating above the critical
threshold could produce a phreatic eruption. Phreatic eruptions are in general very difficult to
forecast, and also to detect from the past geological record. However, there is some robust indication
for at least one phreatic eruption occurred in the area, in 1198 (Scandone et al., 2010).

**7.2 Second scenario**
As an alternative scenario, we should consider the one which stops sometimes without evolving
towards an eruption. Despite the similarity of the recent unrest with the first phase leading to the 1538
eruption, we could in fact consider the notable difference in the cumulative uplift between the past
and present unrests: 10 m., as compared with 4.1 m. The level of ground uplift is critical, because it
indicates the level of stress accumulated underground. As pointed out by Kilburn et al. (2017), when
the level of stress reaches a critical value, the medium rheology becomes totally fragile and any small
amount of incremental stress can cause the collapse (i.e. the catastrophic fracturing) of the shallow
crust, thus producing the eruption. Actually, we don't know the critical stress level for the shallow
crust at Campi Flegrei. Kilburn et al. (2023) claimed, from the observation of the trend of cumulative
number of earthquakes as a function of cumulative uplift, that such critical value would have been
reached and overcome in 2015. However, looking at the data they present, no reliable change in the
trend of seismicity after 2015 can be really observed; furthermore, their assumption that the maximum
internal stress reached in 1984 has been overcome in 2015 is not justified, because only in June 2022
the maximum ground level reached the same maximum value of 1984 (Osservatorio Vesuviano,
2022). Besides any speculation, it is clear that, if the internal stress had really overcome the critical
level in 2015, considering the large additional uplift cumulated since then (about 0.85 m.), and hence
the considerable incremental stress, the system would have already been collapsed, and an eruption
occurred. The very high deformation occurred before the 1538, namely 16 m plus the localized uplift
occurred just at the vent site before the eruption, seems to indicate that the critical stress level is much
higher than the one presently reached. Therefore, there is a possibility that the progression towards



eruption conditions is too gradual to culminate in an actual eruption, and the unrest may cease before
reaching that point.

## 8. **Conclusion**

In this paper, we have presented a detailed reconstruction of the ground deformation, and a
comprehensive analysis of the main observations characterizing the events before, during and after the
1538 Monte Nuovo eruption, the only eruption occurred at Campi Flegrei caldera in historical times.
This reconstruction has allowed us to correct some widely diffused but erroneous reconstructions,
found in the past and recent literature, based on clear historical evidence. Specifically, we
demonstrated that subsidence in the area began during the Greek colonization (VIII century BC) and
persisted through Roman times, with documentation dating back to 90 BC. Additionally, we
reconstructed the evolution of ground deformation at Pozzuoli harbor during the Middle Age,
demonstrating that maximum subsidence occurred around 1430. We also tracked the ground level from
1430 until the first half of the 19$^{th}$ century, using historical data on the height of the Serapeum floor
relative to sea level.
Furthermore, by reconstructing the subsidence and uplift of the Via Herculea, based on ancient
chronicles, we provided clear evidence indicating that the local uplift preceding the eruption at the
Monte Nuovo site, situated near Via Herculea, did not exceed 5-7 meters. This evidence disproves
claims in recent literature (Di Vito et al., 2016), that suggested local uplift around M. Nuovo, reached
elevations as high as 19 m immediately before the eruption.
Our reconstruction of geophysical anomalies (mainly ground displacement and seismicity) preceding
and following the 1538 eruption has been tentatively interpreted in comparison with observations and
data collected during the recent unrests. This approach has enabled the formulation of two possible
scenarios for the evolution of the present unrest, which, so far, has shown notable similarities to the
long-term precursors of the 1538 eruption.
The first scenario involves the progression of phenomena towards an eruption, suggesting that, in the
near future, earthquakes with magnitude up to 5 or slightly higher may occur, both preceding the
eruption and persisting for several decades afterward. Conversely, the alternative scenario, implies that
the unrest may cease before an eruption occurs. This possibility is supported by the fact that ground
uplift observed from 1950 to 2023, compared with the uplift occurred over an equivalent period from
1430 to 1503, is significantly lower (4.1 m as compared to 10 m). Since the overpressure in the system
is somewhat proportional to the amount of uplift, it is plausible that the recent unrest has not reached
the critical value for catastrophic fracture of shallow rocks. In addition, if cumulative stress increases
too slowly, a substantial amount of previous stress can be cleared depending on viscoelastic relaxation
and its characteristic times. While the exact critical threshold and viscoelastic relaxation time remain



unknown, they can be tentatively inferred from the maximum deformation observed before the 1538
eruption. The bell-shaped cumulative vertical displacement centered at Pozzuoli, before the 1538
eruption, was much larger, reaching 16 m., compared to the about 4 m recorded from 1950 to 2023.
This substantial difference, assuming the rheology and strength of shallow rocks in the 0-3 km depth
range remain unchanged, suggest that we are currently far from reaching the critical stress threshold
necessary for an eruption.

**Data availability**
All raw data can be provided by the corresponding authors upon request.

**Author contributions**
GR, GDN and CT analyzed historical and volcanological data; GDN and CT analyzed earthquake
intensity/magnitude data; MS analyzed seismic data; GR, MS and MDL wrote the manuscript draft
and prepared the figures; GDN, CT and MS reviewed and edited the manuscript.

**Competing interests**
The authors declare that they have no conflict of interest.

**Acknowledgments**
The authors want to thank Prof. Marina Petrone who helped to recover some important Middle Age
references on Campi Flegrei.

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

inside the inlet, which took the shape of a lake (Fig. 2a and Fig. 4).

**2-** This point can be traced back, from a historical and chronological point of view, to the 8th century BC. In
the diagram it is positioned at approximately 5 m above sea level, suggesting a subsidence of the coastal bar
of about 2 m from the previous point. In fact, from a writing by Diodorus Siculus (Book IV) we know that::*..*
*this dam was continually invaded and ruined by the stormy sea, which often made it impassable…*It is known
from coastal dynamics studies that waves breaking against a dam, placed above a seabed 7 m deep, reach a
height equal to 3/4 of the depth of the same seabed, in this case approximately 5 m, i.e., a height equal to the
barrier above the sea level. Therefore, the via Herculea, hit by violent waves, constituted an impassable road
for the inhabitants of Cuma to reach the lands they cultivated in the surroundings of Pozzuoli, which, starting
from the 8th century, took the name of Via Herculea (Fig. 2b and Fig. 4). Finally, the hypothesis of a height
of 5 m, as resulting from submersion started since the 17th century BC, seems likely.

**3- 4 -** The body of water formed by the coastal bar, in the 1st century BC, was owned by Sergio Orata. The
lake, making generous profits from fish farming, was named "*Lucrino*", derived from the Latin Lucrum (profit)
(Fig. 2c). The owner, around 60 BC, to protect his interests turned directly to the Roman Senate to have the
Via Herculea repaired, because at that time, being at a height of about 2 m above sea level, it had almost been
destroyed by the waves that crossed it, preventing him from practicing his lucrative fish farming business
(point 3). The Senate appointed Julius Caesar, who in 59 BC built a breakwater barrier, located outside the
dam towards the open sea (Opus Pilarum). He also ordered the installation of canals closed by opening
platforms (Claustre). Julius Caesar's project defended the Via Herculea essentially from the horizontal force
exerted by violent wave motion, not understanding the effect of subsidence. In 37 BC, general Agrippa, by
order of Octavian, engaged in the naval war against Pompeo Sextus, chose the coastal sector between the lakes
Lucrino and Avernus for the construction of a new military port system, called *Portus Julius*. A new main




entrance was built, consisting of a canal with two long banks in 'opus pilarum', cutting and equipping the Via
Herculea with a mobile bridge, to access its interior, while at the same time widening the narrow opening that
connected the Averno and Lucrino lakes to allow access of large ships in the shipyard (Fig. 2c). Furthermore,
Agrippa reinforced the Via Herculea and added piers, supported by orthogonal pillars and having also sensed
a problem of subsidence,... ***raised its level (Strabone, 1 century BC-1 century AD)*** (point 4).

**5- 6 -** The abandonment of Portus Julius by the Roman fleet, starting from 12 BC, as well as of the remaining
part of Lake Lucrino, due to the impossibility of continuing fish farming, was the result of the continuing
subsidence, which, according to Aucelli et al. (2020), between 37 BC and the beginning of the 1st century AD
further accelerated.
In the 5th century AD the dam, few meters above sea level (point 5), was also damaged by a violent sea storm.
An attempt to restore the dam again was made by Theodoric, regent of the Ostrogothic kingdom in Italy from
493 AD, who decided, in 496 AD, to repair the damage and probably also raised its level (***Cassiodorus, Varia,***
***Book 1***) (point 6). This can be also deduced from the fact that Lake Lucrino was still well identified in 522
AD (G.C. Capaccio - Puteolana historia, in Parascandola 1943).

**7-8 -** Around the second half of the 6th century (556 AD), some fishermen attempted to reactivate fish farming
in Lake Lucrino, but the dam soon could not guarantee an adequate yield, because it had reached a height of
just a few meters above sea level (point 7), not allowing fish farming (Parascandola, 1943).
As we will show in Appendix-2, historical documents indicate that, at the lower city around Pozzuoli, the
famous Serapeo (Macellum) began the phase of submersion below sea level in the 4th-5th century AD. At the
area facing the Avernus, the above historical documents indicate that the submersion most likely occurred
between the 6th and 7th centuries AD. This could be related to either height increasing interventions and /or
to a lower speed of subsidence at the site of Via Herculea, as compared to the Serapeo.

**9 –** In the 14th century we have evidence of the submersion through the writings of Petrarca and Boccaccio.
Below we will report some sentences from the two poets, giving indications on the subsidence in this period
(Parascandola 1943):
- Petrarca, who lived in Naples in 1341, visited the coastal area of Avernus, ***(…I then saw the places of***
***Avernus and Lucrino........ and the superb road of Gaius Caligula now swallowed up by the waves….. Note***
***that Opus Pilarum mistakenly believed to be the road of Caligula).*** From this observation we deduce that
Opus Pilarum was submerged in the 14th century (Fig. A1). From the same observation it further seems likely
that, since the 4-5 m high pylons, submerged for a couple of metres, are not visible, and given the pylons were
higher than Via Herculea of about 3 meters, the already submerged Via Herculea should have been submerged
at that time for about 5-6 m*.*
**-** Boccaccio came to Naples in 1348 and, after visiting the Averno area, he clearly expressed the concept,
although indirectly, that Lake Lucrino was not recognized as it was invaded by the sea, mixing with the waters



of Avernus (…*to Avernus, connected in ancient times with the nearby lake Lucrino where it recalls the*
*waters of portus Iulius:* Boccaccio, 1355-1373).
Boccaccio noted that, since there was no barrier on the Via Herculea which formed the Lucrino, the rough sea
even broke into Lake Averno. Therefore, we can undoubtedly say that in the 14th century via Herculea was
completely submerged and Lake Lucrino disappeared because it was invaded by the sea.

**10 -** As we will demonstrate later, in the 15th century the ground movements of the Campi Flegrei area changed
from subsidence to uplift. The uplift began, the actual amount of which in the Averno area can be only given
in an approximate but equally significant way, because it is ascertained, from the writings of all the chroniclers
of the time (see Parascandola, 1943) that the Via Herculea did not re-emerge in this period (fig 2d). What is
reported by the historian San Felice is almost common to all the chroniclers: ***The sea had taken possession of***
***Lucrino, so that the name could no longer be given to the ancient lake.***

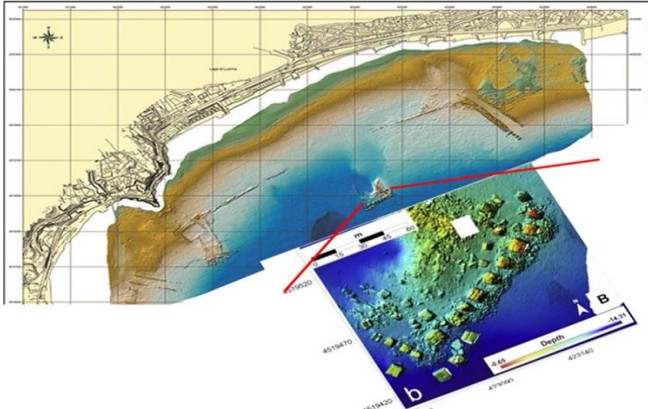


1440       **Fig. A1 - The remains of the Via Herculea currently located at 4-5m bsl, with the columns of Opus**

**Pilarum approximately 300m away in the open sea. An enlargement of the structure of Opus Pilarum is**
**also reported**
Shortly before the eruption, the general caldera uplift was also accompanied by a localized uplift of the area
where Monte Nuovo would have risen shortly after, in 1538, located in close contact with the Lucrino basin
(Fig. 2d). Such a local uplift was estimated at about 7 m (Parascandola, 1943), so the Via Herculea would
certainly have emerged if it had been close to the sea surface at the end of the 15th century. A significantly
larger uplift, of 19 m as hypothesized by Di Vito et al (2016), can be certainly ruled out from the observation
that Via Herculea did not reemerge.
The topic of the local uplift before eruption is relevant, so we insist on other aspects linked to the entire area
buried by the products of 1538 Monte Nuovo eruption. Until a short time before the eruptive event, two small
tuff hills, called *Montagnella* and *Monticello del Pericolo* (Parascandola, 1936), overlooked the Averno Bay,



above which the *village of Tripergole* extended. This village, thanks to the Angevins, developed with the
construction of a hospital with 30 beds, to access the numerous springs and thermal facilities available to the
hospitalized patients, with an adjoining pharmacy. Ancient buildings used for thermal baths (*Trugli*) present
in the Tripergole area were highly compromised between the end of the 15th century and the beginning of the
16th, when the Pozzuoli area was hit by major earthquaks. The earthquakes caused extensive damage to the
thermal health and ecclesiastical buildings of Tripergole, but not so devasting than expected if a ground uplift
about 20 m high would have occurred. Also the so-called **Temple of Apollo,** still present along the north-
eastern bank of the Averno lake (Fig. A2), testifies against a so large and sudden uplift. The structure is an
imposing building identified as a grandiose thermal room, covered by a dome, now partly collapsed, which
measured approximately 38 metres in diameter, built in the 1st century AD to exploit a series of hydrothermal
springs along the eastern side of Avernus, then expanded with the large octagonal hall (the one that is still
visible) in the following century. This structure was identified by Biondo da Forlì as the bathroom of Cicero
(Lanzarin, 2021), that, due to its particular location protected by the Averno crater belt, was not involved in
the burial of the *Monticello del Pericolo*, the *Montagnella* and the village of *Tripergole*, with its renowned
thermal baths.

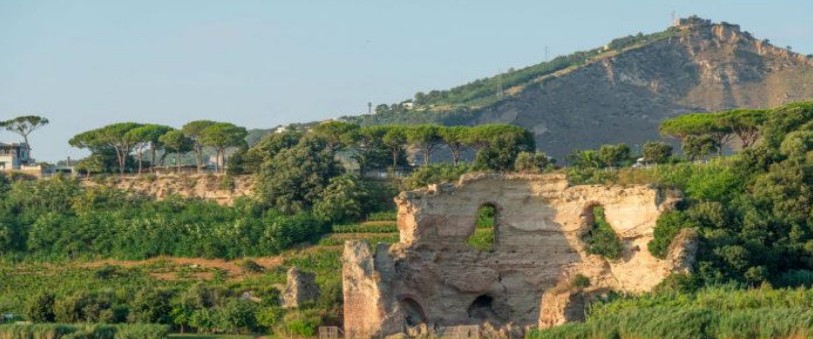


**Fig. A2 – The so-called Temple of Apollo on the east bank of the Avernus. You can see the remains of a**
**circular building with a "cap" vault, which later collapsed, typical of a "Truglio", i.e. a spa building**
**(internet source)**

**Appendix 2 - *Evolution of the ground movements involving the Pozzuoli area***

Phases of submersion during the Greek age have been detected in the Pozzuoli area by Gauthier (1912),
specifically in the eastern sector of Agnano. The author discovered Greek walls beneath the ruins of Roman
baths which were restored in the 6th century AD. These, in turn, underlie lacustrine sediments that filled an
ancient lake originally existing within the Agnano crater. However, the most evident subsidence phases have
been recorded since Roman times, by the structures of the so-called Temple of Serapis in Pozzuoli. Built in



the 2nd century AD and restored and completed in the 3rd century AD, during the Severan era, this structure exhibits the typical architecture of a Roman market ("Macellum").

To determine whether the construction preceding the 2nd century AD had a connection with a temple, we must go back to 105 BC, when a contract was stipulated between the municipality of Pozzuoli and a college of builders for repairs of public buildings (lex parieti faciundo). Among these was the Ades Serapis (Parascandola 1947), indicating that a temple dedicated to Serapis, (an Alexandrian deity often regarded as protector of merchants and sailors) existed during this period. By the end of the 2nd century BC, the cult of Serapis had spread throughout the Mediterranean and its sanctuaries, as well as those of other Egyptian deities, were frequented by Roman-Italics. It is probable, therefore, that the introduction of the cult of Serapis in Puteoli is related to the presence of an Egyptian community in the Puteolan port (Soricelli 2007). It is important to try to establish the relationships between this building and the Macellum built later, specifically whether the Ades Serapis could have an ancestral link with a more recent cult building, that was then transformed into a typical Roman market. This relationship is suggested by the discovery of a statue of Jupiter Serapis during the excavations of the Macellum in 1750 (see below). However, data reconstructed by Amato and Gialanella (2013; Fig.3), indicate that the first floor present in the substrate below the Macellum dates from the Flavian period (69 -96 AD). The finds in the reworked pyroclastic materials which are 4 meters thick below the first floor indicate a chronological interval between the end of the Republic and the beginning of the Empire (44 BC - 14 AD). This suggests that the Ades Serapis was likely built in a different position from the macellum, with which it therefore has no ties. The architectural elements of Macellum are part of the restoration works carried out on the Serapeo during the Severan Age (194 - 235 AD), with the installation of the 4th floor around 230 AD, located approximately 2 m above the 3rd floor. The existing structure (Fig. 6), still present in the same area today, provides important evidence for reconstructing the ground movements. These movements can be identified in:

∗The marble floor of the macellum (4th floor; see also Fig. A3b);

∗ The height of the three columns of the pronaos (12.70 m high, with the first 6.2 m displaying a 2.70m band perforated by lithophagus colonies (Fig. A3).

The historical information about the ground movements, is schematized in Fig. 6 of the main text, as follows:

**1 -** In the 2nd century AD the 3rd floor of the Serapeum reached approximately 1m above sea level.  It was sporadically invaded by the sea, to the point that, it was considered appropriate to build a 4th floor in 230 AD, located at 2m above sea level.

**2 -** The flooding progressively affected the coast, leading to the transfer of ships from the port of Puteoli to Constantinople in 325-330 AD (Gianfrotta 1993).  It is important to highlight that the 4th floor was invaded by the sea in 394 AD. The bank was restored on the left side and the right side of the macellum, in the area where structures functional to the port and the emporium were located, and to protect it from the sea waves with the construction of coastal embankments. These important works were supervised by the Campanian Consul Valerius Hermonius Maximus (Camodeca 1987, Caruso 2004).




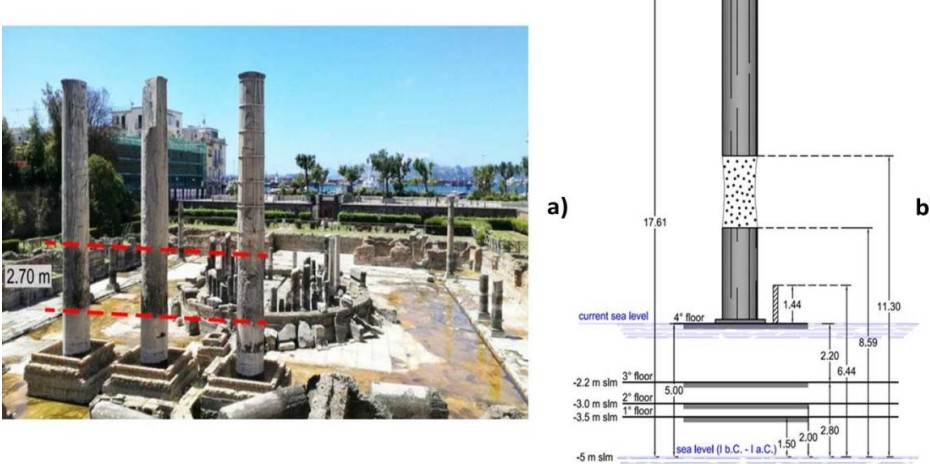


**Fig. A3 – a) Macellum showing pronao columns, b) Floors underlying columns**

**3** - In the 6th-7th century, the citizens who had completely depopulated the lower part of Pozzuoli felt the need
to take refuge in a sort of fortified citadel (castrum), equipped with a drawbridge, giving rise to the Acropolis
of the Rione Terra (Varriale 2004).

4 - In the 9-10th century, according to Parascandola (1943), the maximum submersion of the 4th floor of the
Serapeum occurred. Due to the subsidence of the Pozzuoli area, between the 8th and 10th centuries, the Agnano
Plain, immediately east of Pozzuoli, was invaded by water for the stagnation of thermal and rainwater,
transforming it into a lake (Annecchino, 1931).

**5 -7 -** In such a context, the most critical periods of the submersion phase occurred. The sea increasingly
surrounded the Rione Terra, that appeared like a medieval village, with a drawbridge at the entrance to the
cliff. The same context was depicted in the 11th century by the Arab geographer ***Idrisi*** in his ***Opus***
***Geographicum***, describing Pozzuoli as a ***"castle"*** (Varriale, 2004).
In the 12th century subsidence was still active. A writing deriving from an account of Benjamin ben Yonah
de Tudela who, visiting the Jewish communities of the Mediterranean, passing through Pozzuoli, described:
***turres et fora in acqua demersa quae in media quondam fuerant*** (Russo Mailer C. 1979, Caruso 2004). The
Pozzuoli district continued to subside in the 13th century, as can be deduced from an account written in 1251
by the historian Niccolò Jamsilla (***Historia de rebus gestis Frederici II imperatoris ejeusque filorum***
***Corradiet Manfredi Apuliaeet Siciliae regnum***) describes the places between Agnano and Pozzuoli as
follows: …***videlicet Putheolum mari mantibusque inaccessibilius circumquaque conclusum…***(Fuiano

1541 1951).



In essence, what was observed by the Arab geographer Idris in the 11th century, was also written by the
historian Jamsilla in 1251, confirming that Rione Terra "was ***an unapproachable mountain completely***
***surrounded by the sea".*** This highlights that, over more than 3 centuries, the sea level rose due to subsidence
of the tuffaceous walls of the Rione Terra.

**8 –** Further eyewitness accounts from by Boccacio, who lived in Naples between 1327 and 1341, reported that
a fisherman's wharf in the Bay of Pozzuoli became completely submerged (Mancusi, 1987). This document
supports the description of the lower part of the city being completely submerged.

**9 – A** gouache from 1430, known as ***Bagno del Cantariello***, part of the famous Balneis Puteolanis of the
Edinburgh Codex (Di Bonito & Giamminelli, 1992) indicates the complete submergence of the 4th floor of
the Serapeum by at least 10 meters. (Fig. 7). This context is supported by a description from 1441 indicating
that in 1441 "*the sea covered the littoral plain, today called Starza"* (De Jorio, 1820; Dvorak and
Mastrolorenzo, 1991) (see Fig. 8).
For a more precise description of this morphological context, it is useful to refer to the excavation of the
Serapeum carried out in 1750, when this monument was freed from the blanket of sediments that buried it (see
Fig. 12), made up of approximately 8 m of filling sediments, plus two meters of deposits from the pyroclastic
flow of the M. Nuovo eruption. By replacing the latter materials with the approximately 2 m blade of sea water
in the 1430 scenario (Fig. 7c), we arrive at the landscape picture in Fig. 7a, exemplified in Fig. 8d.