# Peer review of "The 1538 eruption at Campi Flegrei resurgent caldera: implications for future unrest and"

_EGUsphere, 2024_

## Author Response (AR1)

**Authors' Response to reviewers' comments**

**Response to first reviewer**

Review of: "The 1538 eruption at Campi Flegrei resurgent caldera: implications for future unrest and eruptive scenarios", by Rolandi et al.

**Overview**

This paper contains a great deal of useful information about historical unrest at Campi Flegrei. The information comes from the published literature and from new reconstructions of the volcano's behavior. The large amount of information has led to a lack of focus in the main theme and the mixture of old and new data has obscured its novelty. In addition, the conclusions have not obviously been derived from the observations provided. Judicial editing and restructuring of the text would yield a paper in which the logical thread of the analysis and the advance in understanding are made clearer. I have made numerous suggestions below. The number of changes indicates major revision. Even so, the adjustments should be straightforward.

Common themes are:

(1) Reorder the text. One option is:

- Describe observations and current views on ground movement and seismicity before the Monte Nuovo eruption in 1538 and since the unrest that began in 1950. Describe the key features of the subsurface structure of the volcano and connect this to the processes (*g.*, transport of gas and magma) that may be contributing to unrest.

- Describe *new* reconstructions of events before 1538. There is no need to repeat in detail previous studies – these can be acknowledged by citations in the text. Give just the key points and show how they have been modified by the new studies here.

- Compare the pre-1538 and post-1950 behavior to investigate whether the same sequence of events can account for both sequences. The different roles of the transport of magma and magmatic fluids could be a key topic.

- Discuss future scenarios based only on the data presented.

Reply: We thank the reviewer for the useful suggestions, so we re-organized the paper the way he suggests.

(2) Clarify which material comes from previous studies and which is new to this paper. Much of the historical data has appeared elsewhere and does not need to be described at length here. Previous work must be fully cited: this would highlight the **novelty** of this work.

Reply: Yes we have done it in the revised version.

(3) Remove extraneous topics that do not contribute to the goals and conclusions of the paper (*e.g.*, the account of the Monte Nuovo eruption).

Reply: We aim to emphasize that the account of Mt. Nuovo eruption significantly contributes to the objectives and conclusions of thispaper. The study focuses on accurately describing the events occurred before, during and after the 1538 eruption, in order to provide insights into potential future scenarios. In fact, we highlight some observations that have been never reported in previous studies, which are crucial for assessing what could happen during a future eruption. As an example, it is generally thought that only ash fall from the 1538 arrived at Pozzuoli; but we unambiguously identified, in outcrops at intermediate distance, pyroclastic flows arriving there, and forming the about 2 m of deposits found on the Serapeum columns. This important finding implies that, even in case of a very small eruption like the 1538 one (VEI=2) pyroclastic flows can reach over 3 km of distance on flat terrain.

(4) Explain the significance of a resurgence occurring as a block. Repeated pressurization at a common depth (which may happen to be shallow) is expected to produce the observed patterns of ground movement without needing to invoke wholesale movement of bounding faults (see for example Acocella (2019)). Hence, it would be good to have further guidance on the importance of whether or not fault-bounded movement is essential and whether this can constrain the potential for eruption. For instance, might a block model be appropriate only after a critical amount of movement has been achieved (*e.g.*, movements before 3.700 years ago) but not significant otherwise (*e.g.*, movements before 1538 or since 1950)?

Reply: There is substantial evidence supporting the central block resurgence at Campi Flegrei, as reported in the cited references. The necessity of differential movement along the bounding faults has also been thoroughly assessed in numerous studies: not only to reproduce the observed ground deformation pattern, but also to explain the occurrence and distinctive characteristics of seismicity. In this paper we simply recall and reorganize this evidence to explore its implications for unrest evolution as well as potential  pre-eruptive and eruptive patterns. Nevertheless, we have anyway clarified better this matter, adding more references, in the revised version of the paper.

(5) Remove the text that is peculiarly negative in tone. In some sections, new material is presented with the apparent aim to demonstrate where others are wrong, rather than to show how it can improve current understanding.

Reply: Our aim is solely to address and correct previous interpretations of the pre-eruptive, eruptive and post-eruptive patterns associated with the 1538 eruption, which are demonstrably inconsistent with some fundamental observations. This is not intended as criticism of the researchers themselves, but rather of their scientific interpretations. Such efforts reflect the natural progression of science. Nevertheless, we revised the paper to ensure that our objective — disproving weaker scientific interpretations to enhance the understanding of volcanic phenomena, in line with standard scientific practice — is made even more explicit.

**Specific Comments**

**Abstract**

This will need to be adjusted to accommodate the revised text.

Reply: We checked the abstract, which now reflects well the revised text.

**Introduction**

Lines 32-34. These lines cite previous work on reconstructing historical movements at Campi Flegrei and then state that all their conclusions need to be modified. However, only the work by Parascandola (1943) and Di Vito et al. (2017) is later discussed in detail. Dvorak and Mastrolorenzo (1991) is not mentioned, while work published by Bellucci et al. (2006) is repeated but not cited. The referencing of earlier studies must be more inclusive throughout.

Reply: We have done that and enhanced the referencing throughout the entire paper.

**Eruptive history of Campi Flegrei.**

Lines 68-100. Reduce text by 30% and focus attention on the movement of a central block. The conventional interpretation in the literature is that the caldera formed about 40,000 years ago, during the eruption of the Campanian Ignimbrite (CI) and that the collapse during the Neapolitan Yellow Tuff (NYT) eruption represents a subsidiary movement. To state that the caldera formed only 15,000 years ago may well be correct, but it is not yet generally accepted. The key point here is that evidence of a resurgent block is only available since the NYT eruption, regardless of whether the caldera formed then or during the CI eruption. To acknowledge the uncertainty, perhaps the text could be adjusted to something like:

"Campi Flegrei is an active caldera to the west of Naples in southern Italy. About 12-14 km across, its southern third is submerged beneath the Bay of Pozzuoli. Following the most recent – and perhaps only (Rolandi et al., 2020) - episode of caldera formation, some 70 eruptions have occurred across the caldera floor, ranging from the effusion of lava domes to explosive hydro-magmatic (?) eruptions (Di Vito et al., 1999; Smith et al., 2011; Isaia et al., 2015). The most recent eruption occurred in 1538, producing the cone of Monte Nuovo (Di Vito et al., 2016)…"

For completeness, add citations to previous work on resurgence at Campi Flegrei, including: Luongo et al. (1991), Orsi et al. (1996, 1999) and Acocella (2010).

Reply: We actually made the suggested changes.

**Subsidence and uplift before the 1538 eruption**

Lines 103-368. This key section presents an exhaustive evaluation of historical observations on ground movement at Campi Flegrei. However, it is not clear which observations are new and which are referring to previous studies. To highlight the new material, it would be helpful to have:

1. A paragraph identifing previous interpretations, by bringing forward Fig. 13 a and b (and related text) to provide a starting reference. I would add also the trends proposed by Dvorak & Mastrolorenzo (1991), Bellucci et al. (2006) and Morhange et al. (1999, 2006).

2. A clear statement at the end of the section that specifies the novel results from this study. For example, how new are the trends for the via Herculea (Fig. 3) and Serapeo (Figs 6 & 13c)?

Reply: We thank the reviewer for the useful suggestions and actually incorporated them. In particular, we added a paragraph reporting previous interpretations. However, the observations regarding via Herculea and the trends at Serapeum are almost enterely new, as shown by the marked difference from previous interpretations, and well supported by historical sources.

Lines 109-110. I would avoid claiming that this paper is "correcting misrepresentations or erroneous reconstructions". This is a value judgement and may be viewed by readers as hostile. Why not say something like "the new data allow more detailed (?) reconstructions than have previously been possible and, as a result, provide tighter constraints on the mechanisms driving unrest." Such an approach conveys the positive idea that this paper in *building on* previous work, rather than *correcting* it.

Reply: Ok, we made the changes accordingly. However, science progresses through the gradual correction of incorrect hypotheses, based on new observations. As non-native English speakers it is possible that the formal language we used to express this evident concept may have sounded somewhat unclear. We adjusted our statements in line with the phrasing suggested by the reviewer, and believe now our purely scientific aim is more clear.

Lines 128-171. Only a single source (Parascandola, 1943) has been cited for the descriptions here. If this is correct, then what advances have been made by the current paper? If this is not correct, can the authors identify the sources of their new data?

Reply: Most of the historical sources are actually included in the Appendix, which provides the references for each numbered point shown in figure. However, we agree that including them in the main text as well would improve readability. We made this adjustment. It must be noted that, regarding the Via Herculea, most of the references were already given in Parascandola (1943); the problem is that they have been mostly ignored in the subsequent literature, leading to several misinterpretations.

In Fig. 3, the different trends in ground position shown by the continuous and dashed lines need to explained in the caption; similarly, what do the numbers 1-10 indicate? Which measurements have been published before (*e.g.* by Parascandola (1943)?) and which are new to this paper? If the original data have been moved to Supplementary Material, why not add a table here that gives short descriptions?

Reply: As previously explained, the sources are provided in the Appendix. In the revised version, we included a table, in the main text, with brief descriptions and corresponding references. The meaning of dashed lines has been clearly explained in the caption.

Line 197 (and Line 116). It may help to include a subheading (not necessarily numbered) that the following text has switched observations of ground movement from Via Herculea to the Serapeo. If so, an equivalent subheading for Via Herculea could be inserted around Line 116.

Reply: In the revised version, we have used such suggested subheadings.

Lines 238-265. These lines describe Fig. 7. The text is virtually identical to that in Bellucci et al. (2006, pages 149-50), for which Rolandi is co-author. There is no need to repeat the text at the same level of detail: it can be shortened with reference to Bellucci et al. (2006), which ought to have been cited in the first place.

Reply: Ok, we have done that.

The numbered sources in Fig. 6 need to be identified and not relegated to Supplementary Material. Why not add these to the table already recommended to accompany Fig. 3? What is new about this plot? How does is differ from previous measurements? How are the dates on the right of the graph related to the trends? In particular, nine dates are given from 1430 to 1538, yet only three points are shown defining the uplift between c. 1430 and 1538: why have more dates been shown than the number of points?

Reply: The dates on the right, in black, indicate the occurrence of the main earthquakes. We made an error with the first date, which is listed as 1430, but we corrected to 1448. We have further clarified the caption regarding these dates. Additionally, we include two tables in the main text to provide the sources for the data points shown in figures 3 (as already specified) and 6.

An important omission is the interpretation of ground movement by Bellucci et al. (2006) and which the present authors have used in subsequent publications (*e.g.*, Troise et al. (2007, 2019)). It includes a possible uplift around 700-800 AD, following archaeological evidence presented by Morhange et al. (1999). Although Morhange et al. (2006) later removed this uplift without explanation, no clear evidence has been presented to resolve whether or not the uplift occurred. Any new data in Fig. 6 are thus *especially significant* because they offer an opportunity to clarify this point. Indeed, Fig. 6 shows horizontal arrows to highlight two measurements that happen to show continued

subsidence across the relevant time interval. Is this deliberate or a coincidence? An explanation is necessary here.

Reply: The two arrows mark the limits of submersion, as indicated by the range altered by lithodomes. Historical chronicles clearly show that the submersion progressed during the time interval between the 6th-7th centuries and the 9th-10th centuries.in the 6th-7th centuries, the citizens of Pozzuoli were forced to abandon the lower part of the city, which had been invaded by water. By the 9th-10th centuries, subsidence had become even more pronounced, leading to the formation of Lake Agnano in the Plain of the same name. So, the interpretations of Morhange (1999) are contradicted by historical data. We have been more clear on this points, in the revised version.

Lines 295-312. Please add references to previous studies in the main text: don't hide them in figure captions. Can the authors specify where they have added their "own reworking" and so highlight the advances in this paper?

Reply: In fig.8, only panel A reproduces the original figure by Soricelli (2007). The panel B, redrawn by us, focuses on a specific part of the area shown in A (as already mentioned in the caption), which was partially submerged in 1430 as depicted in panel C. Panel C was reconstructed and drawn by us, with the primary source for this reconstruction being De Jorio (1820), an observation also cited by Dvorak and Mastrolorenzo (1991). We left just the references needed to explain the main sources, in the caption.

Lines 318-328. The authors present evidence why they believe some inferences about ground movement made by Di Vito et al. (2016) need to be revised. Can they suggest which observations led Di Vito et al. (2016) to estimate different values? As for their comments on Claims (2) and (3), it is notable that the trend shown in Fig. 6 from the 12th Century to 1538 supports that presented by Bellucci et al. (2006, Fig. 7), which should be referenced to avoid giving the incorrect impression that all the data are new to this paper.

Reply: No, we were unable to identify, either in the main text of the Di Vito et al. (2016) paper, or in its supplementary material, the observations that led these authors to propose the three mentioned interpretations. Although it is hence not clear, in their paper, the source of those interpretations, we have demonstrated, in a clear way, they are in contrast with robust historical evidence. We cited more specifically the paper by Bellucci et al. (2006), which had been however cited multiple times in our paper including in this specific context.

Lines 336-338. The inference of a maximum subsidence in the 1400s is not new to this paper. Dvorak & Mastrolorenzo (1991) and Bellucci et al. (2006) reached the same conclusions: at best, this paper confirms their results (although the similarity with Bellucci et al. (2006) may also indicate that some results are being repeated, especially since the lead author here was a co-author on the earlier paper).

Reply: Certainly, this paper, in some sections (though not all) supports the findings of Bellucci et al. (2006) and, to some extent,, by Dvorak and Mastrolorenzo (1991). However, in this study we provide additional and more detailed historical references. This approach

aims to eliminate any ambiguity that may have led subsequent authors to propose differing interpretations (e.g. Di Vito et al., 2006).

Lines 341-346. The occurrence of seismicity is out of place. I would move the observation to a general discussion after the sections that describe the seismicity more fully. The new paragraph could then begin directly with "Our findings..." (Line 346).

Reply: We use here the reference to the 1448 only to strength the finding that ground uplift starts after 1430. So, we believe it is better to stress here this point, rather than in the session devoted to seismicity; which further starts several paragraphs later.

Line 358-9. As above, the comment on seismicity can be moved to a later discussion.

Reply: Ok, we have eliminated this comment here.

Lines 366-8. The comment about the "anomalous" pre-eruptive uplift at the site of the Monte Nuovo eruption appears without context. I suggest mentioning this behaviour at the start of the reconstruction and then explain why it can be neglected when investigating ground movement across the caldera.

Reply: The reference to the anomalous uplift at the site of the future eruption is very important and, in our opinion, must be recalled here. In fact, the occurrence of a sharp and very localized uplift, breaking the bell-shaped trend common to all the unrest episodes not culminating with an eruption, could likely be the most clear precursor of a new eruption.

**Ground movements after the 1538 eruption**

Lines 370-428. This section describes subsidence between 1538 and c. 1950. To emphasize the relevance, it may help to compare rates of subsidence with those before 1538 and, perhaps, draw conclusions about whether conditions in the crust before the 1430-1538 uplift might have been similar to those before 1950, hence further supporting the idea that the two sequences may have been driven by similar subsurface processes.

Reply: We thank the reviewer for the suggestion. In the revised version, we have discussed similarity and differences of subsidence rates before and after 1538 eruption, and also compared the uplift rates in the period 1430-1503 with those in the period 1950 to present.

**Schematic model for the preparatory phases of the 1538 eruption**

Lines 448-455. I would adjust the text to argue that the resurgent-block model is *consistent* with observed ground movement and seismicity, but is not the only possible interpretation. Numerous models have accounted for the observations without involving block movement (among others, Berrino et al., 1984; Bianchi et al., 1987; Amoruso et al., 2008, 2014; Woo & Kilburn, 2010). The authors could then explain the significance of whether or not block movement is significant to determining uplift before 138 and since 1950 (bringing forward text from Lines 472-478).

Reply: As previously explained in response to a similar comment by the reviewer we believe that the resurgent block model is well constrained by the ground deformation and seismicity patterns recorded over several decades in recent times, as well as by reported observations from ancient times. Regarding the models that attempt to explain ground deformation and seismicity without invoking subsidiary movement along the ring faults bordering the resurgent block, the models proposed by Berrino et al. (1984) and Bianchi et al. (1987) presented significant drawbacks, as demonstrated since the 1980s (see De Natale et al., 1991). These limitations formed the basis for the first studies suggesting the role of ring fault movements (see De Natale et al., 1993; 1997). The paper by Amoruso et al. (2008; 2014) aimed only to explain the ground deformation patterns, not seismicity, and relied on an 'ad hoc' layered model to simulate the observed shape of ground deformation. The issue with such models is that they are generally designed to explain single ground deformation events, but it is far more challenging to account for the remarkably consistent ground deformation and seismicity patterns observed over decades (or potentially centuries). This difficulty is compounded when considering likely changes in source and/or medium properties over such extended periods. Importantly, the concept of resurgent block model has been widely used since the earliest studies of caldera unrest. In this paper, our aim was simply to refine the identification of the moving block by integrating the most constraining data. However, this is not the primary focus of the paper. We have however clarified this point, and added more references.

Lines 455-471. I would move the description of subsurface structure to the start of the paper, as part of the background context.

Reply: This section has been included here because the involvement of a resurgent block naturally emerges from the description of both secular and recent ground deformation, based on comparisons between historical sources and recent observations. The detailed description of all the observations and data supporting the existence of the resurgent block is logically linked to this purpose. However, we further clarified this part in the revised version.

Lines 475-483. Could the authors clarify the contradiction in proposed substructures? The authors argue that tuff occupies the upper 1.5-2 km of the crust (Lines 474, 482 and 499). How does this contradict the "layer of loose pyroclastics from recent eruptions" (Lines 476-7)? How "loose" will these deposits really be 1.5 km below the surface? Will they behave differently from a tuff in any significant way? Presumably the upper levels **do** contain deposits from recent eruptions too.

Reply: The tuff within the deposits inside the resurgent block is lithoid, with a compression strength of approximately 30-50 kg/cm$^2$, whereas the loose pyroclastics, even when compacted are not lithoid, having a compression strength below 10 kg/cm$^2$). Therefore, they are clearly distinguishable. While it is true that the upper levels contain deposits from more recent eruptions, the upper level of the Neapolitan Yellow Tuff (NYT) is a clear indicator of the block resurgence. We have anyway clarified these points, specifying that the NYT is lithoid, in contrast with loose pyroclastics.

Lines 491-507. The text would be easier to follow by starting with field data that support the presence of a thermometamorphic horizon (*e.g.*, borehole and gravity data (Rosi & Sbrana, 1987)) and fluid filled rock at shallow depth (doesn't this correspond to Campi Flegrei's hydrothermal system? If so, just say this and cite suitable papers) and *then* speculate on its origin. I don't see that the discussion on Lines 491-498 adds anything to the argument. It could be omitted, perhaps moving the citation to Vinciguerra et al. (2006) elsewhere.

Reply: This section aims to clarify that, with the 3-4 km depth range, there exists a thermo-metamorphic low permeability layer, that confines the upper, high permeability zone. This zone can accommodate gas overpressure from below, which occasionally ruptures when pressures become too high, causing the injection of supercritical gas into the upper layers. The tests conducted by Vinciguerra et al. (2006) are particularly significant as they suggest that earthquakes are likely confined to the lithoid tuff zone. Nonetheless, we clarified this point further in the revised version.

Lines 519-522. Why a mush? The authors have argued that renewed uplift may have occurred during c. 1400-1538 and since 1950. If driven by magma intrusions, then presumably the maximum volume of magma intruded corresponds to the amounts associated with these uplifts. Would not the amount involved have fully solidified by now?

For example, the slowest rate of heat loss is expected to be by conduction. The thickness of a solidified layer is on the order of $(4kt)^{1/2}$, where the thermal diffusivity k is ~ 4 x $10^{-7}$ $m^2$ $s^{-1}$, and time t is in seconds. For a single sheet-like body, the thickness will thus be about 7 m in 1 year, 22 m in 10 years, 35 m in 100 years and 80 m in 500 years. Since cooling occurs across the upper and lower boundaries, the total thickness solidified will be about twice these values. If the surface area of the sheet is ~10 $km^2$ (*e.g.*, a circular sheet c. 2 km in radius), then the corresponding minimum volumes of magma involved are 0.15, 0.45, 0.7 and 1.6 $km^3$. In others words, an intruded volume of 1.6 $km^3$ before 1538 may have completely solidified before the return to uplift in 1950. At the other extreme, an intrusion of as much as 0.6 $km^3$ in 1984 would have solidified before the return to uplift in 2004.

Although these calculations are approximate, they suggest that it may be difficult to preserve a mush layer at depths of 2-4 km over the required time intervals; indeed, the authors suggest as much when discussing the uplift since 1950 in Lines 594-631. If this is correct, can the authors address the apparent contradiction between mush and full solidification and, if necessary, propose an alternative scenario that does not depend on the existence of mush at such depths?

Reply: According to Bachmann and Bergantz (2006), as cited in our paper, crystallized magma (which could equivalently be referred to as hot crystalline mush, since solidification is a continuous process), can be mobilized by the inflow of exsolved fluids coming from mafic magmas, a process known as 'gas sparging'. Therefore, the mechanisms described are sufficiently general to be independent of the degree of cooling of the original magma. However, we further clarified this point to make it easier for readers to understand.

Lines 529-534. Subsidence has been occurring between 3,700 years ago and 1538. Can the authors offer some independent checks for consistency? For example, if the rate has been similar to that in historical time (1.5-2 m per century), then ground level 3,700 years ago must have been some 50-60 m higher than in 1538. Is this consistent with the reconstruction of prior resurgence?

Reply: There is no data available to constrain the rate of ground subsidence before the 2$^{nd}$ century AD. However, the rate of subsidence since III century AD to 1430 has been of 1 m per century; if it started 3700 years ago with the same rate, then the ground level 3700 years ago would have been about 20 m higher than in the III century AD, and 14 m higher than in the 1538. We do not see, however, the relevance of this observation.

Lines 537-546. Is the level of detail necessary here? None of the specified processes can be demonstrated to have operated at Campi Flegrei. I would simply postulate that a mush zone exists – perhaps referencing more direct evidence from the seismic-tomographic analysis by Zollo et al. (2008), who argued that the crust at these levels contain dispersed patches of molten rock. Additional papers can be left as citations.

Reply: We have simplified the discussion, but anyway we believe a more detailed explaination of these concepts is important; even because the possibility of mush remobilization is generally lacking in the scientific discussion about a possible impending eruption at Campi Flegrei.

Lines 546-551. I found the text confusing. Do the authors mean that the rapid uplift between 1430 and 1538 could have been caused by the release of gas? If so, why not just state this in a single sentence?

Reply: The uplift between 1430 and-1538 was likely caused by a combination of gas inflow and magma inflow. The mechanism described here can support both processes. We have further clarified this concept.

It may be of interest that Mormone et al. (2011) have presented evidence from melt inclusions that for deeper magma entering the "8-km" reservoir (mush zone?).

Reply: Yes, this suggests new magma inflow into the 8 km reservoir, which could be the mechanism driving the present unrest through an accelerated degassing of the magma chamber.

Lines 552-583. The description jumps back and forth between different levels in the crust. I would restructure and shorten the text – perhaps following progress upwards from the "8-km" reservoir.

Reply: Ok, we have been clearer and concise in the revised version.

Lines 552-554 could form part of a revised introduction to the Campi Flegrei's subsurface structure. Links to Yellowstone can be removed, or moved to a later discussion that compares *in a single section* the authors' model of Campi Flegrei with other volcanoes.

Reply: these few lines are strictly functional to explain the hypothesized mechanism leading to uplift and seismicity even without direct magma intrusion. So, we have clarified better this part, but not eliminated.

Lines 557-561. Can the authors connect their description of saturated rock to the hydrothermal system that (according to the literature) exists at similar depths?

Reply: Yes, we are obviously inside the hydrothermal system, here. We clarified that in the new version.

Lines 567-8. These lines need to be adjusted. Kilburn et al. (2023) proposed common depths for magma intrusion and gas accumulation *below* the thermo-metamorphic horizon at depths of c. 3 km (and not beneath a shallower cap-rock). I would also suggest changing how the comment is presented. For example: "We consider that magmatic gases may not necessarily be restricted to below the thermo-metamorphic horizon (Kilburn et al., 2023), but may instead accumulate at shallower levels beneath the "summit" lava at a depth of c. 2.5 km." In general, I would avoid statements that previous work is "wrong" or "incorrect". After all, the authors have **not** demonstrated this: they have instead proposed an alternative view for consideration (which may or may not turn out to be an improvement).

Reply: Ok, we made this change in the revised version.

Another constraint to consider is whether the proposed depth of 2.5 km for gas (and pressure?) accumulation is consistent with depths closer to 3-3.5 km estimated by geodetic modelling of uplifts in 1982-84 and since 2004 (*e.g.*, Amoruso et al., 2008, 2014; Amoruso & Crescentini, 2022).

Reply: These depths are probably not distinguishable within the errors due to lack of detailed knowledge about the properties of the medium and the secondary movements of the ring faults. However, most of the papers cited in this study suggest a source depth within the range of 2.5-3.0 km.

Line 567. The term "summit" lava is distracting. Why not refer to the "lava level at 1.5 km" (or something similar) to connect with the lava mentioned in Line 560-1. Moreover, could this unit be an intrusion rather than a "lava", which implies that it flowed over the surface?

Reply: Yes, we have changed the term, using 'magma'.

Lines 572-645. The ideas described are interesting, but have jumped to processes operating today, rather than before 1538, which is the title of the subsection. I suggest reorganising the text to appear as a separate section focussing on processes operating today, followed

a separate discussion comparing the pre-1538 and modern behaviour (see potential connection to Lines 838-856 below).

Reply: Ok, we followed part of these suggestions and have clarified this part by specifying better the comparison between old and new observations. It is only functional, here, to support the hypotheses on the mechanisms involved in the unrest leading to 1538 eruption.

Lines 594-635. The authors describe the intrusion of magmatic sills (presumably at the depths of 2.5-4 km (Line 602)?) and solidification of these sills within 20 years (Line 610). This conclusion is based on previous work. It appears to contradict Lines 546-551, which argue that gas transfer is the mechanism driving unrest. The connection between gas release and magmatic intrusion needs to be clarified – and succinctly: to be honest, I think the description could easily be reduced by 40% (which would force the authors to focus on essential features). Fig. 16 is attractive, but essentially shows collections of arrows indicating an upward transfer of pressure. Which arrows refer to gas transfer and which to magma movement? I would redraw simplified figures that specify the features described in the text. There is no need here to rely on speculative generic models, such as that proposed by Bachmann & Bergantz (2006): the figure can be based solely on the evidence presented in this paper.

Reply: As previously mentioned, we believe the uplift before 1538 was caused by both gas and magma intrusion. Therefore, there is not contradiction with our earlier statements. We made however these parts clearer and more concise.

Line 585. See comment on Lines 567-8: Kilburn et al. (2023) placed the zone of pressurization *below* the thermo-metamorphic horizon.

Reply: Ok, we corrected accordingly.

Lines 587-593 can be omitted*. All they do is introduce what is about to written next. Go straight to the next paragraph. (*Or perhaps a shorter version can be moved to the conclusions?).

Reply: Ok, we cancelled this part.

**The eruption of 1538**

Lines 647-719. Why is this section included? The eruption of Monte Nuovo doesn't contribute to the rest of the paper? It is really a review of previous work and does not present new information relevant to the main arguments of this paper. (N.B. In Fig. 19, Wohletz's figure of magma-water interaction and eruptive explosivity has previously been applied to Campi Flegrei. I can't remember the authors or journal, but believe it was published in the late 1980s or 1990s.).

Reply: As mentioned earlier in response to a previous question, we want to emphasize that the account of Mt. Nuovo eruption directly contributes to the goals and conclusions of this

paper, which aims to accurately describe the events before, during and after the 1538 eruption, in order to provide insights intowhat could occur in the future. In fact, we highlight in this paragraph some observations that have never been reported in previous literature, which are also important for assessing what might happen during a future eruption.

**The seismicity before and after the 1538 eruption**

Lines 731-822. The connection between earthquake magnitude and intensity is extremely interesting. Again, though, I think too much detail has been added. It breaks the structure of the paper and loses sight of the key objectives. Figures 18 and 19 are the key results and the text could be shortened so that they become more prominent. Thus I would (1) move the methodology to supplementary material; and (2) tabulate the sequence of events, because they have already been described comprehensively by Guidoboni & Ciuccarelli (2011). I expect the whole section could be reduced to two figures, one table and half a page of text.

Reply: Ok, we followed the suggestion and moved the description of the intensity to magnitude conversion in the supplementary material; we also shortened the text.

**Post-eruption seismicity**

Lines 825-835. How relevant is this section? It hasn't clearly been connected to understanding the processes operating beneath Campi Flegrei.

Reply: This short section is important because it shows that seismicity does not necessarily cease after an eruption and can continue for several decades. This constitutes a crucial scenario for understanding the possible evolution of the present unrest, with clear implications for civil protection.

**Comparison of precursory phase of the 1538 eruption with current unrest**

Lines 838-856. If the authors could present a graph showing the seismic behaviour during unrest since 1950, this paragraph could then be moved to the end of the revised description of pre-1538 seismicity (Figs 18 & 19).

Reply: This paragraph compares all the precursors of the 1538 eruption with the present anomalies; hence, it includes considerations on both ground deformation and seismicity. So, it cannot be moved in the paragraph describing the pre-1538 seismicicty.

Lines 858-918. The alternative scenarios are a good idea, but not clearly connected with the picture of gas and magma transport the authors have been developing. This is a missed opportunity.

Reply: we made the connection with the developed hypotheses clearer, but the goal is to highlight the alternative scenarios. We believe this goal is clear and actually achieved.

The discussion of future vent location and final rapid ascent of magma (Lines 871-894) is interesting but does not naturally follow from the previous text. It thus seems arbitrary and can be removed.

Reply: We do not agree with the reviewer on this point. We clarified further, but the key point is that the most likely location for a future vent is along the ring faults surrounding the resurgent block. Among these, the part of the ring fault exhibiting the maximum shear stress, i.e. the Solfatara-Agnano area. This area also has important implications for civil protection. Such considerations are clearly linked to the discussion about the resurgent block, in turn also linked to the shape of ground deformation and seismicity.

The same holds for Lines 897-918. This section passes comment on previous work – especially Kilburn et al. (2023) – without evidence or serious analysis. If the authors wish to propose alternative views, they should do so in a separate paper that presents an in-depth analysis; in any case, these views have nothing to do with the present paper and are not supported by the data in the preceding text. [For the record, the statements in Lines 904-914 are simply wrong: (1) the seismic time series *did* change in 2015-17 and, as we know now, evolved towards a full seismic crisis; (2) the *corrected* uplift in 2015 *had* reached its 1984 level; and (3) major rupturing *did* begin in August-September 2023.] .

Reply: We understand the issue, as the reviewer is also the author of the paper. However, what is stated here is almost self-evident and supported by the evidence: there has been no eruption nearly 0 years after the supposed start of the 'critical phase' (which in other volcanoes, as demonstrated by the author in several papers, typically anticipates an eruption within a very short time). The reason we referenced the 2023 paper is that his model of critical stress is very important, and we could not cite the 2017 paper without also mentioning the 2023 follow-up. To address the points raised:

1) Seismicity at Campi Flegrei has increased since 2006, both in magnitude and in frequency, following the stress increase model (though this model does not account for the strong dependence of seismicity on the rate of deformation, which is very evident at Campi Flegrei). However, an eruption did not occur, even 10 years after the supposed entry into the 'inelastic state', whereas, for instance, at Rabaul the same authors found the eruption occurred only 2-3 years after entering the inelastic state, and after much less time at the other studied volcanoes (Robertson, R. M. & Kilburn, C. R. J. Deformation regime and long-term precursors to eruption at large calderas: Rabaul, Papua New Guinea. *Earth Planet. Sci. Lett,.* **438**, 86–94 (2016)).

2) Correcting the uplift compensating it for the average secular subsidence rate implies that secular, slow subsidence is caused by the same source mechanism of the rapid uplift during unrest periods. However, the mechanism behind the present unrest is likely gas inflow, or possibly magma; secular subsidence, whose mechanism remains unclear, should be due to different factors, like the progressive cooling and volume reduction of the main magma chamber.

3) The same considerations as in point 1.

However, we do not believe that quoting the 2023 paper and its likelihood is essential for the goal of this study. Therefore, we removed this part.

It would be more interesting to see how the revised analysis of events before 1538 might better constrain what is happening today. As I understand it, the authors' pre-1538 reconstruction suggests early uplift driven by gas transfer, followed by magmatic intrusion and eruption (comparison with the model of Bodnar et al. (2007) and Lima et al. (2009) may be valuable here). If this is correct, can the authors identify comparable stages today – or is the modern sequence significantly different in some way? Given the large amount of work in preparing the reconstruction, it seems a pity not to take full advantage of the new results.

Reply: We thank reviewer for the suggestion, and we have done that in the revised version.

**Figures**

Most of the figures are relevant and of high quality.

Figures 3 & 6. The time axis is labelled as "Centuries". This can cause confusion, because the century may be taken to be the preceding hundred years (for instance, in standard English usage, the fourth century refers to 1300-1399). I would replace the times with actual years (as done in Fig. 13).

Fig. 15. Consider changing the colours to make the shapes outlining zones clearer on the map (especially the red of the caldera outline).

Please check language. Some terms remain in Italian: *e.g.*, in Figs 3, 6 and 13, time "AC" should be "AD" (use either the combinations BC and AD, or BCE and CE ("Before Common Era" and "Common Era"), and in Fig. 13 "duttile" should be "ductile".

Reply: we made all the required corrections to the figures

**Language**

The English is generally good, but contains grammatical oddities that could be spotted with help from a native English speaker. I would avoid phrases that describe interpretations as "likely". Their use may be mistaken as attempts to support uncertain interpretations without sufficient evidence. They are not necessary here.

Reply: we checked the English form, which should be now acceptable.

**References**

Please check that all references are in alphabetical order (*e.g.*, Lines 1013-1024 need to to be brought forward), and also some of the dates that have a number missing (*e.g.*, Line 1010). Also, check that citations in text are presented in *chronological* order.

Reply: we added almost all the suggested references and checked the order of them.

**Recommended Additional References**

Acocella (2010) Bull Volcanol 72:623–638.

Acocella (2019) Front Earth Sci 7:173, doi: 10.3389/feart.2019.00173

Amoruso et al. (2008) Earth Planet Sci Lett 272:181–188.

Amoruso et al. (2014) J Geophy. Res 119:858–879.

Amoruso & Crescentini (2022) Remote Sens 14:5698. doi.org/10.3390/rs14225698.

Berrino et al. (1984) Bull Volcanol 47:187–200.

Bianchi et al. (1987) J Geophys Res 92:14,139–14,150, doi:10.1029/JB092iB13p14139.

Isaia et al. (2015) Geol Soc Am Bull,  doi:10.1130/B31183.1.

Luongo et al. (1991) J Volcanol Geotherm Res 45:161-172.

Morhange et al. (1999) Phys Chem Earth 24:349-354.

Morhange et al. (2006) Geology 34:93-96.

Mormone et al. (2011) Chem Geol 287:66–80.

Orsi et al. (1996) J Volcanol Geotherm Res 74:179-214.

Orsi et al. (1999) J Volcanol Geotherm Res 91:415–451.

Troise et al. (2007) Geophys Res Lett 34, L03301, doi:10.1029/2006GL028545.

Zollo et al. (2008) Geophys Res Lett 35:L12306, doi:10.1029/2008GL034242.

**Citation**: https://doi.org/10.5194/egusphere-2024-2035-RC1

**Response to second reviewer**

This is a very interesting paper, in which historical data are thoroughly revised to propose a revisited chronology of the ground (vertical) displacement over a long period time preceding  the Mte Nuovo eruption (1538 CE) at Campi Flegrei (CF). I appreciate the historical background on which the study is founded, particularly the analysis of the relative position of the Roman artifact in Via Herculaea, which puts constarints to the maximum  uplift.

First of all, the new reconstruction is interesting because it shows a steeper but rather continuous ground uplift prior to eruption. This steep uplift actually looks like the one observed in recent times, during the 1982-84 unrest episode. Here is my first comment: this similarity is not discussed and perhaps it should be in the double-scenario section. Worth noting is the long duration (abot 100 years prior to eruption) of such a steep uplift, with implications also for uplift rates observed during the current crisis and the 1982-84 one.

Reply: We thank the reviewer for the suggestion. We agree and so addressed this point more thoroughly in the revised version of the paper.

Secondly, the new reconstruction of the pre-1538 uplift and seismicity highlights the two and half scenarios that should be expected. I say "... and half" because the phreatic eruption is considered somehow apart in case the system does not reach the threshold for a magmatic eruption. This is correct and it should be a scenario "per se", which should be more emphasized by the Authors. In my view we do not know of many phreatic events at CF simply because they were soon erased by the following magmatic-explosive activity. However, phreatic eruptions must have occurred and likely few of them have represented some kind of peak activity for longtime, before the culmination in an magmatic-explosive eruption. The phreatic scenario deserves a lot of attention for the current evolution of the CF unrest.

Reply: We appreciate the suggestion and agree. In the revised version we discussed the possibility of a phreatic eruption in greater detail.

Thirdly, the new reconstruction offers a semi-quantitative criterium to establish the evolution of seismicity as uplift continues: the M 5 event from historical sources is put into the evolutionary context of the 1538 CE eruption. Also worth noting, although from a qualitative point of view, the role of thermal cracking which accompanies the steep uplift.

Although the Vinciguerra et al paper is a central one, note that thermal cracking was also invoked in previous papers, particularly Chiodini et al. (2015, EPSL) and Moretti et al. (2018, SciRep), in which thermal cracking was approached from different perspectives. Also past papers from the same Authors of this study mentioned it.

Reply: We agree, and included these references along with additional ones relevant to this point.

Of particular interest is also the role attributed to a resurgent block (see also below), which is the same as the one nowadays active, and which in literature was already described. I wonder whether the resurgent block which the Authors refer to corresponds to that in Capuano et al. (2013) : just see their figure 6 and the text part in which its descriptions starts by saying : "We refer to the South-western deeper sector as the 'undeformed-to-subsiding portion of the Pozzuoli Bay," and to the North-eastern shallower sector as the "resurgent portion of the caldera' (Figure 6)". The descriptions which follows of the resurgent block refines previous findings by Orsi et al. on  JVGR (1999; vol. 91, pp 415-451).

Reply: We agree and included more references in the revised version, also emphasizing the similarities between our hypothesis and previous ones. However, our definition of the resurgent block, is somewhat different to the previous hypotheses and, more importantly, better constrained by geological evidence, earthquake locations and active seismic soundings.

Let me remark that in this paper the cause of unrest at CF - also the current one, if I uderstood well - is attributed to some hybrid mechanism of heat and fluid release from a magma batch ascending from the deep regional body (Zollo et l., 2008 GRL) and percolating through the caldera bottom. How significant is this view for the advancement of the current scientific debate around the causes of unrest at CF ?

Reply: the source of present unrest is the key point for its scientific interpretation and for civil protection purposes. For the 1430-1538 pre-eruptive unrest, we propose a hybrid mechanism involving both the inflow of hot magmatic fluids into the shallow hydrothermal system, and episodes of direct magma injection at shallow depths. A key observation is that, after the uplift of 16 m., only about half was recovered through subsequent subsidence. This suggests that approximately 8 m. of non-recovered uplift can be attributed to direct magma intrusion. This is strikingly similar to the 1982-1984 unrest, where only half of the total uplift (1.8 m in that case) was recovered after 21 years of ground subsidence. Another notable observation is that, despite such a significant uplift, implying the intrusion of some km³ of magma at shallow depths, only 0.02 km$^3$ were erupted:. This suggests that most of the intruded magma had already crystallized into mush and was no longer fluid by the time of the eruption. We have further analyzed and discussed these arguments in the revised version, as they are central to understanding the problem.

Finally, I appreciated the reconstruction of the low-energy pyroclastic flow that reached the Pozzuoli temple (Serapeum). It is in line wth the rationale of the study and its title, but it seems a bit disconnected from the main text flow. This is just about text organization and perhaps the Authors could harmonize the whole thing by recalling in the Conclusions that the spectrum of phenomena which result from this study and apply to the eventual evolution of the present-day unrest include 1) increasing seismic activity and M 5 events , 2) phreatic eruption and 3) pyroclastic flows reaching Pozzuoli in case of an eruption like the 1538 one.

Reply: We thank the reviewer for the suggestion, and we addressed this in the revised version.

The paper requires an attentive English revision and also more attention to previous work, which should be properly cited. I understand that there is a bunch of literature, but this can be improved.

Some minor points:

- With ref. to Troise et al. 2019, at line 863 the Authors say "The same authors calculated that such a sill intrusion should have solidified, in form of mush, after about 20 years, i.e.

around 2003. Closely rleated are the statements also reported at lines 609-10 and 518-519. Actually, rapid solidification of a thin sill like mahgma in the order of 10 to 20 years was put in evidence by Woo and Kilburn (2010 JGR), Moretti et (2013 EPSL) and it was modelled by Moretti, Troise et al. (2018 Sci Rep).

Reply: Yes, we included the more recent publications along with their references. However, we agree that it is more appropriate to cite all the previous works on this matter, and we included all such citations in the revised version.

- At line 849 the M 4.2 event on October 2023 is reported as the maximum one recorded at the time of writing, but somewhere in the Introductory part the Authors refer to a M 4.4 occurred in 2024
**Citation**: https://doi.org/10.5194/egusphere-2024-2035-RC2

Reply: Yes, at the time of the first draft, the M=4.4 event had not yet occurred. We added it in the revised version.

---

## Referee Report (RR1)

Review of Rolandi et al., The 1538 eruption at Campi Flegrei resurgent caldera: implications for future unrest and eruptive scenarios.

**Summary**

The revised version of the manuscript naturally divides into two parts. One part successfully highlights the meticulous reconstruction of ground movement and seismicity before Campi Flegrei's only historical eruption. The results provide an important new reference for constraining interpretations of the volcano's current unrest and deserve to be published after modest changes to the English.

The other part is more speculative and draws conclusions beyond those possible from the new reconstructions. I recommend that this part is severely edited to avoid distracting from the merits of the new data; it can then be recast as offering interpretations to be tested, rather than affirmations. As in its earlier version, the text keeps slipping into a negative approach by insisting that the ideas of others are wrong. This is not objective. A positive approach can be achieved by focussing on the merits of the authors' new reconstruction, regardless of the alternatives. I have made copious comments on the manuscript, mainly to illustrate how subediting might enhance the flow of the text. The streamlined version should be ready for publication.

**Specific Comments**

Please see the annotated manuscript for additional recommendations on editing the text.

Section 2. Caldera formation.

This section on the history of Campi Flegrei is more detailed than necessary. The information is fine, but doesn't follow naturally from the introduction. The main point seems to argue that prehistoric ground movements are consistent with the displacement of a central block – which is later used to interpret historic unrest. If that is correct, I would shorten this section and start around Line 78 with something like "Ground movement since caldera collapse is consistent with the centre of the caldera behaving as a single block (REFS)". The description of magma chemistry isn't obviously relevant here.

Section 3. Reconstructing ground movement before the eruption in 1538.

Lines 142-154. What type of new evidence did later studies use to modify Parascandola's 1947 reconstruction (e.g., information from additional contemporary accounts). Specific details do not need to be described: citations to papers will suffice. The authors could then note that (1) the later reconstructions were still based on partial data sets and (2) the new work uses a more comprehensive data set (and so provides a test of previous interpretations). Stating this here will simplify the later discussion of Fig. 13 and avoid repetition when comparing new and old reconstructions.

Lines 169-177. These repeat previous text. I'd consider omitting this paragraph and starting at Line 178.

Lines 491-492 (Figure 13). Fig. 13 shows only three of the five reconstructions mentioned in the text. To highlight how the new work clarifies previous ambiguities, please add the reconstructions by Dvorak & Mastrolorenzo (1991) and Bellucci et al. (2006).

Section 4*. Schematic model for the preparatory phases of the 1538 eruption.
[*Check formatting. The numbering of sections has been set back to "2".]

Lines 504-594. This section makes the case for movement along faults to be a major influence on observed patterns of ground deformation (as had previously been proposed by some of the authors). However, it loses focus by intermittently mentioning that alternative models are wrong. The assertion has not been justified. It would require a full account of the alternative models and their assumptions. I would simplify the section by concentrating on the evidence for block movement. The commentary on alternative interpretations can be omitted. This would make the text easier to follow and also allow the authors to highlight that their reconstruction demonstrates that fault-bounded movement is a realistic interpretation.

Lines 545-548. These lines can be omitted. The authors can support their interpretation, but they have NOT shown alternative views to be incorrect. That needs a separate paper in its own right. I would simply concentrate on the authors' reconstruction and their description of (and terminology for) the stratigraphy. The discussion of terms is a distraction about terminology, in that deformation models are distinguished by the values of physical properties used, such as elastic modulus, and not by their qualitative description. Moreover, the later assertions that the lithoid tuff is heavily fractured calls into question the relevance of the distinction being made here.

Lines 561-574. Try omitting these lines. I don't see they add anything new to what has previously been written. The previous and following paragraphs would then be linked through the references to Battaglia et al. (2008).

Lines 589-591. References to mush are out of place here. The rest of the section describes observations. No mush has been observed and its presence is speculative. I would omit these lines and leave speculations about mush to the final discussion.

Section 5.2. The preparatory phases of the 1538 eruption.
This section would be better placed after the reconstruction of pre-eruptive seismicity and will be discussed later.

Section 5.3. The eruption of Monte Nuovo.
It's not clear why this section has been included. I can't help feeling it belongs in another paper. The account of the eruption is interesting but, as far as I can tell, does not add to the information already available in the published literature. Unless the authors have a pressing need to keep the account, I would consider removing it, so that this paper can focus on the novelty of the new reconstructions before the eruption.

Section 6 (?) The seismicity before and after the 1538 eruption
(Please check numbering of sections; it appears as "3" at the moment)
This section nicely compares the seismicity in the century before the 1538 eruption with events recorded during the current unrest. Transforming the size of historic events from intensities to magnitudes is a neat way to compare with modern methods for characterising the size of an earthquake.

The classification into long-, medium- and short-term sequences is instructive and relevant to understanding current unrest. However, I would reorganise the text so that the characteristics of each sequence is presented before offering an overall interpretation. Thus, I would group together Lines 827-830, 835-844 and 861-865 and integrate them into the final discussion after Section 6.2.

Section 6.2. The post-eruption seismicity.
Lines 869-872 can be omitted. Start with something like "Post-eruption seismicity was recorded in…". I suggest combining this as a final paragraph to the previous section, rather than keeping it as a standalone section.

Sections 7 and 5.2
[I'm assuming Section 7 starts on Line 880]
These sections repeat themselves and could easily be combined into an interpretation of events preceding the 1538 eruption. For example, Lines 885-901 could be followed by text connecting the reconstructed ground uplift and seismicity before 1538 to a following summary of Lines 827-830, 835-844 and 861-86. This will identify water, gas and magma as favoured sources of overpressure at depth. The role of gas and water can summarised by combining Lines 622-626 and 637-646 (from Section 5.2); the role of magma can the be described succinctly in terms of ascent from a main reservoir to form shallow intrusions. The descriptions can then lead to the two scenarios (Sections 7.1 and 7.2).

The interpretations in Lines 589-621 and 626-677 are speculative at the level of detail presented. They may very well be reasonable, but the supporting evidence is superficial and so the arguments lack conviction (especially when compared with the painstaking reconstruction of behaviour before 1538): in particular, the insistence that small shallow sills can consist of magmatic mush after more than a few years is not fully justified. For example, sills intruded at depths of c. 3 km are shallow enough for their mean thicknesses to be similar to the amounts of surface uplift they produce – namely a few metres. Even under the conditions of slowest cooling by conduction, such bodies are expected to have solidified completely within years (remember the magma has only to cool below its solidus to be completely solid). For such conditions, the assumption that magma remains as mush that can be remobilised is not very strong. I thus strongly recommend the authors reduce this text by about 50-70% - or even remove it altogether. Just as for the description of the 1538 eruption, it feels as though it belongs to a separate paper.

Incidentally, in Lines 700-705, the notion of repeated intrusions of small bodies, rather than the growth of a single shallow source has been applied by several authors to the unrest since 1950

(Woo & Kilburn, (2010) and other references): applying it also to before 1538 shows how comparisons between 1430-1538 and 1950-Present may be valuable in both directions, and not only from 1430-1538 to 1950-Present.

Sections 7.1 & 7.2. Scenarios
Lines 913-920. See comments above about "mush".

Lines 928-951. None of this text follows from the results of the current study. The arguments are generic and really need to be developed further to be convincing. They are not essential to justifying the importance of the new reconstructions. Omitting them would produce a better focussed paper.

Lines 961-967. The description of the results in Kilburn et al. (2023) is misleading. As it happens, significant seismicity resumed at Campi Flegrei in 2017 as had been expected. There is no basis for the statement in Line 967 that "the system would already have collapsed". I suggest removing the whole comment.

Lines 968-973. The logic of the argument and its implicit assumptions need to be more clearly articulated. For instance, the authors are assuming that the crust was equally relaxed in 1430 and 1950. Maybe it was; maybe it wasn't. The assumption, though, must be made explicit. I also don't follow the logic that "conditions [are] too gradual to culminate in an actual eruption" (Line 971). Supporting evidence is essential here given that the statement is used to suggest that unrest may continue for another century or more. [Have the authors anticipated the notion of viscoelastic behaviour, which only appears in the conclusions?]

Conclusions
[Please check numbering of section headings]
Lines 997-1012. The text contains additional information about the scenarios. This should be moved to the earlier sections which introduce the scenarios. Viscoelastic behaviour has not previously been mentioned and ought not to appear for the first time in the conclusions. The conclusions could thus be shortened to Lines 979-996, followed by Lines 1016-1018, adding to the list (1) that the outcome of the current unrest is uncertain and two scenarios can be identified, and (2) that, in the case of an eruption, post eruptive seismicity may continue to present a significant hazard (from Section 6.2).

[revised manuscript text omitted]

. This new unrest has been accompanied by progressively increasing seismicity, which has
*[handwritten: Kilburn et al, 2023]*

substantially intensified, both in frequency and maximum magnitude (Troise et al., 2019; Tervolino
*[handwritten: The increase in seismicity began when]*

et al., 2024). The maximum magnitude reached M=4.4 on May 20, 2024,  the maximum ground level attained at the end of 1984 was reached (in July 2022) and surpassed. The progressively increasing seismicity confirms the predictions of Kilburn et al. (2017 *[handwritten: 2023]*) and Troise et al. (2019), who based their forecast on the correspondence of the ground level with stress levels at depth. This seismic activity represents a significant and continuous hazard for the edifices in such a densely populated area, given the very shallow depth of the earthquakes (about 2-3 km). Furthermore, the current crisis poses an even higher threat as it could potentially be a precursor to a future eruption in the area.

The present study is aimed to reconstruct and interpret the events  the 1538 eruption.
*[handwritten margin: Yes — but to aid interpretation of what his happening today]*

This analysis follows three main paths: i) the accurate reconstruction, of the ground movements in this area since early historical times, using historical testimonies and documentation; ii) the accurate reconstruction of the uplift movements that evolved from 1430 to 1538, accompanied and followed by significant seismic events; iii) the analysis of stratigraphic and geophysical parameters, which, although collected in the recent era, provide important elements for the reconstruction and interpretation of the unrest related to the 1538 eruption.

*[handwritten bottom: Please adjust – either specify largest magnitude until (say) the end of 2024 — or change to 4.6 in  March 2025.]*

Finally, the interpretation of the events preceding, accompanying and following the 1538 eruption is used to provide insight into possible evolution scenarios for the present unrest,  (Troise et al., 2019; Scarpa et al., 2022)

**2. Caldera formation and post-caldera volcanic activity 14 ka - 3.7 ka**

Campi Flegrei is an active caldera to the west of Naples in southern Italy. About 12-14 km across, its southern third is submerged beneath the Bay of Pozzuoli. Following the most recent, and likely only (Rolandi et al., 2020a; 2020b; De Natale et al., 2016), episode of caldera formation, i.e. the Neapolitan Yellow Tuff eruption 15 ka, some 70 eruptions (linked to 35 visible vents) have occurred across the caldera floor, ranging from the effusion of lava domes to explosive hydro-magmatic eruptions (Di Vito et al., 1999; Smith et al., 2011; Isaia et al., 2015). The most recent eruption occurred in 1538, producing the cone of Monte Nuovo (Di Vito et al., 1987; 2016). The caldera collapse resulted in many new fractures, which gradually became eruptive vents. Through these vents, the eruptions continued, exhibiting the characteristics of a volcanic field (Druitt and Sparks, 1984), resulting in the so-called post-caldera activity. Dome-shaped uplift of NYT occurred after the caldera formation in the central zone of Campi Flegrei, with uplift up to hundreds of meters on the caldera floor (Rolandi et al., 2020b). The significant uplift involved a large intra-calderic NYT block, making Campi Flegrei a typical example of resurgent caldera (Luongo et al., 1991; Orsi et al., 1996; 1999; Acocella (2010); Rolandi et al., 2020b). The post-caldera activity gave rise to numerous craters, predominantly tuff cones and tuff rings (Fig. 1a,b), displaying the typical characters of monogenic volcanoes (Marti et al., 2016). Within Campi Flegrei, 35 small eruptive centers have been identified, since the NYT eruption (Di Vito et al., 1999; Smith et al., 2012), producing about 70 eruptions. The magmas associated with these eruptions are typically trachytes and alkali trachytes, with smaller amounts of latite and phonolite (Di Girolamo et al., 1984; Rosi and Sbrana, 1987; D'Antonio et al., 1999). The post-caldera eruptions can be then classified in two periods, occurring between 14 ka and 8.2 ka BP and 5.8 and 3.7 ka BP., respectively, with an interval of significant subsidence without eruptions from 8.2 to 5.8 ka BP (Rolandi et al., 2020b).

[Figure]

This seems out of place.
It does not follow naturally from the introduction.
What do we need all the details that are not used later on?
Go STRAIGHT to point that the distribution and timing of eruptions since the Neapolitan Yellow Tuff are consistent with a resurgent block; THIS is the key point for later discussion.

Border of the NYT caldera (after Sacchi et. al., 2014; Steinmann et al., 2018)

Limit of the NYT resurgent structure (after Sacchi et. al., 2014; Steinmann et al., 2016)

Edge of the La Starza marine erosional terrace (subaerial morphology)

Edge of the Pozzuoli marine depositional terrace (offshore morphology, IPW in Sacchi et. al., 2014)

[revised manuscript text omitted]

*Handwritten top: from the 8th Century BC until 1538*

*Our* reconstruction of the level of Via Herculea,  *is* shown in Fig 2  *and 3.*

*A*t the end of the 1st century BC and  4th century AD, works were carried out to increase its height *of the route* above sea level

*As a result,*  the submersion of the  *route* was delayed from  *about* the  *3rd* century BC  *until* the 7th   *← 3rd or 4th?*

century AD (Fig. 3).  *A* date of submersion around 6-7th century *AD* is  consistent with the observations  *by*  Parascandola (1943) that the  Via Herculea  *was*

above sea level for much of the 6th century. *Since sinking below sea level, the*

*Once The* that Via Herculea  *has remained submerged ever since (even* before and during the  *eruption* 1538 (Parascandola, 1943)*) and relicts can be seen today*

about 4.5 meters bsl, as (Fig.4)  Somma et al., (2016).

[Figure]

**Fig. 3 – Diagram showing the trend of ground movements at the Via Herculea, as referred to sea level, along 33 centuries. Numbers on the curve indicate the times of references for the inferred level: they are synthetically reported in Table 1 and extensively explained in Appendix 1. Dashed lines represent hypothesized subsidences: the first one connecting to the likely initial elevation, the second one showing the likely subsidence path in absence of the restoration works (points 4 and 6), the third one showing the likely uplift linked to 1538 eruption.**

| Number | Time | Event | Reference source | Reported by |
|---|---|---|---|---|
| 1 | 3.7 ka and after | Formation of the coastal bar | This paper | |
| 2 | 8th century BC | Subsidence of the via Herculea | Diodorus Siculus (Book IV) | Parascandola, 1943 |

| | | | | |
|---|---|---|---|---|
| 3 | 60 BC | Sergio Orata, owner of the 'Lucrino' lake fish farm, asked the Senate to have via Herculea repaired, because at around 2 m asl. Cesare repaired it | Parascandola, 1943 | |
| 4 | 37 BC | Agrippa raised the level of via Herculea | Strabone | Parascandola, 1943 |
| 5 | 12 BC | Abandonment of Portus Julius and Lucrino fish farming, because of accelerated subsidence of via Herculea | Aucelli, 2020 | |
| 6 | 496 AD | Theodoric, King of Gotes, repaired and raised level of via Herculea | Cassiodorus, Varia Book I | Parascandola, 1943 |
| 7-8 | 556 AD | Failed attempts to restore fish farming in the Lucrino lake: the level of Dam was too low | Parascandola, 1943 | |
| 9 | 1341-1348 | Petrarca and Boccaccio writings indicate via Herculea was about 5-6 m bsl | Boccaccio, 1355-1373 | Parascandola, 1943 |
| 10 | 15th century | Uplift starts, but Lucrino lake however disappeared and via Herculea never re-emerged | Several chroniclers of the time | Parascandola, 1943 |

**Table 1 - Sinthetic sketch of the main historical sources used to reconstruct the ground**
**deformations shown in Fig.3 (see Appendix 1 for more details).**

[Figure]

**Fig. 4 – Shaded reflief map of the coastal area of the Pozzuoli Bay based on high**
**multibeam bathymetry (Somma et al., 2016). Arrows indicate the submerged remains of the**
**breakwater pilae of the via Herculea.**

*[handwritten: Do you mean the whole caldera or only to the west of Pozzuoli? ?]*

**3.2.2 Ground movements at Pozzuoli** *[handwritten: Roman times to 1538.]*

While Via Herculea records the most ancient subsidence  *[handwritten: in the eastern]*, the best evidence for
subsidence in the Pozzuoli area, where maximum ground movements  *[have been]* recorded, comes from  *[Roman market place, Serapeo.]*
*[Recent]* Amato and Gialanella (2013) discovered  drilling  *[has revealed]*, four successively
superimposed floors, ranging from the Augustan age (31 BC-14 AD) to that of the Severi (193-235
AD),  indicating  *[a]* progressive subsidence  (Fig. 5). The  *[youngest]* floor
this epoch we will follow the historical traces of further subsidence and
*[Fig 6 shows the]*  time evolution of the approximate level of the *[uppermost]* 4th floor
*[CAPTION]*
*[It]* subsided below  sea level in the 5th century,  (about 200 years after
its construction during the Severi Age).  *[By the time it had]* 3.6 m bsl,  *[(around]* the
7th century AD) the columns  *[bare of sediments had covered the bare of]* (which formed the so-
called "fill" *[of]* Parascandola, (1947))
*[Colonade/ those]* lithodomes attached  *[the]* column  *[s near]* (between 3.6
and 6.30  *[by), creating s]* pitted band *[s]* above the *[about 2.7m thick]*
sedimentary  *[layers]* 2.70m. This process occurred until the 9th century AD, when
the fourth floor was located to a depth of 6.3 m below sea level  *[ ? [PUT ELSEWHERE] ]*
In the
same period,  ground subsidence caused  *[to flood]* by thermal and rain waters)  the
Agnano plain,  east of Pozzuoli,  *[where they formed new]* lake (Annecchino,
1931). This  *[s]* indicated *[s]* a general persistence of subsidence in the Pozzuoli area,
(Fig. 7a)  Appendix 2).  *[ing]* the
conclusion *[s]* by Morhange et al. (1999; 2006),  *[that an]* of several meters *[occurred]*
in the period 7th-8th century. *[(although Morhange et al. (2006) also questioned their previous interpretation).]*

*[handwritten margin bottom-left: Add reference to evidence for libration!]*

[Figure]

**Fig. 5 – Floors underlying columns of Serapeo (redrawn from Amato and Gialanella, 2013).**

**The dotted part of the column indicates the boring due to colonies of *Lithodomus Litophagus*.**

** Evidence for persistent subsidence comes from the Arab geographer Idisi (11th cntry)
and the historians Benjamin --- (12th cntry) and Nicolò J--- (13th cntrs), which

, clearly  *descrbe be* the morphology of Rione

Terra as a medieval castle surrounded by the sea on three sides,

 (Costa et al., 2022)  Appendix

2).  Boccaccio (),  *also* reported

 that the fisherman's wharf in the Bay of Pozzuoli  *had become* completely submerged  (Parascandola 1947; Table 2 & Appendix 2).

We can  *obtain*

 a more precise estimate of the depth below sea level reached by the 4th floor  *Serapeo's* *from*

 the painting "Bagno del Cantariello" (Fig. 7a), part of the famous Balneis Puteolanis of the

Edinburgh Codex of 1430 AD (Di Bonito and Giamminelli, 1992). The painting depicts the Rione

Terra encircled by vertical yellow tuff walls, from which the beach of Marina Della Postierla extends (towards the observer) to the base of the S. Francesco hill, the source of the thermal spring Cantariello (foreground) near the coast northeast of the submerged  *Serapeo*. Behind the visitors of the thermal spring, the painting clearly shows the upper part of the three marble columns of  *the Serapeo above*

 *sea level --* people fishing directly from the shore (Fig. 7b). From this painting *are also shown*

THIS WAS SHOWN IN Bellucci et al. (2006). Their work MUST be
referenced. The significance of Fig. 7 is NOT a new discovery &.

we can make a rough estimate of the portion of columns below the sea level at that time, taking into account that a significant part of the columns is submerged. Historical records from the 1750

excavations, (see further) indicate that the buried part of the columns amounted to about 10 m (see

Parascandola, 1947); the shallowest 2 meters of the excavations were formed by pyroclastic flow deposits of the 1538 eruption (see further paragraphs).

*Honestly, not citing previous studies here is wholly unacceptable. The authors are FULLY AWARE of this paper.*

[Figure]

Time (Years)

[revised manuscript text omitted]

*eventually (?)*

*Move text.*

*Put the comparison with previous models*
*both AFTER techn of dilucidy*
*what happen after 1538.*

more constructively by Agrippa in 37 BC, raising the surface of the Via Herculea with respect to the sea level (see  detailed explanation in Appendix 1).

*The second claim appears unrealistic*

, because  an uplift in the Monte Nuovo

*would have raised* *which did*

area higher than few meters, the Via Herculea  above the sea level (Fig.3d), *not occur*

not confirmed by the testimonies collected until 1430, which instead indicate the (Di Bonito and Giamminelli, 1992; Bellucci et al., 2006).

*Claim 3* — *subsidence continued beyond 1251*

[Figure]

**Fig. 9 – The uprise of the land (marked by the two arrows on the sides) was observed and**

**described by Loffredo Ferrante in 1530:** ***"the sea was very close to the plain which was at the foot***

***of the Starza hill"*. In this context, the 4th floor of the Serapeum had reached a height of**

**approximately 4 m above sea level.**

*I think this can be omitted. Implicit from previous paragraph.*

*Our reconstruction indicates*

on reliable historical documentation, we  that the hypothesis that maximum submergence depth of the 4th floor of the Serapeum was reached in the 9-

10th century, proposed by Parascandola (1947) and Amato and Gialanella (2013), is not realistic. Nor it is the hypothesis by Di Vito et al. (2016), who place the date of the transition between subsidence and uplift in the 13th century and precisely in 1251.

*They* *consistent with the interpretation* *of Dvorak & Mastrolorenzo (1991) and* *Bellucci et al (2006)*

Our findings, dating the starting phase of uplift around 1430, are  supported by the occurrence of the first  powerful earthquake in 1448 (Colletta, 1988: see also next paragraph), which induced King Ferdinand I of Aragon to suspend the so-called "fuocatico" (a mediaeval tax collected for each fire lit by a family unit; see Colletta, 1988). We know , from the recent unrests, that earthquakes only occur during  uplift  at Campi Flegrei (Troise et al.,

*to citizens*

2019). It is also well known that, between 15.. nd 1511, the municipality of Pozzuoli granted the

*new* lands that emerged, as a result of the increasingly "drying up sea" (Fig. 9), requesting them (Parascandola, 1947).

(partly already used by Bellucci et al., 2006 and Troise et al.,

*SIMPLIFY.*

MORE

2007), support their interpretation,  of new data.  | When did uplift begin before the eruption of R. Nuovo in 1538. | In particular,

The next important question is : was the 4th floor of the Serapeum above sea level as early as at the beginning of 16th century? Parascandola (1947) answered this question through a sentence found in an account by Loffredo Ferrante from 1580: *In 1503 the sea was very close to the plain which was*

*at the foot of the Starza hill* (Fig. 8). So, it can be deduced that the floor of the Serapeum  in 1503, was just above sea level, that is, it had risen about 10m in about 73 years, with a rate of 136 mm/y.

There is clear evidence that the uplift phase continued until 1538, when the eruption occurred. The maximum uplift occurred in the Pozzuoli area, close to the Rione Terra cliff,  and had reached / reaching as much as 5-6 m by 1538 (Fig. 6)

At Averno to the west,  uplift,  was unable to  raise above sea level the Via Herculea/

to the east of Pozzuoli,

At Nisida island, the pier did not emerge above sea level (Parascandola 1947). Hence it is likely that the uplift phase had a bell-shaped trend, very similar to what we have seen in the recent unrests,

GOOD

→ NEW  CLOSING PARAGRAPH : Local large uplift at future site of M. Nuovo immediately (48 hours?) before the eruption & (Parascandola, 1947) --- dyke.

**1. Ground movements after the 1538 eruption**

The period between the end of the 16th century and the beginning of the 17th century lacks  written documentation about the ground movements at Pozzuoli. It is likely that  Lafter the eruption, subsidence earliest,  in  shows

The  by Cartaro,  1584 (Fig. 10a),  the Rione Terra in the foreground,
     Contemporary paintings provide constraints on  when subsidence began.

* Cite Dvorak & Mastrolorenzo (1991).

with the Neronian pier  almost completely above sea level, which means for about 5-6 M.?

[Figure]

m.

**Fig. 10 – a) Engraving by Cartaro (1584) showing the Neronian pier at the base of the Rione**

**Terra, emerging from the sea for 5-6m, showing 10 of the 15 piles of which it was made up in**

**roman epoch, b) The remains of the pier piles, without the upper arches, highlighted in an**

**engraving from the mid-18th century, c) Detail of the same piles highlighted in another**

**engraving from the same period, where the height of the 1-2m piles is observed in more detail,**

**subject to marked erosion**

The pier also appears still partially complete, with about half pylons still connected with arches (*Opus*

*Pilarum*). In comparison, paintings from the middle XVIII century (Fig. 10b,c)  Show the pier completely destroyed, and  submerged. The painting  in Fig. 10c  shows the pylons in more detail, allowing  in their height to be estimated at 1-2 m asl. Fig.

11  from Hamilton ( ), shows  Similar for ruins  the Neronian pier

[and, in addition,  that floor of the columns of Serapis Temple, with → (LATER)

] the Serapeo

   as

 Hence it appears that the pier subsided by about 4-5 M
      from 1580 to 1750.

[Figure]

Fig. 11 – a) View of the Gulf of Pozzuoli and the Cape Miseno peninsula (Hamilton 1776).

Both the remains of the Neronian pier and the newly excavated Serapeo are also visible

[Figure]

Fig. 12 – Illustration of Serapeum, as excavated in the three-year period 1750-1753. It can be noted that the height of the lighter parts of the columns, including the pitted band of the lithodomes, is preserved by oxidation, because packed by the just removed sediments. The darker upper part, oxidized since staying outside the cover, has a height of approximately

2.50m, estimated on the same figure. This leads us to consider that the pack of sediments removed had a thickness of approximately 10m, that is, the height of the hill where the *vineyard*

*of the three columns* was located before the excavation (Niccolini, 1842).

Fig. 11 also indicates that the floor of the Serapeo was almost at the same

level as the pier in 1750.

Its level in 1538

*therefore* can be estimated at 5 – 6 m. above sea level (Fig. 6), *and* while in 1750 it should be at about 1m  *asl in 1750*

with an estimated subsidence *in 1580 ✓ O.K.* -1750 of about 4-5 m. This  estimation is  *reported by P— (1947)*

*consistent with*  measurements by Niccolini (1846), who found the 4th floor of Serapeo to have a height above sea level varying in the range 0.9 - 0.6m throughout the 18th century.  During the  excavations (Fig. 12) the *of DATE – DATE,*

*therefore, the  was at* floor  approximately at 0.7 m above sea level.

*The initial 4-5 m of subsidence -oc. may have slowed*

*after* 1580,  *may* have  around the middle of the 17th century, *after* it  had a value of 2 - 3 m,  *until* the end of the

*WHAT IS YOUR EVIDENCE?? This conclusion has appeared from where? ?* *CHECK FIG. 13.*

It is also interesting to compare the average subsidence rate before 1430 with that observed  1538

*and* till 1950. The overall rate of subsidence after 1538 is about 2 cm/year, almost double with respect to *between*

that observed before 1430. However, when excluding the first phase of sharp subsidence occurred just after the 1538 eruption, the subsidence rate becomes very similar to that observed since the roman era until 1430. — *See comment on Fig 13.*

In particular, regarding the post-1538 subsidence phase, the data shown,

Data from the levelling surveys were still provided also during the occurrence of the most recent unrest phases, i.e. in 1950 -

52, 1969 – 72, 1982 – 84 and until 2001. Since 2001, continuous measurements *have been*  provided by GPS *Stations,*

(RITE, see Fig. 13b,c) installed at Rione Terra (Del Gaudio et al 2010). *Fig 13)* *including Station RITE*

? ← Since the 1850s, survey  data have recorded  ground movements at Campi Flegrei with increasing precision.  The Military Geographic Institute (IGM) started frequent high precision levelling surveys in 1905. *in particular,*

[Figure]

PLEASE ADD TRENDS
FROM
DVORAK & MASTROLORENZO (1991)
and
BELLUCCI et al (2006)
Since both have been
compared with new
interpretation here.

Could uplift
actually have
started here?
Is this possible?

Fig. 13 a) Reconstruction of the ground level of the Serapeum floor, with respect to the mean sea level (blue line), as proposed by Parascandola (1947); b) Reconstruction of the Serapeum floor ground level, recently proposed by Di Vito et al. (2016); c) Reconstruction of the ground level of the Serapeum IV floor, since III century A.D. to present, inferred by this study. Each point in the diagram corresponds to an appropriate historical indication reported in Table 1

and in the Appendix 2.

## 2. Schematic model for the preparatory phases of the 1538 eruption

### 2.1 Dynamics of the resurgent block in response to temperature and pressure perturbations

The ground deformation at Campi Flegrei,  [before and after] the 1538

eruption,  [appears to have been] concentrated in a small area of few km  [in] radius around Pozzuoli,  [similar to that observed during unrest since 1970]

(De Natale et al., 2001; 2006; 2019). Such a concentration agrees with the presence of a resurgent block.

Evidence for  [movement during unrest] resurgent block the  [was first highlighted] by De Natale and Pingue (1993).  [who] pointed out that the concentration of the uplift in a small area, the high uplift values, and the invariance of the uplift and subsidence shape, as well as of the seismic area,  [was consistent with] the up and down movement of a bordered by ring faults  (see also De Natale et al., 1997; Beauducel et al., 2004; Troise et al., 2003; Folch and Gottsmann, 2006) Some authors proposed that ground deformations could be explained also without any effect of  faults (Berrino et al., 1984;  Amoruso et al., 2008, 2014; Woo & Kilburn, 2010); however, most of these models required some 'ad hoc' distribution of rock rigidity, sometimes not realistic (see De Natale et al., 1991), or required an unrealistic constancy of the source geometry able to explain the remarkable constancy, during several decades or centuries, of the shape of deformation during both uplift and subsidence (see De Natale et al., 2006). All of these models, in addition, do not explain the peculiar shape of the seismic area, being almost elliptical around the most uplifted area.

Active high-resolution reflection seismic surveys have imaged the presence, in the Gulf of Pozzuoli, of an inner resurgent antiformal structure or

"block" bounded by a 1-2 km wide inward-dipping ring fault system associated with the caldera border, whose limits have been also documented by the survey (Sacchi et al., 2014 Steinmann et al,

2016; Sacchi et al., 2020a). Further constraints for the extent on-land of the resurgent block come from stratigraphic evidence. In particular, the old well CF-23, drilled in the Agnano area, presents about 900 m of NYT deposits, topped by only ͏ m of more recent deposits (Rolandi et al. 2020b).

The presence of uplifted, thick layers of NYT, characterizes the stratigraphy of all the wells contained in the resurgent block (Fig.14a,b,c), thus allowing to map its extent on-land, although only the CF-

*Just show your evidence that to support a resurgent block. Your  point is that the interpretation is consistent with observation. Don't worry about other models.*

23, by far the deepest one, clarifies the whole thickness of the NYT deposits in the resurgent area (Fig. 14a,c,d).

The extent of the resurgent block on-land appears also reasonably well defined by a gravimetric maximum (Capuano et al., 2013).  *would also favour*

resurgent block, mostly detached from the external caldera rocks,  the almost constant, highly concentrated shape of ground displacement, during both uplift and subsidence.

. Fig. 15a-c shows how the resurgent block is well  *shown* by passive seismic data (Fig. 15b, c) and by earthquake locations (Fig. 15a). (Troise et al., 2003)

The presence of the central, resurgent block significantly affects the dynamical behavior in response to temperature and pressure perturbations. This is particularly evident in the central, most deformed and seismic area, where the shallow crust involves approximately 1.5 km of lithoid tuff. This contradicts substructure models proposed by various authors (Rosi and Sbrana, 1987; Vanorio et al.,

2002; Lima et al., 2021; Kilburn et al., 2017/2023), which  assume a thick shallow layer of loose pyroclastics from recent eruptions, typically represented by the stratigraphy of well SV1 (see Fig.

14e). But THESE STILL USE THE SAME PHYSICAL PROPERTIES AS OTHERS!

The physical state of the shallow structure within the resurgent block can be inferred by seismic tomography analyses presented by several authors (e.g. Aster and Mayer, 1998; Vanorio et al., 2005;

Vinciguerra et al., 2006; Battaglia et al., 2008; Calò and Tramelli, 2018). These analyses consistently indicate a high Vp/Vs ratio centered below Pozzuoli town down to 1-2 km, interpreted as highly water saturated tuff.

Please clarify. You state that the upper 1-2 km consist of water-saturated tuff.  Are you saying that the material did NOT come from recent eruptions? Be careful. The key feature as far as deformation is concerned - is the physical resistance of those layers. The specific terminology can  be changed ("loose pyroclasts") without questioning the validity of the analyses. "Loose pyroclastics" does NOT mean . Perhaps unconsolidated layers to depths of 1-2 km! I agree the  description is misleading and ought to be changed. The previous interpretations work even when changing the name to "water saturated tuff."

[Figure]

Fig. 14 - a) Location of the wells explored within the resurgent tuff block, as reported in literature; b) Stratigraphy of the CF23 (S10) well, within the resurgent block; c) Stratigraphy of the SV-1 well, outside the resurgent block, which highlights a stratigraphy where the NYT

tuff blocks are not present with significant thicknesses; d-e) Profiles in the resurgent block which highlight the shallow depth of NYT because of the resurgence.

Of particular significance is the work by Vinciguerra et al. (2006) which compared the results of seismic tomography with laboratory tests. They demonstrated that the tuffs present in the central area of the Campi Flegrei caldera can be either water or gas saturated, and that inelastic pore collapse and cracking produced by mechanical and thermal stress can significantly alter the velocity properties of

Campi Flegrei tuffs at depth. The effect on velocities becomes significant when the temperature rises

*I don't see the added significance of lines 561-574, Vinciguerra et al. (2006) could be added to references cited on line 551.*

sufficiently to induce physical changes, such as volume change and the generation of free water
associated with the dehydration of zeolite phases. This can lead to thermal crack damage (see also
Chiodini et al., 2015; Moretti et al., 2018), further affecting the dynamic behavior of the area. At
higher depths, the well CF-23 indicates the presence of pyroclastic deposits from a depth of
approximately 1.5 km to at least 1.8 km, where a temperature of 300°C was measured (Fig. 14b).
Likely, at even greater depths of about 3km, marine silt and clay layers induce silica mineralization
and the formation of low-permeability horizons. Due to the high temperatures, estimated to be at least
400°C, these layers undergo thermal alteration, forming a thermo-metamorphosed layer (Fournier,
1999; Lima et al., 2021; Cannatelli et al., 2020).

Is important to note that Battaglia et al. (2008) interpreted a low Vp/Vs body, extending to about 3–
4 km of depth, as due to the presence of fractured overpressured gas-bearing formations, confirming
the data of Vanorio et al. (2005). This depth range of 3-4 km likely represents a primary accumulation
zone
In addition.

*[handwritten margin notes: "? do you mean 'or depths of about.'"; "accumulation of WHAT?"; "MOVE TO Line 553."; "coinciding with the depths for a Pressure source inferred from ground deformation (REFS)."]*

14°05' E          14°10' E

Km                                                    a)

| | |
Border of the NYT caldera (after Sacchi et. al., 2014; Steinmann et al., 2018)

Base of the NYT resurgent dome (after Sacchi et al., 2014; Steinmann et al., 2016)

NYT ring fault zone (after Sacchi et. al., 2014; Natale et al., 2022)

Edge of the Pozzuoli Infra-titral Prograding Wedge (after Sacchi et. al., 2014)

Shallow intrusion (after Sacchi et. al., 2014; Steinmann et al., 2016)

Epicenters of earthquakes recorded between 2022 and 2024

Fault multichannel seismic profile

Pozzuoli
Baia
Bagnoli
Epitaffio Valley
Fig. 15b
Bagnoli Valley
Nisida
Pozzuoli Bay
Ring-fault zone
C. Miseno
Fig. 15c
Nisida bank
Miseno bank
P. Palummo bank
M. Dolce-Pampano bank

40°50' N
40°45' N

[revised manuscript text omitted]

*[handwritten left margin:]* IMO is pure SPECULATION and not a consequence of new data here.

*[handwritten right margin:]* rather data from a different paragraph. I think you are over-interpreting either removing this paragraph. I think other sources

*[handwritten bottom:]* Earthquakes → fractures ✓ but not overpressured gas.

*[handwritten bottom right:]* Well – it was 3–4 km on lines 575/6. CLARIFY

**[Handwritten margin notes, top]**

Really – distinguishing
the a difference from
to 25 km is well
within observational error.

← The proposed depth of c.3 km
is to be consistent
with the ground deformation data.

Near the base of the hydrothermal system?

[revised manuscript text omitted]

- The  *long-term seismic precursors* started in 1448  1468 - 1470,  (Io = VII) (Guidoboni and Ciuccarelli, 2011; Francisconi et al., 2019) (Fig. 19a – interval A),   culminated  fumarolic-hydrothermal activity  Solfatara   widespread damage to the vegetation,   This  a broadening of the area affected by intense degassing (Francisconi et al., 2019).   seismic phase was reported (Guidoboni, 2020), with maximum intensity Io = IV - V,   ground uplift  rate. This period  with a strong seismic phase  in October 1498, reaching  maximum intensity (Io = VII).  low-intensity seismic  then followed  1499  1503 (maximum intensity Io = V) (Fig. 19a – interval A). Such a long-term precursory phase  as mainly  degassing,  the deep magma chamber  increasing pressure in the shallow layers of the geothermal system, without significant contribution from  magma intrusion at shallow depth.

-  a new phase of *Medium-term precursors*    seismic events in 1505 and 1508,  of higher intensity  (maximum intensity Io = VIII) (Guidoboni and Ciuccarelli, 2011).  faster ground uplift , resulting in serious damage to buildings and several casualties. This seismic phase could have been caused by either a higher stress associated with increased uplift  or magma intrusion, from the deep magma chamber into shallower levels. This intrusion could have produced higher stress resulting in seismic activity of greater intensity. Although it is obviously difficult to identify, from historic sources alone, the respective roles of the deep degassing into the hydrothermal system versus shallow magma intrusion, we believe that the reported evidence of vegetation damage and increased degassing in the first phase, and the increase of earthquake intensity in the second phase, indicate respectively a main contribution of degassing perturbing the hydrothermal system, in the first phase, and of shallow magma intrusion in the second phase. This phase ended in 1520, ⸞h a medium intensity earthquake (Io = V-VI) (Fig. 19a – interval B)..

*Handwritten annotations:*
- ? Mercalli Scale ? Intense seismicity in  from
- with Vigorous
- accounts that caused
- In surrounding areas
- May indicate
- Interpretation Not Data. "2 km NE of Pozzuoli" (check)
- in 1475
- by accelerating for the next 20 years.
- followed by
- ended
- a
- in
- from
- to
- can be attributed
- to the
- of from
- requiring a
- emerged
- with
- than before
- ed
- caused
-  Do you mean increasing gas pressure?

[Figure]

[Figure]

**Fig. 19 – a) Reported earthquakes occurred before and after the 1538 eruption (after**
**Guidoboni and Ciuccarelli, 2011). The computed intensities of these earthquakes have been**
**converted in magnitudes using the considerations made in the appendix 3. b) Highest magnitude**
**earthquakes (M≥3.5) occurred since 1950 to present.**

After 16 years of relative seismic quiescence,  possibly characterized by low-intensity earthquakes not
reported in chronicles, a short-term precursory phase began in 1536. It started with continuous
seismicity, without major damage (Io = III -IV), continuing with similar features until the early 1537.
It is possible that this last seismic phase, characterized by relatively low magnitude, was caused by
low-frequency seismicity, resulting from magma oscillations during the fractures opening (see
Chouet, 1996). This seismicity became more frequent just before the eruption. In February of the
same year, the seismic activity peaked with stronger events (Io = VI - VII), accompanied by an increase in the fumarolic activity at Solfatara. This provides evidence that this seismicity could be again related to perturbations in the hydrothermal system. A final increase in seismic activity (Io =

VIII), began in mid-June 1538, accompanied by a localized, significant additional ground uplift at the eruption site, located 3 km away from the center of previous maximum uplift (Fig. 19a – interval

C) (Parascandola, 1943, Rolandi et al., 1986; Guidoboni and Ciuccarelli, 2011; Guidoboni, 2020).

-

Cannot be justified yet. Why not stick to the EFFECTS of the seismicity ? which occurred

**6.2 The post-eruption seismicity**

We will now consider the seismic phase following the eruption just described, which we will indicate as the *aftereffect of the 1538 eruption*.

It began in 1564 with earthquakes of medium intensity (Io = V - VI), followed by a phase of lower intensity 2 years later. In 1570 seismic intensity increased (Io = VI - VII), causing damage to the buildings of the city of Pozzuoli. Between 1575 and 1580 a new phase of low seismic intensity began, culminating, in 1582, with two earthquakes, respectively of intensity Io = VII – VIII.

These earthquakes caused partial collapses in several houses and serious damage to churches and buildings, as well as numerous casualties (Parascandola, 1943; Guidoboni e Cucciarelli, 2010;

Guidoboni, 2020).

**4. Comparison of precursory phases of 1538 eruption with current unrest**

evolution of the ground movement and seismicity recent *[handwritten: ground movement and seismicity]*

Common features between the medieval and present-day unrest

*[handwritten: From Our reconstruction of historical unrest, that has identified features common to the medieval]*

The main similarity is that the seismicity, , has been clearly correlated *[handwritten: and present unrest,]*

both with the total uplift and the uplift rate; it is practically absent in periods of subsidence (Dvorak and Gasparini, 1991; Kilburn et al., 2017; Troise et al., 2019).

We found, in particular, that seismicity of period 1950-2024 is on the same order than the period

1430-1503, , *[handwritten: which]* was the first phase of preparation *[handwritten: before]*

the 1538 eruption. Although the  *[handwritten: 10 m]* of uplift in the period 1430-1503, , was more than double than the total uplift recorded since 1950-2023, , the seismicity in the two periods has been remarkably comparable. The maximum magnitude, M=4.4 recently occurred on

[revised manuscript text omitted]

---

## Author Response (AR2)

Dear Editor,

We are resubmitting a revised version of our paper '**The 1538 eruption at Campi Flegrei resurgent caldera: implications for future unrest and eruptive scenarios**', which now includes all the revisions required/suggested by the reviewer who required minor revision. Where the reviewer required clarifications on the text, we have tried to make it clearer; where the suggestions were not compelling, we decided to eventually accept them or to modify in order to make the content clearer, preserving the basic concepts and ideas.
We hope now the paper content can be accepted for publication; we are however ready to eventually improve some figures, if it would be required.

Best Regards

Giuseppe De Natale